# SEPTIN2 suppresses an IFN-γ-independent, proinflammatory macrophage activation pathway

Beibei Fu[1,8], Yan Xiong[1,8], Zhou Sha[1], Weiwei Xue [2], Binbin Xu[2], Shun Tan[3], Dong Guo[1], Feng Lin[1], Lulu Wang[1], Jianjian Ji[4], Yang Luo [5] ✉, Xiaoyuan Lin [6,7] ✉ & Haibo Wu [1,5] ✉

Interferon-gamma (IFN-γ) signaling is necessary for the proinflammatory activation of macrophages but IFN-γ-independent pathways, for which the initiating stimuli and downstream mechanisms are lesser known, also contribute. Here we identify, by high-content screening, SEPTIN2 (SEPT2) as a negative regulation of IFN-γ-independent macrophage autoactivation. Mechanistically, endoplasmic reticulum (ER) stress induces the expression of SEPT2, which balances the competition between acetylation and ubiquitination of heat shock protein 5 at position Lysine 327, thereby alleviating ER stress and constraining M1-like polarization and proinflammatory cytokine release. Disruption of this negative feedback regulation leads to the accumulation of unfolded proteins, resulting in accelerated M1-like polarization, excessive inflammation and tissue damage. Our study thus uncovers an IFN-γ-independent macrophage proinflammatory autoactivation pathway and suggests that SEPT2 may play a role in the prevention or resolution of inflammation during infection.

Macrophage polarization, first categorized as classical activation (M1) and alternative activation (M2) dichotomy, is a highly plastic and dynamic process that is delicately regulated by extrinsic factors and the microenvironment[1]. In the previous M1-M2 model proposed by Mantovani and colleagues, M1-dominant macrophages (e.g., M[IFN-γ + LPS/TNF]) and M2-dominant macrophages (e.g., M[IL-4/IL-13], M[IC + TLR/IL-1R ligands] and M[IL-10]) are linked with T-helper 1 (Th1) and T-helper 2 responses, respectively[2]. However, due to the complexity of the diverse mediators involved, a multipolar view of the macrophage polarization paradigm from an immunological perspective is being established. With M1 and M2 macrophages at opposing ends and other groups of macrophages in between, the polarization spectrum is extended according to the substantial shifts in gene expression depending on the specific stimuli[3]. Any agonist has the potential to alter gene expression and thus transition macrophages from one steady state to another[4]. Moreover, macrophages encountering the stimulus relevant to cardiovascular disease may produce mediators that lie outside the M1-M2 spectrum, mediating other unknown polarization states[5]. Undoubtedly, macrophage polarization behavior is far more complicated than what we know at present.

Macrophage polarization is inseparable from the processes of resolving inflammation[6,7]. In specific pathological processes,

[1]School of Life Sciences, Chongqing University, 401331 Chongqing, China. [2]School of Pharmaceutical Sciences, Chongqing University, 401331 Chongqing, China. [3]Chongqing Public Health Medical Center, 400036 Chongqing, China. [4]Jiangsu Key Laboratory of Pediatric Respiratory Disease, Institute of Pediatrics, Nanjing University of Chinese Medicine, 210023 Nanjing, China. [5]Center of Smart Laboratory and Molecular Medicine, School of Medicine, NHC Key Laboratory of Birth Defects and Reproductive Health, Chongqing University, 400044 Chongqing, China. [6]Institut für Virologie, Freie Universität Berlin, Robert-von-Ostertag-Str. 7-13, 14163 Berlin, Germany. [7]Department of Clinical Microbiology and Immunology, College of Pharmacy and Medical Laboratory, Army Medical University (Third Military Medical University), 400038 Chongqing, China. [8]These authors contributed equally: Beibei Fu, Yan Xiong ✉e-mail: luoy@cqu.edu.cn; linxiaoyuan23@163.com; hbwu023@cqu.edu.cn

macrophages often exhibit a continuous but predisposed polarization spectrum, and this predisposition has an important significance for the development of inflammation[8]. The polarization of M1 propensity is accompanied by the upregulation of inducible nitric synthase (iNOS) activity and proinflammatory factor expression, which can activate the immune system and kill pathogens in the early stage of infection[9]. However, excessive M1 polarization induces a large amount of leukocyte infiltration, and the tissue is flooded with inflammatory mediators, proapoptotic factors, and degrading matrix proteases, which can disassemble the tissue and cause damage[10]. Consequently, balanced regulation of M1 polarization in a specific immune environment is critical for controlling infection and maintaining homeostasis.

Generally, the process of macrophage polarization to the M1 direction requires the induction of interferon-gamma (IFN-γ) as a priming signal. IFN-γ is secreted by activated lymphocytes such as CD4[+] and CD8[+] T cells, gamma delta T cells, natural killer cells, and other immune cells[11]. IFN-γ is recognized by IFN-γ receptors on the surface of macrophages. Cellular effects of IFN-γ include upregulation of pathogen recognition, antiviral status, activation of microbicidal effector functions and immunomodulation[12]. IFN-γ stimulation directly triggers the activation of the downstream JAK-STAT pathway, thus emerging as the global paradigm for class II cytokine receptor signal transduction[13]. Additionally, IFN-γ indirectly increases the expression of proinflammatory factors by enhancing the response of macrophages to other stimuli-that is, the "priming" effect which is mainly reflected in the disruption of the IL-10-STAT3 and Hes1-Hey1 negative feedback loops, as well as the elimination of endotoxin tolerance by promoting TLR signaling-induced chromatin remodeling[14–17]. Thus, IFN-γ has long been considered necessary for the M1-like activation. Mice lacking functional IFN-γ are extremely vulnerable to various bacterial and viral infections[18]. Although the role of IFN-γ is important and unquestionable, IFN-γ-independent M1-like activation still exists[19,20]. Canna et al. reported that IFN-γ-knockout mice showed significant macrophage activation and proinflammatory responses in a model of macrophage activation syndrome (MAS). After being administered with CpG and IL-10R blocking antibody, mice incapable of producing IFN-γ still developed most aspects of fulminant MAS-induced immunopathology, with elevated IL-6 and IL-12 levels in serum[19]. This phenomenon implies that TLR9 activation has the potential to trigger macrophage overactivation in the absence of IFN-γ. Recently, Van Dis et al. reported that CD4[+] T cell-derived GM-CSF drives macrophage M1-like polarization in the absence of IFN-γ signaling. This IFN-γ-independent activation is nitric oxide-independent and contributes to the control of *Mycobacterium tuberculosis* infection. It requires the activation of transcription factor HIF-1α and a shift to aerobic glycolysis[20]. However, how HIF-1α leads to polarization in the absence of IFN-γ remains an open question. This evidence suggests that macrophages can be activated through other pathways and induce proinflammatory factors in the absence of IFN-γ stimulation, but the stimuli and the underlying mechanisms involved in this process have yet to be fully elucidated.

SEPTINs belong to a family of GTP-binding proteins that are recognized as the fourth component of the cytoskeleton. SEPTINs have been implicated in the regulation of molecular mechanisms related to human diseases, such as cancer, neurological diseases, and infections[21]. Studies of host-microbe interactions have highlighted the significance of SEPTINs in bacterial invasion. SEPTIN2 (SEPT2) participates in the assembly of filamentous assemblies called "septin cages". In the case of host cell invasion by bacteria, septin cages are detected as bundles around the bacterium and restrict pathogen cell-to-cell spread and inhibit bacterial division through recruitment of autophagic and lysosomal machinery[22,23]. Recently, an increasing number of studies have indicated that SEPTINs play key roles during inflammation[24]. In an in vitro model of *Shigella flexneri* infection, Lee et al. reported that knockdown of SEPTINs in human monocytic THP-1

cells increased inflammasome activity and host cell death[25]. Mazon-Moya et al. discovered that deletion of SEPTINs significantly increased the secretion of proinflammatory cytokines using a zebrafish model[26]. These studies suggest that SEPTINs may participate in the restriction of bacterial infection and inflammatory disorders. In addition, the role of SEPTINs has been demonstrated in viral infections, including vaccinia virus[27], hepatitis C virus[28], influenza A virus[29], human herpesvirus 8[30], and Zika virus[31]. However, an understanding of the role of SEPTINs beyond "cage forming" is still limited, and the precise function of a particular SEPTIN in the restriction of virus-induced inflammation remains unknown.

In this work, we show that SEPT2 participates in antiviral activity by promoting IFN-γ-independent M1 polarization. Deletion of SEPT2 reduces the acetylation modification of heat shock protein 5 (HSPA5), thereby impairing the unfolded protein clearance ability in virus-infected macrophages. The continuous accumulation of endoplasmic reticulum (ER) stress thus results in M1-like hyperpolarization and excessive inflammation. Our study represents a new pathway for M1 polarization that is independent of IFN-γ signaling, which provides new insights into macrophage activation and highlights the role of SEPT2 from the perspective of antiviral immunity.

## Results
### SEPT2 is involved in IFN-γ-independent macrophage proinflammatory activation

To systematically investigate regulatory factors in the IFN-γ-independent proinflammatory activation of macrophages, we carried out high-content screening (HCS) of 1220 genes associated with immune responses in IFNGR1-deficient immortalized bone marrow-derived macrophages (iBMDM, kindly provided by Dr. Feng Shao, National Institute of Biological Sciences, Beijing, China) in response to infection with vesicular stomatitis virus (VSV) and herpes simplex virus type 1 (HSV-1). In the screening, on the basis of increased/decreased GFP intensity in iBMDMs with stable expression of iNOS promoter::GFP, we identified 34 genes (14 increased and 20 decreased GFP intensity in the VSV infection group) and 11 genes (3 increased and 8 decreased GFP intensity in the HSV-1 infection group) putatively involved in the regulation of IFN-γ-independent macrophage activation (Supplementary Data 1, Supplementary Fig. 1a). Surprisingly, we found that knockdown of SEPT2, a SEPTIN family member, increased GFP intensity in both groups (Supplementary Fig. 1a), implying a potential role of SEPT2 in the negative regulation of IFN-γ-independent macrophage activation.

To investigate the role of SEPT2 in macrophage activation and the host response to infection, we generated myeloid cell-specific SEPT2 knockout (*Sept2*[fl/fl] *Lyz2*-Cre) mice. In the absence of viral infection, *Sept2*[fl/fl] *Lyz2*-Cre mice and their littermates showed similar body weights (Supplementary Fig. 1b), and there was no significant difference in the numbers of macrophages isolated from spleen and lung (Supplementary Fig. 1c, d). These results indicated that *Sept2*[fl/fl] *Lyz2*-Cre mice were phenotypically normal. In the experimental design, inhibition of IFN-γ signaling was performed in mouse models by intraperitoneal injection of αIFN-γ antibody as previously described[32]. By detecting the secretion of IFN-γ in bronchoalveolar lavage fluid (BALF), we confirmed that SEPT2 did not affect the production of IFN-γ (Supplementary Fig. 1e). Compared with the *Sept2*[fl/fl] littermates, the survival time of *Sept2*[fl/fl] *Lyz2*-Cre mice infected with 1×10[7] PFU VSV was significantly shortened (Fig. 1a). By performing H&E staining and ELISA at 7 days post-infection (dpi), we found that *Sept2*[fl/fl] *Lyz2*-Cre mice showed more severe inflammation in the lungs (Fig. 1b, c) and higher cytokine levels of TNF-α, IL-1β, IL-6 and IL-12 p40 in the BALF than control mice (Fig. 1d). These data indicated that SEPT2 deletion led to excessive inflammation during VSV infection in the absence of IFN-γ signaling. In order to discover the reason of this phenomenon, we examined whether SEPT2 affected viral replication and found that

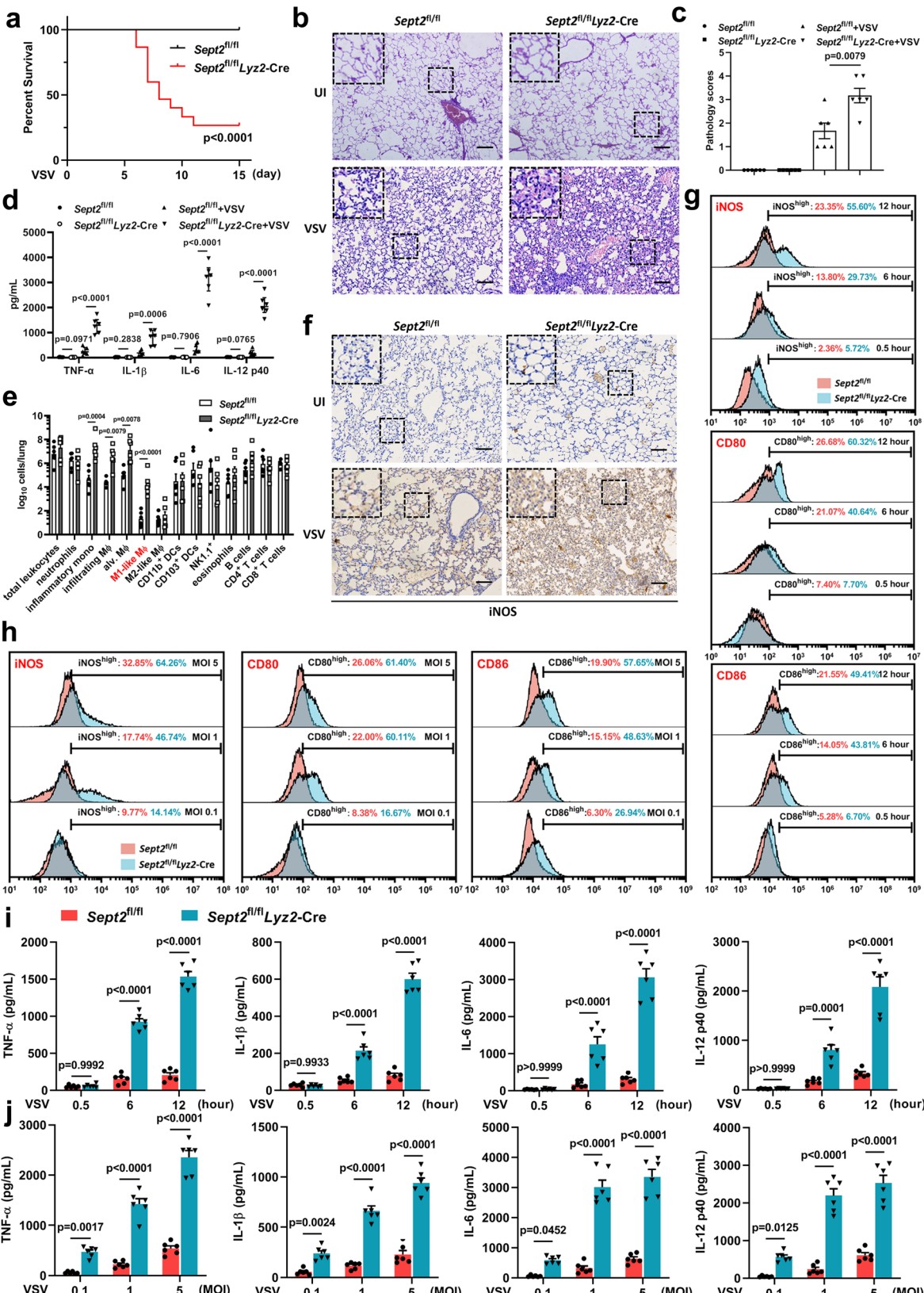

SEPT2 deficiency did not result in increased viral replication in vivo or in vitro (Supplementary Fig. 1f, g). We subsequently calculated the immune cell clusters in the lung tissues of infected mice by flow cytometry (gating strategy defined in Supplementary Fig. 1h). We showed that the numbers of inflammatory monocytes, infiltrating macrophages, and alveolar macrophages were increased significantly

in *Sept2*fl/fl *Lyz2*-Cre mice compared to control mice (Fig. 1e). Further analyses revealed that the M1-like macrophage cluster, but not the M2-like macrophage cluster, was increased in the cell clusters (Fig. 1e). These data suggested that the activation of macrophages toward the M1-like phenotype may be involved in SEPT2-regulated inflammatory responses. Immunohistochemistry data collected from mouse lung

**Fig. 1 | SEPT2 regulates IFN-γ-independent macrophage proinflammatory activation. a** Survival of *Sept2*<sup>fl/fl</sup> *Lyz2*-Cre mice (*n* = 15) and their littermates (*Sept2*<sup>fl/fl</sup>, *n* = 13) after intraperitoneal infection with $1 \times 10^7$ PFU VSV. Daily injection of αIFN-γ (12 mg/kg) from 1 day before infection to the end of the experiments was performed to block IFN-γ signaling. **b, c** H&E staining (**b**) and the pathology scores (**c**) of lung lesions in mice at 7 dpi. Scale bar = 400 μm. *n* = 6 in each group (**b, c**). **d** ELISA analysis of proinflammatory cytokines (TNF-α, IL-1β, IL-6, and IL-12 p40) in mice BALF at 7 dpi. *n* = 6 in each group (**d**). **e** Flow cytometry analysis of innate immune cell populations in mice lungs at 7 dpi. *n* = 6 in each group (**e**). **f** Immunohistochemistry analysis of iNOS in mice lungs at 7 dpi. Scale bar = 400 μm. *n* = 6 in each group (**f**). **g, i** PMs obtained from *Sept2*<sup>fl/fl</sup> *Lyz2*-Cre and *Sept2*<sup>fl/fl</sup> mice were infected with VSV (MOI = 1). The iNOS, CD80, and CD86 levels were detected by

flow cytometry (**g**), and secretion of proinflammatory cytokines was detected by ELISA (**i**) at 0.5, 6, and 12 h post-infection. The gating of iNOS<sup>high</sup>, CD80<sup>high</sup>, and CD86<sup>high</sup> populations was determined against those of the uninfected control. *n* = 3 in each group (**g**). *n* = 6 in each group (**i**). **h, j** PMs obtained from *Sept2*<sup>fl/fl</sup> *Lyz2*-Cre and *Sept2*<sup>fl/fl</sup> mice were infected with VSV at MOI of 0.1, 1, or 5, respectively. The iNOS, CD80, and CD86 levels were detected by flow cytometry (**h**), and secretion of proinflammatory cytokines were detected by ELISA (**j**) at 12 hours post-infection. *n* = 3 in each group (**h**). *n* = 6 in each group (**j**). Data are shown as Kaplan-Meier curves (**a**) and the mean ± s.e.m. (**c, d, e, i, j**). Log-rank (Mantel−Cox) test (**a**) and one-way ANOVA followed by Bonferroni post hoc test (**c−e, i, j**) were used for data analysis. UI uninfected. Source data are provided as a Source Data file.

tissues confirmed that SEPT2 deficiency led to high levels of iNOS, an M1-like activation marker (Fig. 1f). Furthermore, we analyzed peritoneal macrophages (PM) obtained from *Sept2*<sup>fl/fl</sup> *Lyz2*-Cre mice and their littermates. The expression levels of the M1-like activation markers iNOS, CD80 and CD86 were much higher in SEPT2-deficient PMs upon VSV infection at various time points post-infection (Fig. 1g) and with different multiplicities of infection (MOI) (Fig. 1h). Additionally, SEPT2 deletion led to a more M1-like phenotype of gene transcription (Supplementary Fig. 1i, j) and a higher level of proinflammatory factor secretion after viral infection (Fig. 1i, j) than the control. These data indicated that SEPT2 plays an important role in the control of excessive inflammation.

To understand whether SEPT2 deficiency-induced excessive inflammation is exclusive to VSV infection, we infected *Sept2*<sup>fl/fl</sup> *Lyz2*-Cre mice with the double-stranded DNA virus HSV-1. The results showed that loss of SEPT2 also decreased the survival time of HSV-1-infected mice (Supplementary Fig. 2a). Next, several different DNA and RNA viruses were used to infect PMs of *Sept2*<sup>fl/fl</sup> *Lyz2*-Cre mice. We showed that deletion of SEPT2 led to excessive M1-like activation following infection with these DNA and RNA viruses (Supplementary Fig. 2b−e), and additional evidence showed that M2-like activation was almost unaffected (Supplementary Fig. 2f, g). Viral nucleic acids can be recognized by pattern recognition receptors (PRR) and contribute to the activation of the innate and subsequent adaptive immune responses[33]. In this context, we performed knockdown of well-known PRRs involved in the recognition of viral patterns, such as RIG-I, cGAS, TLR3, TLR9, MDA5, and MAVS[19,34–38](Supplementary Fig. 2h−m). The results showed that the magnitude of change in M1-like polarization between control and knockdown groups was basically the same in wild-type (WT) and SEPT2-deficient mice (Supplementary Fig. 2n−s), suggesting that SEPT2 is likely to regulate macrophage activation through a PRR-independent pathway. Considering that type I interferon (IFN) and its downstream genes are important for antiviral response, we examined the secretion of IFN-α and IFN-β in BALF as well as the transcriptional level of *Mx1* and *Isg15* in lung tissues. The results confirmed that the production of IFN-α, IFN-β, MX1, and ISG15 were not affected by SEPT2 deletion (Supplementary Fig. 3a, b). Moreover, by knocking down the type I IFN receptor subunit IFNAR1 or IFNAR2, we excluded the contribution of autocrine/paracrine type I IFNs in this process (Supplementary Fig. 3c−f). Therefore, SEPT2-regulated macrophage activation may be independent of the type I IFN pathway.

Further, we used *Escherichia coli*, *Listeria monocytogenes*, *Mycobacterium tuberculosis,* and Lipopolysaccharide (LPS) to infect/stimulate *Sept2*<sup>fl/fl</sup> *Lyz2*-Cre PMs, and found no significant difference in M1-like polarization between SEPT2-deficient and WT macrophages upon *E.coli* and *Listeria* infection or LPS stimulation. However, SEPT2 deletion in *M.tuberculosis* infection led to excessive macrophage activation (Supplementary Fig. 3g, h). These data suggested that the role of SEPT2 in inflammatory response varies depending on different bacterial infections.

Considering that IL-1β secretion is generally accompanied by pyroptotic cell death, we investigated whether cell death was involved

in M1-like hyperpolarization. The results showed no significant difference in the viability of WT and SEPT2-deficient cells at 0−24 h post-VSV infection (Supplementary Fig. 3i). Further, we found no difference in pyroptosis and apoptosis at 12 h post-infection (Supplementary Fig. 3j, k). Since the duration of in vitro infection in this study did not exceed 12 h, these data suggested that the excessive secretion of IL-1β may not be caused by pyroptosis. Taken together, these data suggested that SEPT2 deletion results in IFN-γ-independent hyperactivation of macrophages to an M1-like phenotype, thereby leading to excessive inflammation and tissue damage during infection.

## SEPT2 deficiency regulates M1-like hyperpolarization through ER stress

To further clarify the relationship between SEPT2 and IFN-γ-independent M1-like activation, we generated tamoxifen-inducible SEPT2 conditional knockout mice by using the estrogen receptor 2 (ERT2)-Cre system (Supplementary Fig. 4a). Regardless of the presence or absence of IFN-γ, SEPT2 level was gradually upregulated, and iNOS activity was mildly increased in VSV-infected control PMs (Supplementary Fig. 4b, c), suggesting controllable M1-like activation and appropriate inflammatory response against infection (Supplementary Fig. 4d). In contrast, when *Sept2*<sup>fl/fl</sup> *Lyz2*-Cre-ERT2 PMs had SEPT2 levels reduced by tamoxifen treatment, M1-like activation increased dramatically and was accompanied by an excessive inflammatory response (Supplementary Fig. 4b−d). The above results were further validated by using the iBMDM model with stable transfection of the tetracycline (Tet)-on-based SEPT2-shRNA system (Supplementary Fig. 4e−h). These data indicated that SEPT2 deficiency is coupled with IFN-γ-independent M1-like hyperpolarization and excessive inflammation.

To understand the mechanism by which SEPT2 regulates M1-like polarization, total RNA was isolated from virus-infected *Sept2*<sup>fl/fl</sup> *Lyz2*-Cre PMs and subjected to RNA sequencing. The consistency of each sample was analyzed by principal component analysis (PCA) (Supplementary Fig. 5a). Compared with the control, 2744 genes were differentially expressed ($P_{adjust}$ <0.01) by two-fold in the SEPT2-deficient PMs upon VSV infection (1207 were downregulated and 1537 were upregulated). A total of 1980 genes were differentially expressed in the HSV-1 infection group (826 were downregulated, and 1154 were upregulated). Gene Ontology (GO) pathway analysis indicated that 1060 pathways in the VSV infection group and 794 pathways in the HSV-1 infection group were enriched ($P_{adjust}$ <0.01). In addition to regular immune and inflammatory pathways, we noticed that regulation of endoplasmic reticulum pathways was enriched in both the VSV and HSV-1 groups (Supplementary Fig. 5b). The overlapping differentially expressed genes located in the enriched GO terms closely related to ER (i.e., negative regulation of response to ER stress, ER overload response) were selected for analysis, and we showed that the levels of ER stress-related genes increased significantly after viral infection in SEPT2-deficient cells (Fig. 2a). Then, transmission electron microscopy was performed to observe the morphology of the ER. We found that SEPT2-deficient PMs developed more severe ER swelling, cytoplasmic vacuolization

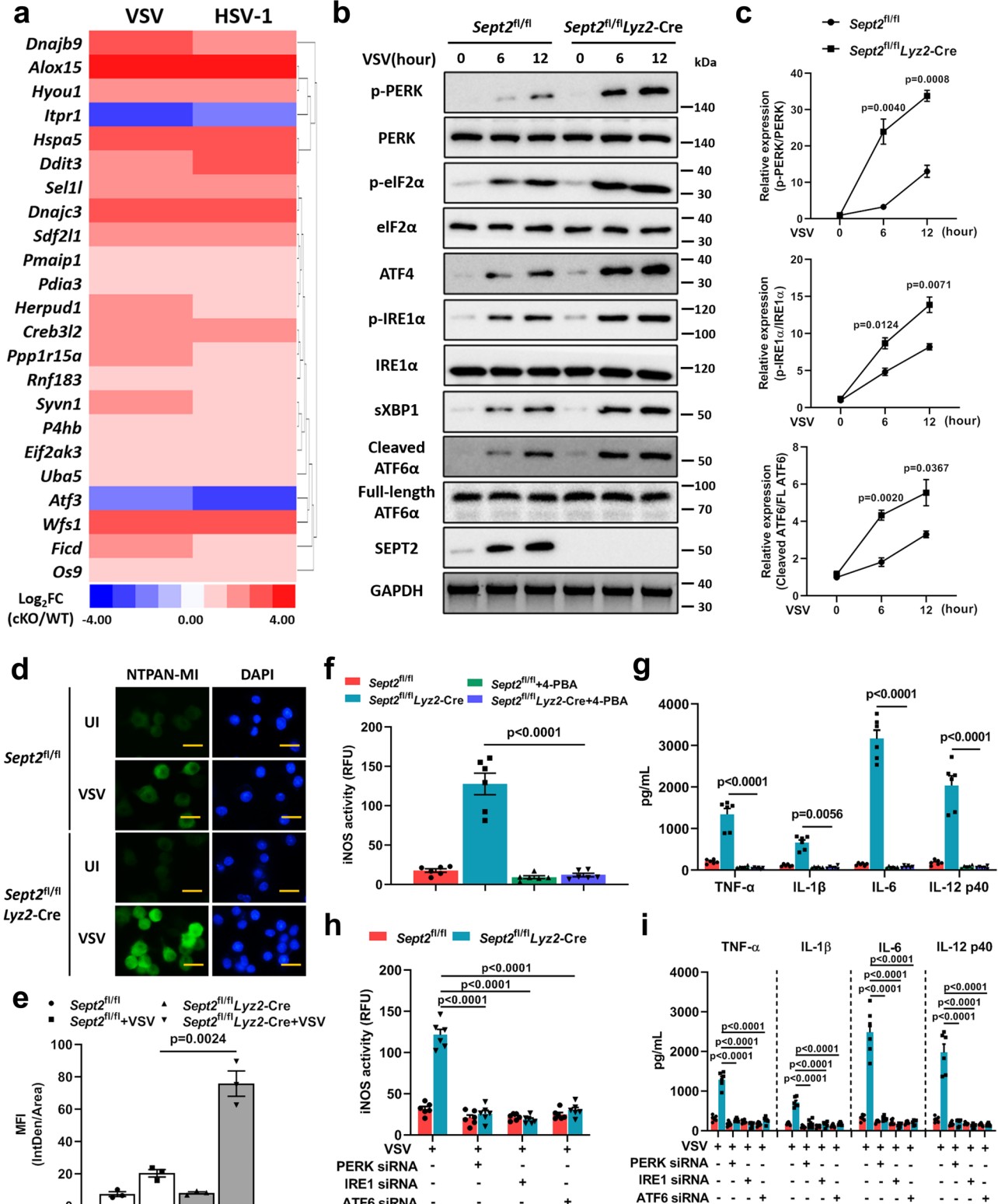

**Fig. 2 | SEPT2-deficiency regulates M1-like hyperpolarization through ER stress.**
**a** Heatmap of upregulated (red) or downregulated (blue) ER stress-related genes in
*Sept2*^fl/fl *Lyz2*-Cre mice PMs after being infected with VSV (MOI = 1) or HSV-1
(MOI = 5) for 12 h. **b, c** Immunoblot analysis of UPR-related signaling pathways in
*Sept2*^fl/fl *Lyz2*-Cre and *Sept2*^fl/fl PMs after being infected with VSV (MOI = 1) for the
indicated times (**b**). Quantitative data are graphed in (**c**). *n* = 3 in each group (**b, c**).
**d** NTPAN-MI probe was used to determine the accumulation of unfolded proteins in
*Sept2*^fl/fl *Lyz2*-Cre and *Sept2*^fl/fl PMs after being infected with VSV (MOI = 1). **e** The
mean fluorescence intensity (MFI) was quantitated and shown as IntDen/Area. Scale

bar = 20 μm. *n* = 3 in each group (**d, e**). **f, g** iNOS activity (**f**) and proinflammatory
cytokines (**g**) in *Sept2*^fl/fl and *Sept2*^fl/fl *Lyz2*-Cre PMs after being infected with VSV
(MOI = 1) in the absence or presence of 4-PBA (5 mM) for 12 h. *n* = 6 in each group
(**f, g**). **h, i** *Sept2*^fl/fl and *Sept2*^fl/fl *Lyz2*-Cre PMs were transfected with siRNAs targeting
PERK, IRE1 or ATF6 for 24 h, followed by infection of VSV (MOI = 1) for 12 h. The
iNOS activity (**h**) and proinflammatory cytokines (**i**) were detected. *n* = 6 in each
group (**h, i**). Data are shown as the mean ± s.e.m. (**c, e–i**). One-way ANOVA followed
by Bonferroni post hoc test (**c, e–i**) was used for data analysis. UI uninfected. Source
data are provided as a Source Data file.

and ribosome abscission than control PMs (Supplementary Fig. 5c). These data indicated that severe ER stress is accompanied by SEPT2 deficiency-induced M1-like hyperpolarization.

The ER senses stress mainly through three stress sensor pathways, namely, protein kinase R-like ER kinase (PERK), inositol-requiring enzyme 1 (IRE1), and activating transcription factor 6 (ATF6)[39]. To confirm our findings, we used VSV to infect PMs and showed that all three pathways (PERK, IRE1α, ATF6), as well as their downstream eukaryotic initiation factor-2α (eIF2α), activating transcription factor 4 (ATF4) (Fig. 2b, c), IRE1-mediated X-box binding protein 1 (XBP1) mRNA splicing (Fig. 2b, Supplementary Fig. 5d) and cleaved ATF6 (Fig. 2b, c), were more highly activated in SEPT2-deficient PMs than in control PMs. We also tested the activation of signaling pathways (IRE1-TRAF6-IKK, JNK-AP1 and NF-κB) that have been reported to regulate macrophage proinflammatory activation[40,41]. The results showed that SEPT2-deficient PMs displayed earlier and higher activation of the indicated pathways after viral infection (Supplementary Fig. 5e). The overactivation of these pathways may arise from the excessive ER stress caused by SEPT2 deletion, and the downstream product TNF-α can also trigger these proinflammatory pathways, further accelerating their activation. Considering that stress sensors induce unfolded protein responses (UPR) after the recognition of unfolded/misfolded proteins, we used the NTPAN-MI probe[42] (Supplementary Fig. 5f) to indicate the unfolded proteins, and the results showed that SEPT2 deletion resulted in a more substantial enrichment of unfolded proteins upon viral infection than the control (Fig. 2d, e). We then used 4-phenylbutyric acid (4-PBA), a chemical chaperone that eliminates the unfolded proteins in the ER, to confirm this result. We showed that inhibition of unfolded protein-derived ER stress by 4-PBA alleviated excessive M1-like activation (Fig. 2f, Supplementary Fig. 5g) and its associated inflammatory cytokines (Fig. 2g) in SEPT2-deficient cells. Furthermore, we used specific siRNAs to induce PERK, IRE1 and ATF6 knockdown (Supplementary Fig. 5h), and found that suppression of one pathway could relieve the excessive M1-like activation (Fig. 2h) and inflammatory responses (Fig. 2i) in Sept2[fl/fl] Lyz2-Cre PMs. In contrast, knockdown of PERK, IRE1 or ATF6 had little effect in Sept2[fl/fl] cells (Fig. 2h, i). Consistently, knocking down any pathway inhibited p-eIF2α level in SEPT2-deficient cells rather than in WT cells (Supplementary Fig. 5i). These results suggested that SEPT2 might exert a potential influence on the redundancy among the UPR pathways.

Additionally, we examined whether other triggers of ER stress are associated with SEPT2-regulated M1-like hyperpolarization. Mag-Fluo4 probe staining showed that there was no significant difference in intracellular free calcium (Ca$^{2+}$) after VSV infection between WT and SEPT2-deficient PMs (Supplementary Fig. 6a). Oxidative stress is another trigger of ER stress and we found that the oxidative stress level was increased in SEPT2-deficient cells (Supplementary Fig. 6b). Considering that SEPT2 deletion resulted in massive iNOS/NO release (Fig. 1f–h), we used 1400 W dihydrochloride to inhibit iNOS activity (Supplementary Fig. 6c). The results showed no significant difference in oxidative stress between WT and SEPT2-deficient PMs under iNOS-blocking condition (Supplementary Fig. 6d). Moreover, iNOS blockade could not attenuate the excessive UPR and proinflammatory factor secretion in SEPT2-deficient cells (Supplementary Fig. 6e, f). Therefore, we hypothesize that oxidative stress is not the primary cause of SEPT2-regulated high level of ER stress. Disorders of lipid metabolism are also one of the factors that cause ER stress. Through lipid droplet staining, we found that the lipid content was not significantly changed (Supplementary Fig. 6g), and qRT-PCR results validated that the key genes regulating lipid metabolism in the ER (e.g., Fas, Acc and Srebp-1c) were not significantly affected by SEPT2 deletion (Supplementary Fig. 6h). Taken together, these results demonstrated that SEPT2 deficiency-induced M1-like

hyperpolarization is probably mediated by abnormal accumulation of unfolded proteins and is not directly associated with dysregulation of calcium flux, oxidative stress or lipid metabolism.

## SEPT2 limits the proteasomal degradation of HSPA5

HSPA5, also known as Bip or GRP78, is a resident chaperone in the ER that promotes the folding and assembly of proteins. Under the stimulation of ER stress, HSPA5 is often upregulated, acting as a central sensor and adapting to changes in the ER. When the unfolded proteins begin to accumulate, HSPA5 dissociates from the stress sensors PERK, IRE1, and ATF6 and binds nascent peptides to facilitate proper folding[43]. Other proteins involved in homeostasis and apoptosis, such as CCAAT/enhancer-binding protein homologous protein (CHOP) and ATF4, are preferentially translated[44]. Interestingly, we observed that CHOP and ATF4 protein presented an earlier and higher level of upregulation, while HSPA5 expression remained abnormally low in VSV-infected Sept2[fl/fl] Lyz2-Cre PMs (Fig. 3a, b).

We determined whether HSPA5 expression is regulated by SEPT2. Using the Tet-on system based on the SEPT2-shRNA stably transfected iBMDM cell line (Supplementary Fig. 4e), we found that HSPA5 protein levels were increased in the first 6 h after VSV infection. However, with the reduction in SEPT2 after the addition of doxycycline (DOX) to the culture medium, the protein level of HSPA5 decreased (Fig. 3c). In contrast, the mRNA level of HSPA5 measured by qRT-PCR showed a continuously increasing trend (Fig. 3d). To further verify this finding, we performed SEPT2 knockdown by using siRNAs in Sept2[fl/fl] PMs (Supplementary Fig. 7a). After VSV infection, knockdown of SEPT2 did not affect the mRNA level (Supplementary Fig. 7b) but decreased the protein level of HSPA5 (Supplementary Fig. 7c). Furthermore, we showed that HSPA5 protein levels, but not mRNA levels, were elevated after SEPT2 replenishment in SEPT2-deficient cells (Supplementary Fig. 7d, e). These data suggested that SEPT2 negatively regulates the HSPA5 protein in an indirect manner.

Next, we used cycloheximide (CHX, a protein synthesis inhibitor) to investigate how SEPT2 regulates the protein stability of HSPA5 and found that deletion of SEPT2 resulted in the protein degradation of HSPA5 (Fig. 3e, Supplementary Fig. 7f). Therefore, specific inhibitors targeting the lysosomal (E64d + PepA) or proteasomal (MG132) pathway were used to identify the degradation pathway of HSPA5. We showed that HSPA5 was no longer decreased when the proteasomal degradation pathway was blocked by MG132 (Supplementary Fig. 7g). Then, we evaluated the ubiquitination level of HSPA5 in Sept2[fl/fl] Lyz2-Cre PMs infected with VSV or HSV-1 and found that SEPT2 deletion resulted in enhanced ubiquitination of HSPA5 (Fig. 3f, g, Supplementary Fig. 7h). These data indicated that SEPT2 deficiency promotes the proteasomal degradation of HSPA5.

After ruling out the possibility that SEPT2 acts directly as an ubiquitin E3 ligase of HSPA5 via an in vitro ubiquitination system (Supplementary Fig. 7i), we performed HCS by using a Ubiquitination Compound Library to screen ubiquitin E3 ligases that mediate HSPA5 degradation (Supplementary Data 2). We identified seven potential candidates, and then we validated these candidates. The results showed that SEPT2 deficiency-induced degradation of HSPA5 was rescued when sodium channel epithelial 1 subunit beta (SCNN1B) was knocked down (Supplementary Fig. 7j). This result implied that SCNN1B is a potential E3 ligase for the proteasomal degradation of HSPA5. By using the SEPT2-shRNA stably transfected iBMDM model, we found that the degradation of HSPA5 was inhibited by SCNN1B knockdown (Fig. 3h, Supplementary Fig. 7k). Additionally, in vitro ubiquitination assays proved that SCNN1B mediated the ubiquitination of HSPA5 directly (Fig. 3i). To further define the linkage type of ubiquitination (Supplementary Fig. 7l), we used a series of ubiquitin mutants (K6O, K11O, K27O, K29O, K33O, K48O, and K63O), each of which contained only one lysine available for polylinkage[45]. The results showed that only K48-linked ubiquitin chains contributed to SEPT2

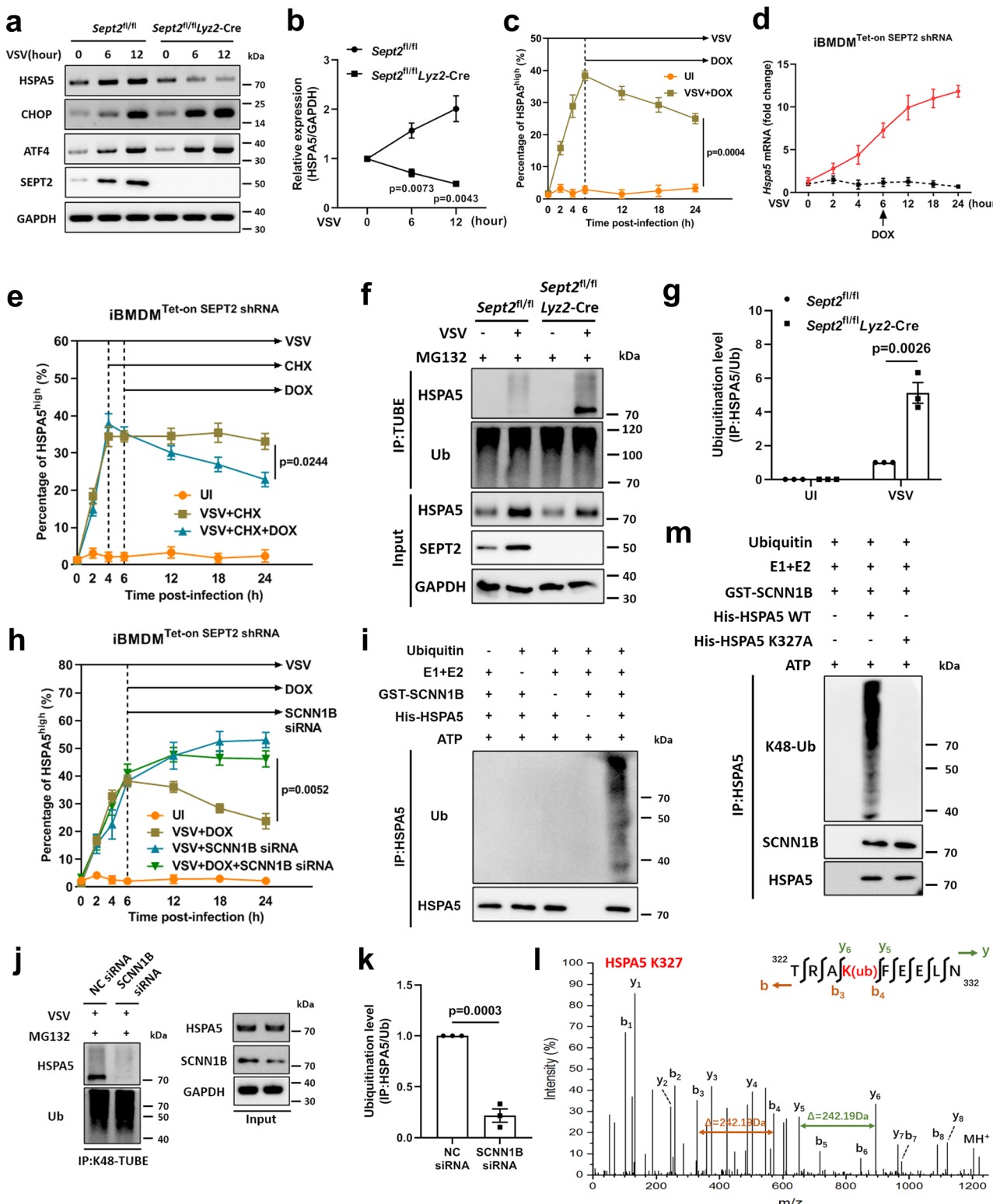

deficiency-mediated HSPA5 ubiquitination (Supplementary Fig. 7m). When SCNN1B was knocked down by siRNA, K48-linked ubiquitination of HSPA5 decreased (Fig. 3j, k). Furthermore, high-performance liquid chromatography-tandem mass spectrometry (HPLC-MS/MS) revealed that the Lysine 327 (K327) site of HSPA5 was ubiquitinated in SEPT2-deficient PMs after VSV infection (Fig. 3l). In vitro ubiquitination assays also proved that SCNN1B failed to modify the HSPA5 protein bearing a K327A mutation, although the interaction between them still existed (Fig. 3m). Due to the inhibition of ubiquitination, the stability of

K327A mutant was higher than that of the WT HSPA5 (Supplementary Fig. 7f, n). These results collectively showed that SEPT2 deficiency promotes the proteasomal degradation of HSPA5 in virus-infected macrophages.

**SEPT2 promotes the acetylation of HSPA5 by recruiting ATAT1**
Next, we investigated the role of SEPT2 in the SCNN1B-mediated proteasomal degradation of HSPA5. We found that SEPT2 deficiency enhanced the binding of SCNN1B to HSPA5 (Supplementary Fig. 8a, b)

**Fig. 3 | SEPT2 deficiency promotes the ubiquitination of HSPA5. a**, **b**. Immunoblot analysis of UPR-related genes in PMs after being infected with VSV (MOI = 1) (**a**). Quantitative data are graphed in (**b**). $n = 3$ in each group (**a**, **b**). **c**, **d** Flow cytometry (**c**) and qRT-PCR (**d**) analysis of HSPA5 in iBMDMs after being infected with VSV (MOI = 1) and treated with DOX (1 μg/mL). The quantitative flow cytometry data are graphed in (**c**). $n = 3$ in each group (**c**). The dotted line in (**d**) indicated the uninfected control. $n = 6$ in each group (**d**). **e** Quantitative flow cytometry data of HSPA5 in iBMDMs after being infected with VSV (MOI = 1) and treated with CHX (50 μg/mL) and DOX (1 μg/mL). $n = 3$ in each group (**e**). **f**, **g** TUBE analysis of HSPA5 in PMs after being infected with VSV (MOI = 1) for 12 h (**f**). Quantitative data are graphed in (**g**). **h** Quantitative flow cytometry data of HSPA5 in iBMDMs after being infected with VSV (MOI = 1), treated with DOX (1 μg/mL) and transfected with SCNN1B siRNA. **i** The ubiquitination of HSPA5 in an in vitro ubiquitination system was analyzed by immunoblots. **j**, **k** PMs were transfected with SCNN1B siRNA, followed by VSV infection (MOI = 1) for 12 h. The ubiquitination of HSPA5 was detected by K48-TUBE analysis (**j**). Quantitative data are graphed in (**k**). $n = 3$ in each group (**f**–**k**). **l** HPLC-MS/MS analysis of HSPA5 ubiquitination sites in *Sept2*^fl/fl *Lyz2*-Cre PMs after being infected with VSV (MOI = 1) for 12 h. **m** The ubiquitination of HSPA5 and the HSPA5-SCNN1B interaction in an in vitro ubiquitination system were analyzed by immunoblots. $n = 3$ in each group (**m**). MG132 (10 μg/mL) was used to inhibit the proteasomal degradation (**f**, **g**, **j**–**l**). Data are shown as the mean ± s.e.m. (**b**–**e**, **g**, **h**, **k**). One-way ANOVA followed by Bonferroni post hoc test (**b**, **c**, **e**, **g**, **h**, **k**) was used for data analysis. UI uninfected. NC negative control. Source data are provided as a Source Data file.

rather than affecting the mRNA or protein levels of SCNN1B (Supplementary Fig. 8c, d). Considering that SEPT2 hampered the formation of HSPA5 and the SCNN1B complex, we then examined whether SEPT2 affected the general function of SCNN1B, and the results showed that ubiquitination levels of the other two substrates of SCNN1B, namely, WDTC1 and GRK2[46,47], were not affected by SEPT2 deletion (Supplementary Fig. 8e, f). Furthermore, we showed that SEPT2 alone did not affect the binding of SCNN1B to HSPA5 by using an in vitro ubiquitination system (Supplementary Fig. 8g). These data indicated that SEPT2 may hamper the binding of SCNN1B to HSPA5 due to space-occupying effects such as posttranslational modifications or recruitment of other factors. To test this hypothesis, we used pan phosphoserine/threonine, pan-dimethyl-lysine, and pan-acetyl-lysine antibodies to detect the modification status of HSPA5 after VSV infection. The results showed that the acetylation level of HSPA5 was differentially changed after VSV infection between SEPT2-deficient and control cells (Supplementary Fig. 8h). We then examined the acetylation level of HSPA5 at 6 h and 12 h post-infection and found that HSPA5 could not be further acetylated in SEPT2-deficient PMs (Fig. 4a, b). The HSPA5 acetylation data collected from HSV-1 infection were similar to those from VSV infection (Supplementary Fig. 8i, j). Furthermore, we showed that upon treatment with a deacetylase inhibitor cocktail (DIC), a mixture of deacetylase inhibitors that can widely increase the acetylation level of cellular proteins, the M1-like hyperpolarization and inflammatory response induced by VSV infection were significantly relieved in SEPT2-knockdown iBMDMs (Fig. 4c, d). These results implied that the functional role of SEPT2 is connected with the acetylation of HSPA5.

By using an in vitro acetylation system, we showed that SEPT2 was not an acetyltransferase that acetylates HSPA5 directly (Supplementary Fig. 8k), which confirmed our hypothesis that SEPT2 prevents the binding of SCNN1B to HSPA5 through protein modification via recruitment of other factors. To identify the specific acetylase recruited by SEPT2, we performed an siRNA screen based on the acetyltransferase library constructed according to a previous report[48] and found that the acetylation level of HSPA5 was significantly decreased when alpha-tubulin acetyltransferase 1 (ATAT1) was knocked down (Supplementary Fig. 8l). Furthermore, we found that ATAT1 could bind to HSPA5 in the presence of SEPT2 (Fig. 4e). In vitro acetylation experiments also demonstrated that ATAT1 acetylated HSPA5 directly (Fig. 4f).

To understand the role of ATAT1 in HSPA5 acetylation, we constructed an ATAT1-knockout cell line based on iBMDMs (Supplementary Fig. 8m) and found that the acetylation level of HSPA5 was decreased by VSV infection in ATAT1-deficient cells (Fig. 4g, h), which was similar to that in SEPT2-deficient cells. As expected, M1-like hyperpolarization and excessive inflammatory cytokines were accompanied by a decrease in HSPA5 acetylation (Fig. 4i, j). This finding was validated by data collected from HSV-1 infection (Supplementary Fig. 8n–q). Furthermore, supplementation with exogenous protein was performed to investigate the necessity of ATAT1 in the UPR alleviation of *Sept2*^fl/fl *Lyz2*-Cre PMs (Fig. 4k). We found that

the ATAT1 and SEPT2 combination effectively alleviated excessive unfolded proteins (Fig. 4l, m), M1-like hyperpolarization (Fig. 4n) and inflammatory cytokines (Fig. 4o) in SEPT2-deficient PMs, but supplementation with ATAT1 alone did not. These data confirmed that SEPT2 is required for ATAT1 to acetylate HSPA5. Considering the proinflammatory effect of acetylated p65 in the NF-κB pathway, we examined whether ATAT1 can acetylate p65. The results showed that overexpression of ATAT1 did not promote the acetylation of p65 (Supplementary Fig. 8r). It has been reported that oligomerization is important for the recruiting function of the SEPTIN family. We discovered that SEPT2 was oligomerized during the recruitment of ATAT1 using native PAGE (Supplementary Fig. 9a). This finding was further validated by coimmunoprecipitation in WT iBMDMs cotransfected with HA- and Myc-tagged SEPT2 (Supplementary Fig. 9b, c). In addition to homo-oligomerization, SEPT2 has also been reported to form hetero-oligomeric complex with SEPT6, SEPT7 and SEPT9[49,50]. We then examined whether these SEPTINs are participated in the assembly of the SEPT2-HSPA5 complex. The results showed that SEPT2 could interact with SEPT6, SEPT7, SEPT9 (Supplementary Fig. 9d); however these data were negative when performing coimmunoprecipitations against HSPA5 (Supplementary Fig. 9e). To further investigate whether SEPT2 is working alone or in a complex with other SEPTINs, we performed SEPT6, SEPT7 and SEPT9 knockdown (Supplementary Fig. 9f). Interestingly, we showed that knockdown of SEPT7, instead of SEPT6 and SEPT9, promoted M1-like hyperpolarization after viral infection (Supplementary Fig. 9g). Further, we performed immunofluorescence experiments to test the localization of SEPT2 and SEPT7. The results showed that SEPT2 and SEPT7 did colocalize; however, the co-localization did not change upon VSV infection (Supplementary Fig. 9h). These data suggested that SEPT7 may not directly participate in the SEPT2-HSPA5 complex. According to a previous report, depletion of SEPT7 induces partial codepletion of SEPT2[51]. We then examined whether knockdown of SEPT6, SEPT7, and SEPT9 would affect the expression of SEPT2. The results showed that the expression of SEPT2 was downregulated with SEPT7 knockdown (Supplementary Fig. 9i). Therefore, we speculate that knockdown of SEPT7 may promote M1-like hyperpolarization by indirectly reducing SEPT2. However, this hypothesis needs to be verified by further experiments.

Also, we tested the expression levels of *Septin* 2, 5, 6, 7, 9, 10, 11. The results showed that VSV infection upregulated the expression of SEPT2, SEPT6, and SEPT9 in macrophages (Supplementary Fig. 9j, k). Considering that SEPTINs may have an effect on ER stress by regulating mitochondrial behavior, we performed SEPT2, SEPT6 or SEPT9 knockdown in iBMDMs, and detected their interference in mitochondrial fission. The results showed that knockdown of SEPT2 significantly decreased the mitochondrial fission rate (Supplementary Fig. 9l), which is consistent with previous data that SEPT2 depletion induces mitochondrial elongation[51]. Further, we found that the magnitudes of change in mitochondrial fission rate by SEPT2 siRNA treatment were the same between control and VSV infection groups (Supplementary Fig. 9l). This suggests that the impact on mitochondrial fission by

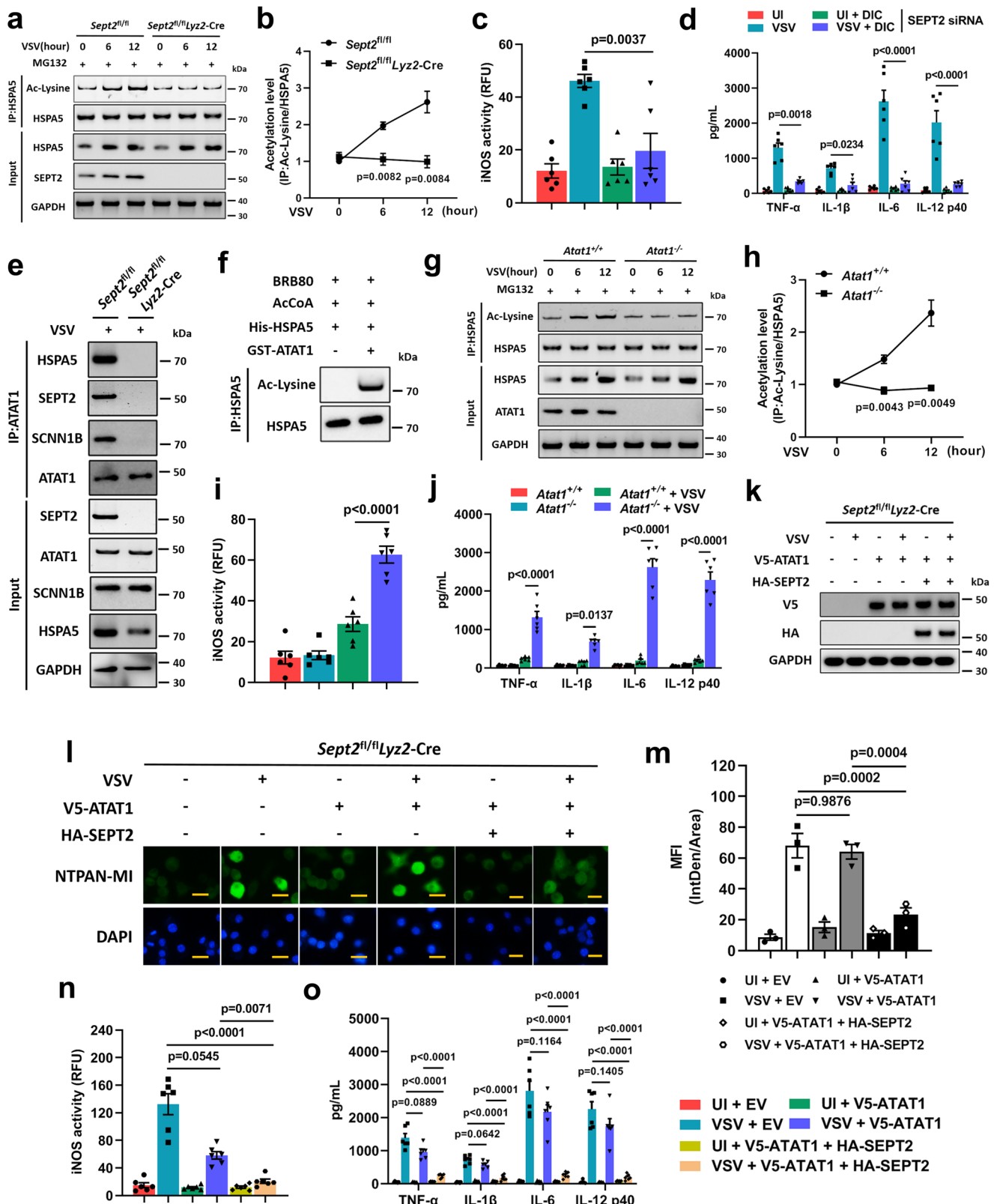

SEPT2 knockdown may not closely linked to the M1-like hyperpolarization process.

In order to further visualize the precise role of SEPT2, we examined the interaction between SEPT2 and HSPA5, as well as the co-localization of SEPT2 and ER (using SEC61B as an ER marker) by immunofluorescence. The results showed that VSV infection could enhance the co-localization of SEPT2 and HSPA5 as expected; however,

the co-localization of ER and SEPT2 was not significantly changed upon infection (Supplementary Fig. 9m). Taken together, our data showed that SEPT2 promotes the acetylation of HSPA5 by recruiting the acetylase ATAT1 and that acetylated HSPA5 is critical for controllable M1-like activation and the inflammatory response during infection. Deletion of either SEPT2 or ATAT1 results in loss of HSPA5 function, which eventually leads to excessive inflammation.

**Fig. 4 | SEPT2 recruits ATAT1 to acetylate HSPA5. a**, **b**. Immunoblot analysis of acetylated HSPA5 in PMs after being infected with VSV (MOI = 1) for the indicated times (**a**). Quantitative data are graphed in (**b**). *n* = 3 in each group (**a**, **b**). **c**, **d** iBMDMs were transfected with SEPT2 siRNA, followed by treatment of DIC (1×) and infection of VSV (MOI = 1) for 12 h. Afterward, iNOS activities (**c**) and secretion of proinflammatory cytokines (TNF-α, IL-1β, IL-6, and IL-12 p40) (**d**) were detected. *n* = 6 in each group (**c**, **d**). **e** Immunoprecipitation analysis of the interaction between ATAT1, SCNN1B, HSPA5, and SEPT2 in PMs after being infected with VSV (MOI = 1) for 12 h. **f** The acetylation of HSPA5 in an in vitro acetylation system was analyzed by immunoblots. **g**, **h** *Atat1*$^{+/+}$ and *Atat1*$^{-/-}$ iBMDMs were infected with VSV (MOI = 1) for the indicated times, followed by immunoprecipitation analysis of acetylated HSPA5 using an Ac-Lysine antibody (**g**). Quantitative data are graphed in (**h**). *n* = 3 in each group (**e**–**h**). **i**, **j** iNOS activities (**i**) and secretion of proinflammatory cytokines (**j**) in iBMDMs after being infected with VSV (MOI = 1) for 12 h. *n* = 6 in each group (**i**, **j**). **k**–**o** *Sept2*$^{fl/fl}$ *Lyz2*-Cre PMs were transfected with V5-tagged ATAT1 or HA-tagged SEPT2, followed by VSV infection (MOI = 1) for 12 h. **k** The expression of V5-ATAT1 and HA-SEPT2 was detected by western blotting. **l** NTPAN-MI probe was used to determine the accumulation of unfolded proteins. Scale bar = 20 μm. **m** The MFI was quantitated and shown as IntDen/Area. **n**, **o** iNOS activities (**n**) and secretion of proinflammatory cytokines (**o**) were detected. *n* = 3 (**k**–**m**) or *n* = 6 (**n**, **o**) in each group. MG132 (10 μg/mL) was used to inhibit the proteasomal degradation of HSPA5 (**a**, **b**, **g**, **h**). Data are shown as the mean ± s.e.m. (**b**–**d**, **h**–**j**, **m**–**o**). One-way ANOVA followed by Bonferroni post hoc test (**b**–**d**, **h**–**j**, **m**–**o**) was used for data analysis. UI uninfected. EV empty vector. Source data are provided as a Source Data file.

## ATAT1 has a stronger affinity for HSPA5 than SCNN1B

SEPT2 promoted the acetylation of HSPA5 by recruiting the acetylase ATAT1 and reduced the ubiquitination of HSPA5 by blocking the binding of the E3 ligase SCNN1B. We examined whether there was a connection between these two separate events. HPLC-MS/MS showed that the K327 site of HSPA5 was acetylated in control PMs but not in SEPT2-deficient PMs after VSV infection (Fig. 5a). The acetylation modification of HSPA5 bearing a K327A mutation was abolished, which confirmed that HSPA5 acetylation occurs at the K327 site (Fig. 5b). As the K48-linked ubiquitination of HSPA5 also occurs at the K327 site (Fig. 3l, m), we speculate that there may be a competitive relationship between the ubiquitination and acetylation of HSPA5.

To investigate this hypothesis, Myc-tagged HSPA5 was overexpressed in control or SEPT2-deficient PMs, and immunoprecipitation was performed to examine the modification status of HSPA5. We found that HSPA5 was prone to acetylation in control cells, whereas ubiquitinated HSPA5 was predominant in SEPT2-deficient PMs (Fig. 5c, d). These data were also validated in iBMDMs in the presence or absence of IFN-γ (Supplementary Fig. 10a–c). In order to investigate whether the role of SEPT2 is specific to macrophages, we performed SEPT2 knockdown in NIH-3T3, L929 and TC-1 cell lines. We found that SEPT2 knockdown resulted in an increase of HSPA5 ubiquitination in all cell types, but was not necessarily accompanied with a decrease in acetylation levels (Supplementary Fig. 10d).

Furthermore, an increase in ubiquitination, along with a decrease in acetylation of HSPA5, was observed when the acetylase ATAT1 was knocked down in *Sept2*$^{fl/fl}$ cells (Supplementary Fig. 10e, f). In contrast, a decrease in ubiquitination of HSPA5 was observed when the E3 ligase SCNN1B was knocked down in SEPT2-deficient cells (Supplementary Fig. 10g, h). Due to the absence of SEPT2, the acetylation level of HSPA5 did not increase with the knockdown of SCNN1B (Supplementary Fig. 10g, h). Additionally, proximity ligation assays (PLA) showed that the acetylation level continued to increase, and the ubiquitination level of HSPA5 remained relatively low in control cells (Fig. 5e). In contrast, the ubiquitination level continued to increase, and the acetylation level remained low in SEPT2-deficient cells during VSV infection (Fig. 5e). When the K327 site of HSPA5 was mutated, both acetylation and ubiquitination were suppressed (Supplementary Fig. 10i). These results were consistent with our former finding that the acetylation and ubiquitination of HSPA5 competitively share the K327 site, and ATAT1 has priority in binding HSPA5 compared to SCNN1B in macrophages.

To reveal the affinity of the two enzymes for HSPA5, we performed an interaction analysis based on molecular structure docking. Supplementary Fig. 10j–l shows the protein structures of HSPA5, ATAT1, and SCNN1B with the inactive structures removed. Then, rigid and flexible docking of ATAT1/HSPA5 and SCNN1B/HSPA5 and two rounds of optimization with Rosetta on the preliminary conformations of the two global dockings were performed (Supplementary Fig. 10m, n). By systematically analyzing the binding interface of the two complexes,

we determined that ATAT1 has a stronger affinity for HSPA5 than SCNN1B (Fig. 5f). In particular, at the K327 residue where ATAT1 modified HSPA5, there were two π-cation interactions that further enhanced the interaction (Fig. 5f). Furthermore, isothermal titration calorimetry (ITC) and surface plasmon resonance (SPR) were performed to determine the accurate dissociation rate constants (Kd) of the ATAT1/HSPA5 and SCNN1B/HSPA5 complexes. Similar results were obtained from ITC (Fig. 5g, h) and SPR (Fig. 5i, j), which showed that ATAT1 had approximately 6.91–8.73-fold greater affinity for HSPA5 than SCNN1B for HSPA5.

Additionally, we used fluorescent resonance energy transfer (FRET) to demonstrate that ATAT1 had a competitive advantage for HSPA5 in *Atat1*$^{-/-}$ *Scnn1b*$^{-/-}$ iBMDMs. ANAP-modified HSPA5 and YFP-tagged ATAT1 were constructed for the intracellular FRET system (Supplementary Fig. 10o, p). The FRET ratio showed that ATAT1 binding to HSPA5 was only slightly attenuated under competition with a high dose of SCNN1B (Fig. 5k). Cell lysates were collected to further analyze the modification status of HSPA5. Consistently, HSPA5 was still preferentially acetylated despite the increase in the SCNN1B dose (Fig. 5l, m). However, we showed that a small amount of ATAT1 was able to hijack HSPA5 (Fig. 5n), accompanied by an increase in HSPA5 acetylation levels and a decrease in ubiquitination levels when different doses of ATAT1 were used for binding competition (Fig. 5o, p). Collectively, these results indicated that ATAT1 and SCNN1B have a competitive relationship for the modification of HSPA5 at the K327 site and that ATAT1 has a stronger affinity for HSPA5.

## K327-acetylated HSPA5 prevents M1-like activation and excessive inflammation

Since acetylation of the K327 site is critical for HSPA5 function, we wondered whether K327-acetylated HSPA5 regains the ability to alleviate the UPR in a mouse model. By using an in vivo transfection method, we supplemented *Sept2*$^{fl/fl}$ *Lyz2*-Cre mice with the WT or K327Q form (acetylation mimic) of HSPA5 (Fig. 6a, b). The K327Q mutant was more stable than the WT (Supplementary Fig. 7f, Fig. 6c). Compared to the HSPA5 WT group, the survival time of *Sept2*$^{fl/fl}$ *Lyz2*-Cre mice infected with VSV was significantly lengthened by exogenous HSPA5 K327Q supplementation (Fig. 6d). Cytokine levels in the BALF (Fig. 6e), as well as HE staining of lung tissues (Fig. 6f, g) at 7 dpi, proved that K327Q, a K327 acetylation mimic of HSPA5, can rescue the excessive inflammation and tissue damage in SEPT2-deficient mice. In addition, PMs were isolated for further examination. In line with our expectations, HSPA5 K327Q remained stable without being degraded by ubiquitination (Fig. 6h-j), thereby reducing unfolded proteins (Fig. 6k, l) and leading to controllable M1-like activation in SEPT2-deficient mice (Fig. 6m, n).

To further validate the role of K327 acetylated HSPA5 in excessive inflammation, we constructed transgenic mice bearing the HSPA5 K327Q mutation using a site-directed mutagenesis approach (Supplementary Fig. 11a). By in vivo transfection of SEPT2 siRNA (Supplementary Fig. 11b), we found that HSPA5$^{K327Q}$ mice survived longer than

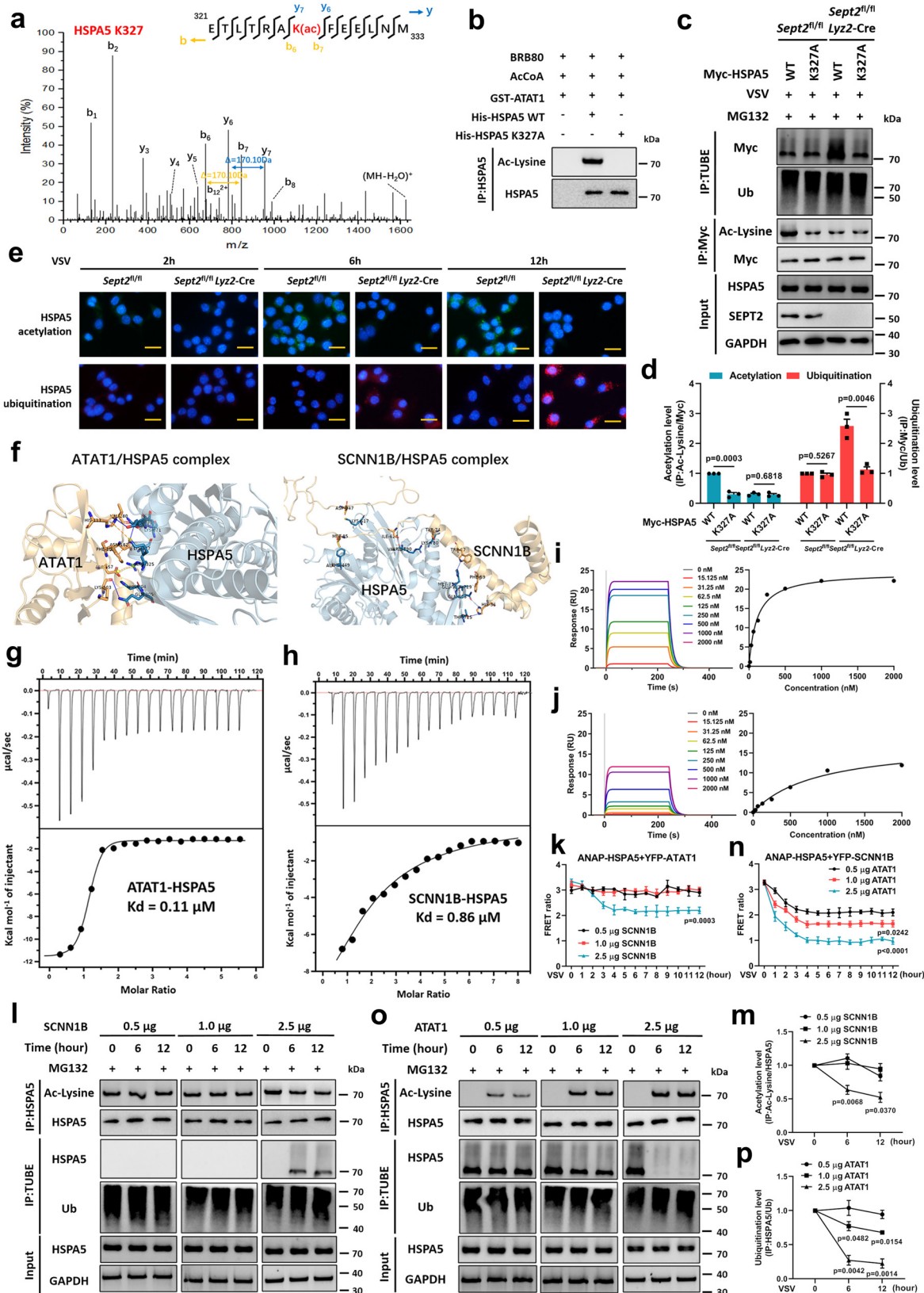

HSPA5[WT] mice under SEPT2 knockdown conditions (Supplementary Fig. 11c). Cytokine secretion (Supplementary Fig. 11d) and lung lesion results (Supplementary Fig. 11e, f) indicated that less inflammation occurred in the HSPA5[K327Q] mice transfected with SEPT2 siRNA. Correspondingly, PMs obtained from HSPA5[K327Q] mice exhibited lower HSPA5 ubiquitination levels (Supplementary Fig. 11g–i), attenuated

unfolded proteins (Supplementary Fig. 11j, k), and milder M1-like activation (Supplementary Fig. 11l, m) than those collected from HSPA5[WT] mice after VSV infection. Taken together, these data showed that K327-acetylated HSPA5 is beneficial for avoiding SEPT2-deficiency-induced excessive M1-like hyperactivation and excessive inflammation upon infection.

**Fig. 5 | ATAT1 has a stronger affinity for HSPA5 than SCNN1B. a** HPLC-MS/MS analysis of HSPA5 acetylation sites in *Sept2*[fl/fl] PMs after being infected with VSV (MOI = 1) for 12 h. **b** The acetylation of HSPA5 in an in vitro acetylation system was analyzed by immunoblots. **c**, **d** Analysis of acetylated and ubiquitinated HSPA5 in PMs after being transfected with Myc-tagged HSPA5 WT or HSPA5 K327A and infected with VSV (MOI = 1) for 12 h (**c**). Quantitative data are graphed in (**d**). **e** PLA of the acetylation and ubiquitination status of HSPA5 in PMs after being infected with VSV. Scale bar = 20 μm. *n* = 3 in each group (**b**–**e**). **f** Overall structure of ATAT1-HSPA5 and SCNN1B-HSPA5 complexes. ATAT1 (orange in left panel), HSPA5 (blue) and SCNN1B (orange in right panel) were shown in cartoon form (Hydrophobic Interactions: gray dotted line; Hydrogen Bonds: blue line; π-Cation Interactions: orange dotted line; Salt Bridges: yellow dotted line). **g**, **h** ITC-binding curve between ATAT1-HSPA5 (**g**) and SCNN1B-HSPA5 (**h**). The ITC experiments were repeated three times independently. **i**, **j** Representative SPR sensorgram of the response (Response Unit, RU) versus time when ATAT1 protein (**i**) or SCNN1B protein (**j**) were injected over HSPA5 protein. The SPR experiments were repeated three times independently. **k**–**p** *Atat1*[−/−] *Scnn1b*[−/−] iBMDMs were transfected with CMV-HSPA5(TAG)-ATAT1-YFP (**k**–**m**) or CMV-HSPA5(TAG)-SCNN1B-YFP (**n**–**p**) constructs, and then treated with 50 μM ANAP. After 18 h, cells were transfected with SCNN1B (**k**–**m**) or ATAT1 (**n**–**p**) plasmids (0.5 μg, 1.0 μg and 2.5 μg, respectively). Graph displays the FRET ratio ($I_{YFP}/I_{ANAP}$) recorded from single cell images (**k**, **n**). The acetylation and ubiquitination levels of HSPA5 were detected by immunoprecipitation and TUBE analysis, respectively (**l**, **o**). Quantitative data are graphed in (**m**, **p**). *n* = 3 in each group (**k**–**p**). MG132 (10 μg/mL) was used to inhibit the proteasomal degradation of HSPA5 (**c**–**e**, **k**–**p**). Data are shown as the mean ± s.e.m. (**d**, **k**, **m**, **n**, **p**). One-way ANOVA followed by Bonferroni post hoc test (**d**, **k**, **m**, **n**, **p**) was used for data analysis. Source data are provided as a Source Data file.

## SEPT2 is a promising target for viral infection-induced cytokine storms

SEPT2-deficient macrophages exhibit M1-like hyperactivation and excessive inflammation, while WT macrophages develop moderate inflammation upon infection. We examined whether SEPT2 acts as a brake in excessive inflammation. PR8M and PR8F, two mouse-adapted variants of the influenza virus strain A/Puerto Rico/8/34, have been reported to induce different inflammatory response strength levels in C57BL/6J mice[52]. Upon IFN-γ signaling blockade, we showed that mice infected with PR8F developed significant lung damage (Supplementary Fig. 12a), a higher percentage of M1-like activation (Supplementary Fig. 12b, c), and a more severe inflammatory response (Supplementary Fig. 12d) than mice infected with PR8M. Furthermore, we found that SEPT2 expression was significantly upregulated in PR8M-infected mice, while SEPT2 expression remained relatively low in PR8F-infected mice (Supplementary Fig. 12e). These data implied that SEPT2 expression is inducible, and we speculate that highly pathogenic viruses such as PR8F may reduce the expression of SEPT2, thus inducing a continuous accumulation of M1-like activation and inflammation. Then, we attempted to attenuate the excessive inflammation in PR8F-infected mice by in vivo transfection of SEPT2. The survival of PR8F-infected mice was significantly prolonged by SEPT2 overexpression (SEPT2-OE) (Fig. 7a). Further analyses showed that mice with SEPT2-OE had reduced lung damage (Fig. 7b, c), macrophage M1-like activation (Fig. 7d), and inflammatory responses compared with the control (Fig. 7e). These data collectively indicated that SEPT2 alleviates the excessive inflammatory response caused by viral infection.

To understand whether SEPT2 is critical for the different outcomes after infection, we detected the SEPT2 level in iBMDMs following infection with different viruses. The results showed that SEPT2 expression was significantly elevated by infection with VSV and PR8M but not PR8F (Fig. 7f, g). These data confirmed that SEPT2 expression in macrophages is inducible. Considering that unfolded protein-derived ER stress plays an important role in SEPT2-mediated cytokine release, we showed that SEPT2 expression could be downregulated by treating macrophages with 4-PBA (Supplementary Fig. 12f). To understand the underlying mechanism of SEPT2 regulation, specific siRNAs were used to block the PERK, IRE1, and ATF6 pathways. Knockdown of IRE1 decreased the expression of SEPT2, while inhibition of the other two pathways did not (Supplementary Fig. 12g). Since XBP1 is the most important transcription factor downstream of the IRE1 pathway, we generated XBP1-deficient iBMDMs to test whether SEPT2 expression is regulated by XBP1 (Supplementary Fig. 12h). The results showed that SEPT2 expression could no longer be induced by viral infection (Fig. 7h). These data suggested that SEPT2 expression is induced by the transcription factor XBP1.

By analyzing the Gene Transcription Regulation Database (GTRD, http://gtrd.biouml.org), we found that there were multiple potential binding sites of XBP1 on the SEPT2 promoter. To locate the binding site, the −2345 to +85 bp SEPT2 promoter region was amplified and truncated by approximately 500 bp each time. A dual-luciferase reporter assay showed that only the 2.5k promoter responded sharply to exogenous sXBP1 (spliced XBP1) expression, suggesting that the binding site was located at the distal end of the SEPT2 promoter (Supplementary Fig. 12i). We then identified that the −1688 to −1684 bp (CCACG) and the −2116 to −2111 bp (CACGTC) fragments of the SEPT2 promoter were conserved binding motifs of XBP1[53,54]. Exogenously expressed sXBP1 significantly increased the activity of the SEPT2 promoter but not a mutant promoter where CACGTC was converted to CAAAAA in a dose-dependent manner (Supplementary Fig. 12j). Furthermore, the results of chromatin immunoprecipitation showed that sXBP1 protein bound to the SEPT2 promoter under physiological conditions, and this interaction was enhanced by VSV infection (Supplementary Fig. 12k). The supershifted band of the electrophoretic mobility shift assay results demonstrated the specificity of the in vitro binding between the sXBP1 protein and the SEPT2 promoter (Supplementary Fig. 12l). These results confirmed that XBP1 is a transcriptional activator of SEPT2.

Based on these results, we explored the clinical significance of this study by measuring SEPT2 levels in peripheral blood mononuclear cell (PBMC) samples collected from influenza patients with/without symptoms of cytokine storms. We showed that SEPT2 levels were relatively lower in the PBMCs of influenza patients with cytokine storms than in those of healthy individuals (Fig. 7i). There was no significant difference in SEPT2 expression levels between males and females, indicating that sex may not affect the SEPT2-related inflammatory responses (Supplementary Fig. 12m). Furthermore, by analyzing the relationship between SEPT2 and proinflammatory cytokines, we discovered that SEPT2 was negatively correlated with IL-6 (Fig. 7j) and IL-12 p40 (Fig. 7k) in influenza patients, which confirmed our finding that SEPT2 is involved in the negative regulation of inflammation. Next, we attempted to suppress the burst of proinflammatory cytokines in clinical samples using the commercial-specific XBP1 activators IXA4 and APY29. The results showed that both IXA4 and APY29 could activate XBP1 splicing (Supplementary Fig. 12n), thereby increasing the expression of SEPT2 in PBMCs collected from influenza patients with cytokine storms (Supplementary Fig. 12o). Additionally, we found that M1-like hyperpolarization was significantly inhibited by IXA4 (Fig. 7l) and APY29 (Fig. 7m). Furthermore, ProcartaPlex multiplex cytokine assays confirmed that IXA4 and APY29 effectively alleviated the outburst of proinflammatory cytokines in influenza patients (Fig. 7n). These data suggest that IXA4 and APY29 are promising preclinical candidates for the treatment of viral infection-induced cytokine storms.

Collectively, we propose SEPT2-mediated negative feedback regulation in IFN-γ-independent M1-like autoactivation and inflammation (Fig. 8). When unfolded protein accumulates due to mild infection, SEPT2 expression is induced due to activation of the IRE1 pathway; additionally, SEPT2 can suppress ER stress by promoting the acetylation of HSPA5. This negative feedback regulation controls the

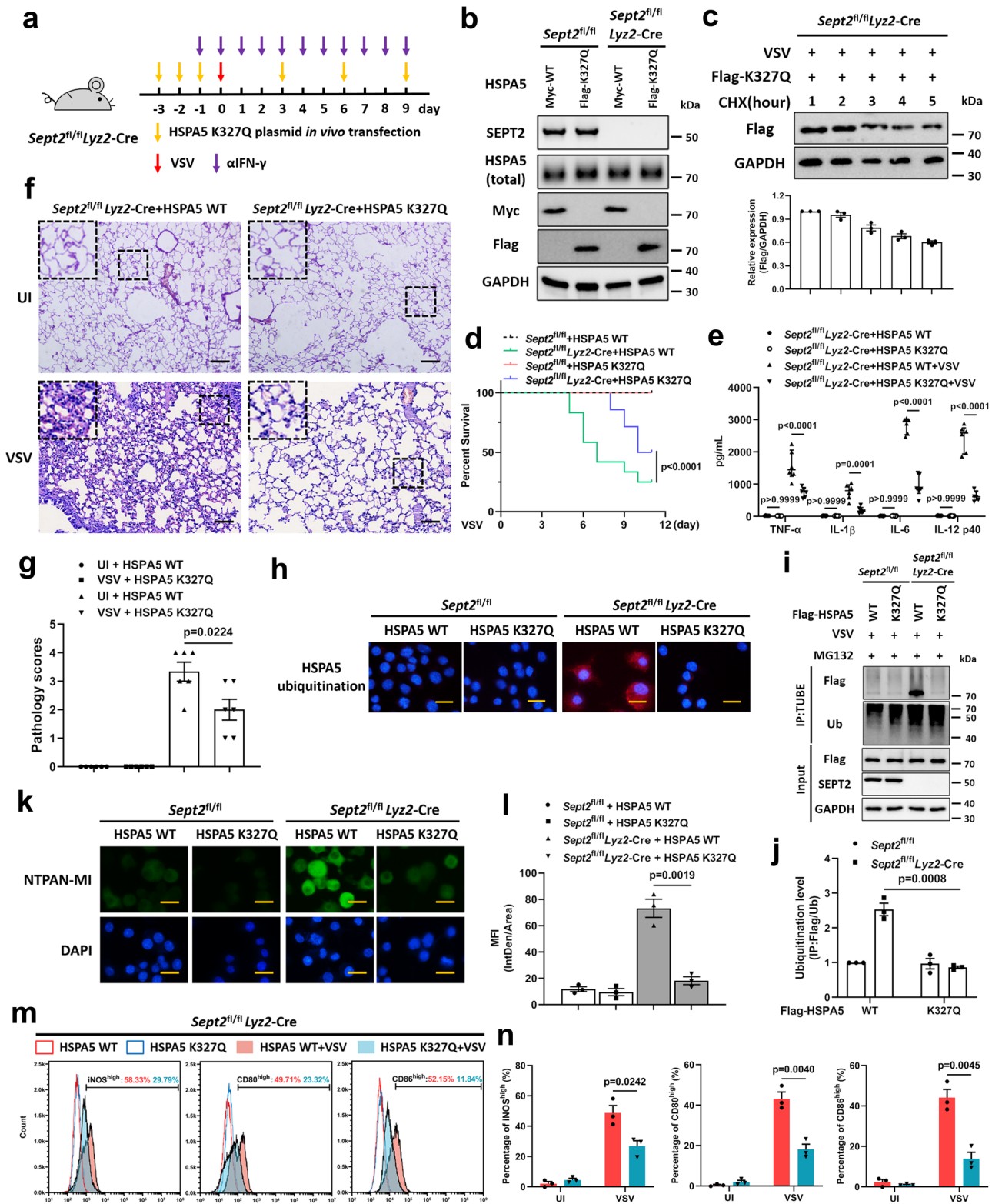

fluctuation of ER stress and inflammation within a certain range, thereby maintaining homeostasis. However, when this mechanism is disrupted due to factors such as SEPT2 deficiency or severe infection, the continuous accumulation of ER stress will lead to excessive inflammation and tissue damage.

## Discussion

SEPTIN family is a major constituent of bud neck during cell division and is increasingly recognized as the distinct fourth component of mammalian cytoskeleton[21]. Previous report has shown that SEPT2 can interact with actin-based structures. The SEPT2-containing fibers physically contact actin bundles and focal adhesion complexes[55]. SEPT2 also participates in the Drp1-mediated mitochondrial fission. SEPT2 depletion reduces Drp1 recruitment and results in mitochondrial elongation[51]. In this study, we show that SEPT2 deletion promotes M1-like polarization in virus-infected macrophages by controlling the UPR. We tentatively term this newly discovered pathway the unfolded protein-derived proinflammatory autoactivation of macrophages

**Fig. 6 | HSPA5 K327Q prevents hyperpolarization in SEPT2-deficient mice.**
**a** Schematic diagram of HSPA5 K327Q plasmid in vivo transfection. **b** The in vivo
transfection efficiency of HSPA5 WT and HSPA5 K327Q was detected in lungs at day
7 without VSV infection. $n = 3$ in each group (**b**). **c** PMs were transfected with HSPA5
K327Q plasmids and infected with VSV (MOI = 1). CHX (50 μg/mL) was used to
inhibit the protein synthesis. The expression of HSPA5 K327Q was detected. $n = 3$ in
each group (**c**). **d** Survival of $Sept2^{fl/fl}$ (HSPA5 WT, $n = 15$. HSPA5 K327Q, $n = 13$) and
$Sept2^{fl/fl}$ $Lyz2$-Cre (HSPA5 WT, $n = 12$. HSPA5 K327Q, $n = 14$) mice intraperitoneally
infected with $1 \times 10^7$ PFU VSV. **e** ELISA analysis of proinflammatory cytokines in
BALF at 7 dpi. $n = 6$ in each group (**e**). **f, g** H&E staining (**f**) and the pathology scores
(**g**) of lung lesions at 7 dpi. Scale bar = 400 μm. $n = 6$ in each group (**f, g**).
**h–n** $Sept2^{fl/fl}$ and $Sept2^{fl/fl}$ $Lyz2$-Cre PMs were transfected with HSPA5 WT or HSPA5

K327Q plasmids and then infected with VSV (MOI = 1) for 12 h. **h–j** The ubiquitina-
tion level of HSPA5 was detected by PLA (**h**) and TUBE analysis (**i**). Scale bar = 20 μm
(**h**). Quantitative data of the TUBE analysis are graphed in (**j**). **k, l** NTPAN-MI probe
was used to determine the accumulation of unfolded proteins (**k**). Scale bar = 20
μm. The MFI was quantitated and shown as IntDen/Area (**l**). **m, n** The expression of
iNOS, CD80, and CD86 was detected by flow cytometry (**m**). The uninfected con-
trols were shown as the blank peaks. Quantitative data are graphed in (**n**). $n = 3$ in
each group (**h–n**). MG132 (10 μg/mL) was used to inhibit the proteasomal degra-
dation of HSPA5 (**h–j**). Data are shown as Kaplan–Meier curves (**d**) and the mean ±
s.e.m. (**c, e, g, j, l, n**). Log-rank (Mantel–Cox) test (**d**) and one-way ANOVA followed
by Bonferroni post hoc test (**e, g, j, l, n**) was used for data analysis. UI uninfected.
Source data are provided as a Source Data file.

(UPAM). In contrast to the fact that macrophage activation by IFN-γ
pathway requires helper cells, the initiation of M1-like polarization by
UPAM is directly derived from unfolded proteins following infection,
and it is a constitutive pathway that is independent of classical IFN-γ
signaling. These data confirm that macrophages have the ability to
induce proinflammatory autoactivation independently. Most impor-
tantly, UPAM not only occurs in the absence of IFN-γ but also does not
rely on TLRs, which can eliminate the endotoxin tolerance effects. In
UPAM, SEPT2 is regulated by the IRE1-XBP1 axis, and its expression
increases with the accumulation of unfolded proteins. On the other
hand, SEPT2 reverses ER stress by promoting HSPA5 acetylation. In this
case, SEPT2 acts as a "brake" in UPAM to constrain M1-like polarization
and proinflammatory cytokine release within a controllable range, thus
maintaining homeostasis. However, when this brake is disrupted due
to SEPT2 deficiency or severe infection, unfolded proteins continue to
accumulate and accelerate UPAM, leading to excessive inflammation
and tissue damage.

Considering that SEPT2 serves as a speed regulator in the UPAM, it
is tempting to speculate that SEPT2 may control the outcomes of
macrophages upon infection. We show that SEPT2 is inversely corre-
lated with the proinflammatory cytokines IL-6 and IL-12 p40 in clinical
PBMC samples (Fig. 7j, k), and the small molecule compounds IXA4
and APY29 could activate SEPT2 expression and effectively alleviate
the proinflammatory cytokine burst in influenza patients (Fig. 7l–n),
indicating that SEPT2 is a promising target for viral infection-induced
cytokine storms. Likewise, using the highly pathogenic (PR8F strain)
and low pathogenic (PR8M strain) influenza A virus infection models,
we found that SEPT2 expression was successfully induced in mice
infected with the PR8M variant. In contrast, the SEPT2 level remained
relatively low with PR8F infection. The failure of SEPT2 induction and
consequent uncontrolled UPAM partly explains the severe inflamma-
tion and increased mortality in mice by PR8M infection. The PR8M and
PR8F variants are homologous in sequence to the PR8 strain. Previous
studies have shown that the differences in the folding of hemaggluti-
nin contribute to the higher virulence of the PR8F variant[52]. We
hypothesized that the failure to induce SEPT2 expression may be
closely related to hemagglutinin of the PR8F variant. However, this
hypothesis needs to be verified further experimentally.

Both acetylation and ubiquitination are widespread and versatile
protein posttranslational modifications that provide precise control for
the organization and function of proteins. The two modifications have
their own unique machinery, while at the same time are interrelated. In
fact, nonhistone acetylation is considered to regulate ubiquitylation
and proteasome-dependent degradation by competing with the same
lysine residue[48]. P300-mediated acetylation of SMAD family member 7
at K64 and K70 prevents ubiquitylation at the same residues, thereby
promoting protein stability[56]. In the current work, we showed that
SEPT2 recruits the acetylase ATAT1 to promote the acetylation of
HSPA5, which is essential for its function in clearing unfolded proteins.
In contrast, SEPT2 deficiency results in the binding of the E3 ligase
SCNN1B to HSPA5, directing the ubiquitination-mediated degradation
of HSPA5. As expected, competition between acetylation and

ubiquitination occurs at the K372 site of HSPA5. Through structural
analysis of ATAT1/HSPA5 and SCNN1B/HSPA5 complexes, we found that
ATAT1 has a stronger affinity for HSPA5, which is the fundamental
reason why HSPA5 is preferentially acetylated under physiological
conditions. Although we showed that SCNN1B interacts with HSPA5
directly (Figs. 3i, 5f), this direct interaction between SCNN1B and HSPA5
appears weak (Fig. 5h). We speculate that other interacting proteins or
posttranslational modifications might be involved in the formation of
this complex. Further, we found that ATAT1 directly interacted with
HSPA5 in the in vitro system (Figs. 4f, 5f, g). Interestingly, the priority of
ATAT1 binding to HSPA5 had to be displayed in the presence of SEPT2 in
the in vivo system (Fig. 4e), indicating an underlying role of SEPT2 in
ATAT1 recruitment. We hypothesize that there may be a negative reg-
ulator that hinders the direct binding of ATAT1 to HSPA5 under phy-
siological conditions, and the role of SEPT2 is to remove this negative
regulator, allowing ATAT1 to preferentially bind to HSPA5. In addition,
we showed that SCNN1B existed in the ATAT1-HSPA5 complex in the
presence of SEPT2 (Fig. 4e), suggesting that SEPT2, ATAT1, SCNN1B and
HSPA5 may form a large complex. However, these hypotheses need to
be verified by experiments.

The SEPTIN cytoskeleton is widely recognized as hetero-
oligomeric complexes. For example, SEPT2 can act as a homodimer
structure which is crucial for the formation of SEPTIN-SEPTIN
interactions[57], such as the SEPT2-SEPT6-SEPT7 complex[49]. In this
study, we show that SEPT2 forms a homo-oligomeric complex when
performing the function of regulating HSPA5 (Supplementary
Fig. 9a–c). Moreover, unlike SEPTINs' other biological functions, which
occur widely in a variety of cell types, the role of SEPT2 in balancing
HSPA5 acetylation and ubiquitination is macrophage-specific (Sup-
plementary Fig. 10d). Therefore, we speculate that the specific func-
tion of SEPT2 in this study is quite different from the cytoskeleton.
Further investigations are required for a full understanding of SEPT2 in
the regulation of HSPA5 acetylation in response to different stimuli,
and the contribution of UPAM in macrophage M1-like activation
should be further determined.

## Methods
### Ethic statements
This study was carried out in strict accordance with the Guidelines for
the Care and Use of Animals of Chongqing University, and was com-
pliant with the "Guidance of the Ministry of Science and Technology
(MOST) for the Review and Approval of Human Genetic Resources".
Animal experimental procedures were approved by the Laboratory
Animal Welfare and Ethics Committee of Chongqing University. Blood
samples were obtained from Chongqing Public Health Medical Center.
Healthy individuals and patients providing blood samples were given
informed consent. The ethics committee approved this consent
procedure.

### Mice
WT C57BL/6J mice were purchased from Jackson Laboratory. To
generate myeloid-cell-specific SEPT2-deficient ($Sept2^{fl/fl}$ $Lyz2$-Cre)

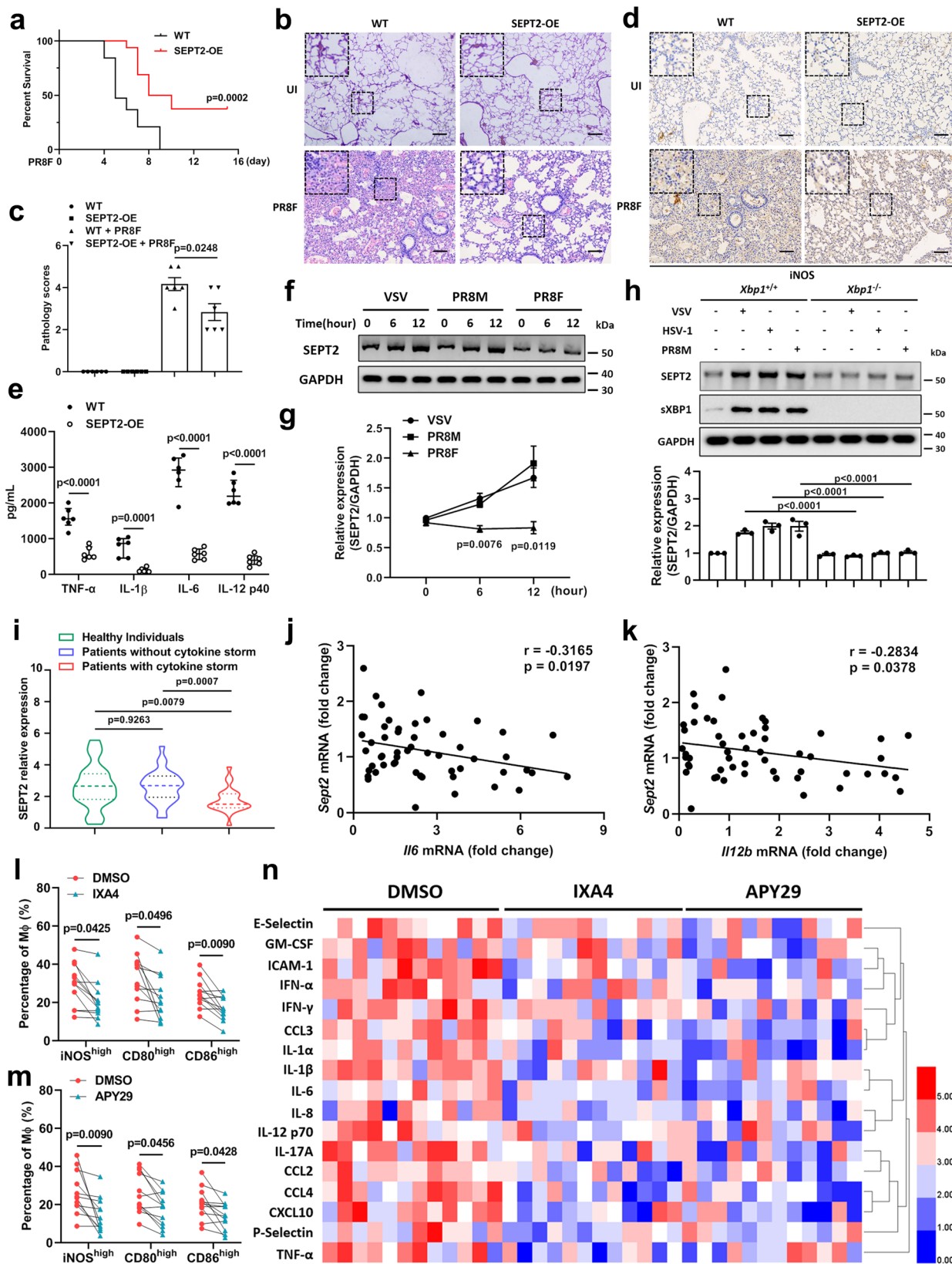

mice and tamoxifen-inducible SEPT2 conditional knockout (*Sept2*[fl/fl] *Lyz2*-Cre-ERT2) mice, we inter-crossed mice that contained loxP sequence flanking the 4-5 exons of SEPT2 (*Sept2*[fl/fl]) with *Lyz2*-Cre mice or C57BL/6JSmoc-*Lyz2*[em1(2A-CreERT2-WPRE-pA)Smoc] mice (Shanghai Model Organisms Center, Inc., Shanghai, China), respectively. HSPA5[K327Q] mice (B6/JGpt-*Hspa5*[em1(K327Q)]/Gpt) were constructed and

identified by GemPharmatech Co., Ltd. (Nanjing, China). Referring to previous reports, the K > Q mutation was constructed to mimic acetylation[58,59]. To obtain the p.K327Q point mutation, the nucleic acid change, c.979A>C, was introduced into the endogenous *Hspa5* mouse locus using sgRNA (5′-TGTCTTCTCAGCATCAAGCA-3′, Chr2: 34774335(+)) and the repair oligonucleotide (5′-

**Fig. 7 | SEPT2 overexpression inhibits PR8F-induced hyperinflammation.**
**a** Survival of C57BL/6 J mice after being intravenously injected with empty vector (WT, n = 19) or SEPT2 overexpressing vector (SEPT2-OE, n = 16) and infected with 1 × $10^4$ PFU PR8F. Daily intraperitoneal injection of αIFN-γ (12 mg/kg) was performed to block IFN-γ signaling. **b**, **c** H&E staining (**b**) and the pathology scores (**c**) of lung lesions at 7 dpi. Scale bar = 400 μm. **e** ELISA analysis of proinflammatory cytokines in BALF at 7 dpi. n = 6 in each group (**b**–**e**). **f**, **g** Immunoblot analysis of SEPT2 in iBMDMs after being infected with VSV (MOI = 1), PR8M (MOI = 1) or PR8F (MOI = 1) for 12 h (**f**). Quantitative data are graphed in (**g**). **h** Immunoblot analysis of SEPT2 in $Xbp1^{-/-}$ iBMDMs after being infected with VSV (MOI = 1), HSV-1 (MOI = 5) or PR8M (MOI = 1) for 12 h. n = 3 in each group (**f**–**h**). **i** qRT-PCR analysis of SEPT2 in PBMCs obtained from healthy individuals (n = 21), influenza patients without cytokine storm (n = 29) and influenza patients with cytokine storm (n = 25). **j**, **k** Relative expressions of SEPT2 and proinflammatory cytokines, Il6 (**j**) and Il12b (**k**), in PBMCs obtained from influenza patients. n = 54 in each group (**j**, **k**). **l**–**n** PBMCs obtained from influenza patients with cytokine storm were pretreated with IXA4 (10 μM) (**l**) or APY29 (1 μM) (**m**) for 6 h. The expression of iNOS, CD80, CD86 (**l**, **m**) and proinflammatory cytokines (**n**) was detected. n = 12 in each group (**l**–**n**). Data are shown as Kaplan–Meier curves (**a**), the mean ± s.e.m. (**c**, **e**, **g**, **h**) and the median ± interquartile (**i**). Log-rank (Mantel–Cox) test (**a**), one-way ANOVA followed by Bonferroni post hoc test (**c**, **e**, **g**, **h**), Mann–Whitney U test (**i**), Spearman rank correlation analysis (**j**, **k**) and paired Student t test (**l**, **m**) were used for data analysis. The statistical tests used in (**i**–**m**) were two-sided. UI uninfected. Source data are provided as a Source Data file.

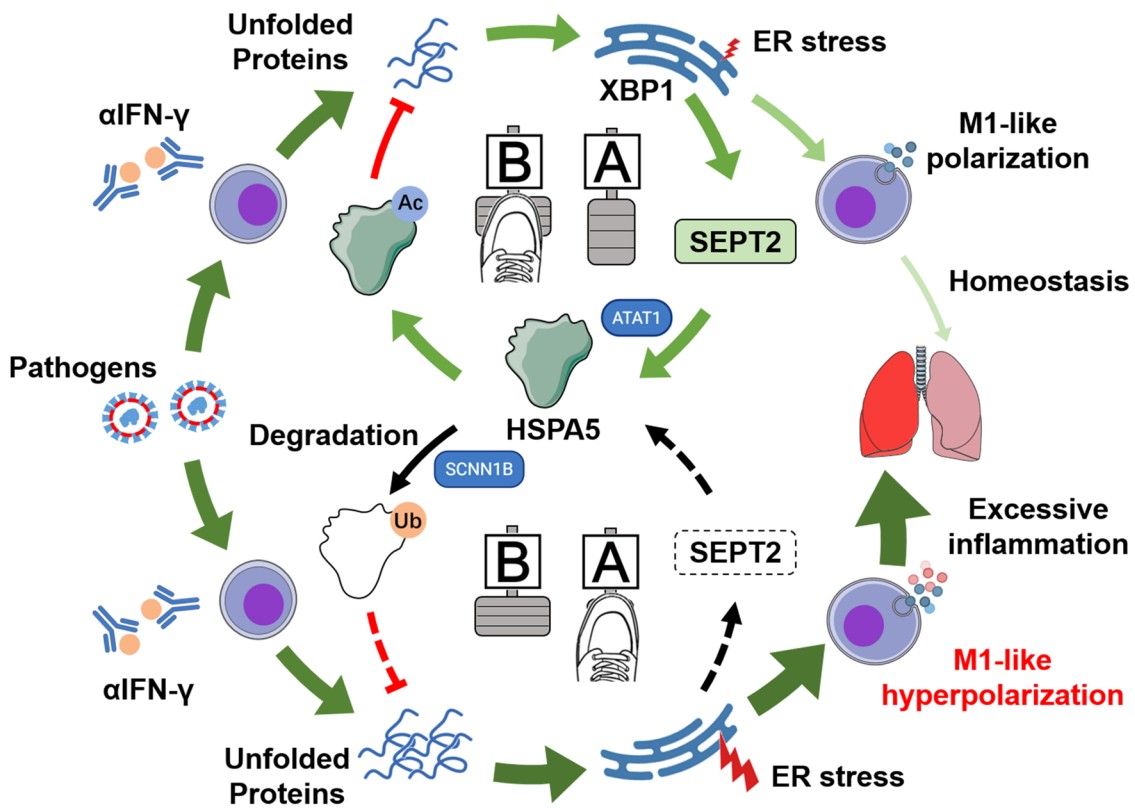

**Fig. 8 | Schematic diagram of this study.** SEPT2 serves as a speed controller in the negative feedback regulation of IFN-γ-independent macrophage proinflammatory activation. A= Accelerator, B= Brake.

AGAAAAGGCTAAGAGAGCCTTGTCTTCTCAGCATCAAGCAAGAATT GAAATTGAGTCCTTCTTCGAAGGAGAAGACTTCTCAGAGACCCTTAC TCGGGCCCAATTTGAAGAGCTGAACATGGTATGCTCCTTGACAGTG CTAATGGAATCCGCTTAGACTGTAGAATTTGGGATAACTAAATAAGG TCTGGGTGGTCAGC-3′). The sgRNA, repair oligonucleotide and Cas9 mRNA were microinjected into pronucleus of C57BL/6J embryos, which were implanted into pseudo-pregnant females to generate F0 heterozygotes. F0 heterozygotes were inter-crossed with WT mice to generate F1 heterozygotes. The F1 heterozygous × F1 heterozygous cross was set up to obtain F2 homozygous and WT littermates. F2 homozygous were used for the experiments. All mice were housed in a specific pathogen-free animal facility, maintained on a 12 h light/ 12 h dark cycle, at a temperature of 22 °C and 45% humidity, with ad libitum access to food and water. At the beginning of the experiment, indicated numbers of six- to eight-week-old mice were randomly grouped and treated accordingly in each experimental condition. Mice of both sexes (equal distribution) were used for experiments.

## Cell lines

iBMDMs were kindly provided by Dr. Feng Shao (National Institute of Biological Sciences, Beijing, China). The construction method refers to previous report[60]. In brief, primary murine bone marrow cells were obtained from C57BL/6 mice and infected with Cre-J2 retrovirus. After 24 h of infection, they were switched to complete DMEM with 20% L929 conditioned media and incubated for another 24 h. Long-term cultures were performed to select iBMDMs and single-cell clones were harvested by limiting dilution. iBMDM$^{Tet-on\ SEPT2\ shRNA}$ cell line was generated by stable transfection of Tet-pLKO-SEPT2 shRNA-puro vector into iBMDMs. The Tet-pLKO-SEPT2 shRNA-puro vector was constructed by inserting the SEPT2 shRNA sequence (ds oligo, 5′-CC GGGACTGATCTCTACCCAGAAAGAATTCGAA AATTCTTTCTGGGTAG AGATCAGTCTTTTTGGTACC-3′,   5′-AATTGGTACCAAAAAAGACTG ATCTCTACCCAGAAAGAATTTTCGAATTCTTTCTGGGTAGAGATCAGT C-3′) between the Age I and EcoR I sites of Tet-pLKO-puro. The recombinant was transfected into iBMDMs for 48 hours, followed by screening for 7 days using puromycin (3 μg/mL). $Atat1^{-/-}$, $Atat1^{-/}$

$^-Scnn1b^{-/-}$ and $Xbp1^{-/-}$ iBMDMs were generated using CRISPR-Cas9. To be detailed, sgRNAs were designed for each gene (sgRNA sequences are listed in Supplementary Data 3), and ligated into pSpCas9(BB)−2A-Puro (PX459) after being digested by Bbs I (New England Biolabs, MA, USA). The recombinant was then transfected into iBMDMs using Lipo 3000 Transfection Reagent (Invitrogen, Thermo Fisher Scientific Inc., CA, USA). After 48 hours, RPMI 1640 medium (Gibco, Thermo Fisher Scientific Inc.) containing 3 µg/mL puromycin was used for screening for 7 days to gain cell pools. Single cell clones were finally obtained by limiting dilution. PMs were isolated as previously described[61]. Briefly, 10 mL of HBSS containing 2 mM EDTA and 2% fetal bovine serum (FBS, Gibco) was injected into the peritoneal space. Gently rub the mouse abdomen and harvest HBSS by insulin syringe. PMs isolation was repeated two to three times until $1 \times 10^6$ adherent cells were obtained. Human PMBCs were isolated from EDTA-treated whole blood through density gradient centrifugation. NIH-3T3 (ATCC, CRL-1658), L929 (ATCC, CCL-1) and HEK-293FT cells (Invitrogen, R70007) were cultured in DMEM medium with 10% FBS (Gibco) at 37 °C, 5% CO$_2$. PMs, TC-1 cells (ATCC, CRL-2493), PBMCs, iBMDMs and other cell lines generated from iBMDMs were cultured in RPMI 1640 medium with 10% FBS (Gibco) at 37 °C, 5% CO$_2$. All cells were negative for mycoplasma.

## Viruses
VSV (ATCC, VR-1238), Sendai virus (SeV, ATCC VR-907), Encephalomyocarditis virus (EMCV, ATCC VR-129B), HSV-1 (ATCC, VR-1789) and Human adenovirus 5 (Adv, ATCC VR-5) were propagated and amplified in Vero-E6 cells. The virus strain was diluted with MEM medium (Gibco) and inoculated in a monolayer of Vero-E6 cells. After the cytopathic effect was observed, the supernatant was harvested, clarified by centrifugation and filtered by 0.45 µm sterile filters. Virus stocks were stored at −80 °C until further usage. Mouse-adapted variants of the influenza virus strain A/Puerto Rico/8/34, PR8M, and PR8F, were rescued using eight-plasmid transfection system based on the WT PR8 sequences as previously described[62,63]. Virus stocks were propagated in the chorioallantoic cavity of 10-day-old pathogen-free embryonated chicken eggs at 37 °C for 48 h. At the beginning of infection experiments, viral titers were determined by TCID$_{50}$ assay in MDCK cells.

## Blood samples
Blood samples from healthy individuals ($n = 21$, 10 males and 11 females, aged $32.13 \pm 8.57$ years) were obtained from Chongqing Public Health Medical Center. Blood samples from influenza patients with/without cytokine storm were obtained from Chongqing Public Health Medical Center between December 2018 and February 2022. Both influenza patients without cytokine storm ($n = 29$, 19 males and 10 females, aged $34.15 \pm 6.34$ years) and patients with cytokine storm ($n = 25$, 17 males and 8 females, aged $37.59 \pm 4.12$ years) were confirmed as being infected with Influenza A virus by qRT-PCR. IXA4 (HY-139214, MedChemExpress, NJ, USA) and APY29 (HY-17537, MedChemExpress) were used to activate XBP1 splicing in PBMCs. The XBP1 mRNA splicing level and the expression of SEPT2 were detected by qRT-PCR. The expression of proinflammatory cytokines in PBMCs was detected using Inflammation 20-Plex Human ProcartaPlex Panel (EPX200-12185-901, Thermo Fisher Scientific).

## Virus infection and IFN-γ signaling blockade
Six- to eight-week-old mice were randomly divided into control and experimental groups. All mice were age- and sex-matched. In order to inhibit IFN-γ-induced macrophage activation, mice received intraperitoneal injection of αIFN-γ (12 mg/kg) every day, starting 1 day before viral infection and continuing until the end of the experiments. The virus was diluted with sterile PBS and each mouse was intraperitoneally infected with VSV ($1 \times 10^7$ PFU), HSV ($1 \times 10^8$ PFU), PR8M ($1 \times 10^4$ PFU) or PR8F ($1 \times 10^4$ PFU). Mice in the uninfected control group were given an equal volume of sterile PBS. Animal experiments adhered to the Guidelines for the Care and Use of Animals of Chongqing University. Mice that reached the humanitarian endpoint (>15% weight loss) or completed the observation period were euthanised by cervical dislocation. For in vitro experiments, cells were plated at a density of $1 \times 10^6$ cells/60 mm plastic dish and infected with VSV (MOI = 1), SeV (MOI = 1), EMCV (MOI = 1), HSV-1 (MOI = 5), Adv (MOI = 1), PR8M (MOI = 1) or PR8F (MOI = 1) for the indicated times.

## Statistical analysis
The sample size was based on empirical data from pilot experiments. The investigators were blinded during data collection and analysis. Results were presented as mean ± s.e.m. Survival data were shown as Kaplan−Meier curves. Paired or unpaired two-tailed Student's $t$ test and Mann−Whitney $U$ test were used for two-group comparisons. One-way ANOVA followed by Bonferroni post hoc test was used for multiple comparisons. Survival data were compared using Log-rank (Mantel−Cox) test. A value of $P < 0.05$ was considered significant. Statistical analysis was performed using GraphPad Prism 8.3.0 and ImageJ 1.52a.

## Reporting summary
Further information on research design is available in the Nature Portfolio Reporting Summary linked to this article.

## Data availability
The RNA-seq data generated in this study is publicly available in Gene Expression Omnibus at GSE213863. Source data are provided with this paper.

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

## Acknowledgements

The authors would like to thank Dr. Feng Shao (National Institute of Biological Sciences, Beijing, China) for providing iBMDMs. This work was supported by the National Natural Science Foundation of China (No. 81970008, H.W., No. 82000020, X.L., No. 82241059, Y.L., No. 82125022, Y.L.), Fundamental Research Funds for the Central Universities (No. 2023CDJXY-009 and 2019CDYGZD009, H.W.), Natural Science Foundation of Chongqing, China (No. CSTB2023NSCQ-MSX0402, H.W.) and Chongqing Talents: Exceptional Young Talents Project (No. cstc2021ycjhbgzxm0099, H.W.). The funders had no role in study design, data collection and analysis, decision to publish, or preparation of the manuscript.

## Author contributions

H.W., B.F., X.L., and Y.L. conceived and designed the study. H.W., B.F., X.L., Y.L., Y.X., Z.S, W.X., B.X., S.T., D.G., F.L., L.W. and J.J. performed the experiments. H.W., B.F., X.L., and Y.L. analyzed the data. H.W., B.F., X.L., and Y.L. wrote the manuscript. All authors read and approved the final manuscript.

## Competing interests

The authors declare no competing interests.
