## [Peer Review File · Nature Communications]

SEPTIN2 suppresses IFN- γ -independent macrophage proinflammatory activationREVIEWER COMMENTS

Reviewer #1 (Remarks to the Author):

Peer Review

“SEPTIN2 negatively regulates IFN- γ -independent macrophage proinflammatory activation”

Content summary with my opinion of the major findings:

The manuscript presented by Beibei Fu and colleagues explores the consequences of Sept2 gene deletion during VSV and HSV infection in animals and in cells. Fu and colleagues reveal Sept2 unexpectedly limits inflammation caused by these infections. Using an array of biochemical and genetic tools, Fu and colleagues propose that Sept2 limits the accumulation of unfolded proteins and ER stress during infection by promoting the acetylation of HSPA5 by ATAT1, which can promote macrophage polarisation in the absence of IFN gamma. The authors further suggest that SEPTIN-2 facilitates the acetylation of HSPA5 at lysine K327, which competitively limits K48-mediated ubiquitination at this same site by the E3 ubiquitin ligase SCNN1B. Acetylation of HSPA5 in turn is hypothesised to relieve the ER stress and limit disease-causing inflammation. In myeloid-deleted Sept2 knockout animals, acetylation of HSPA5 at K327 is reduced and is instead ubiquitinated at this site, causing its proteasomal degradation. The accumulation of unfolded proteins triggers the UPR and drives disease-causing inflammation.

While the findings are of potential interest, the (sheer) number of experiments presented often lack important controls and there are a number of issues that require attention:

- Lack of detail. In many places the figure legend or associated methods section lacks sufficient detail to know how an experiment has been performed or how many times each experiment was repeated, and results verified (which should be clearly stipulated in the figure legends).
- Lack of controls. Some in vivo data and most immunoprecipitations lack suitable controls that prevent conclusions that can be drawn from these experiments. Furthermore, molecular weight markers are missing from almost all western blots, which makes it problematic to assess the bands shown.
- Confusing logic. In some places the logic is difficult to follow. These are noted below and require some clarification/rephrasing to improve the flow of the manuscript.
- Poor analysis of ubiquitination. As noted particularly in my comments for Figure 3, much of the analysis performed examining ubiquitination is not sufficient. For example, performing a co-IP and probing for ubiquitin does not mean that the protein you pulled down is ubiquitinated. The ubiquitin could be on other interacting proteins. Alternative methods (some I have noted below) should be employed to properly examine the ubiquitination of HSPA5.
- I did not see any measure of viral titres throughout the manuscript. The authors need to determine if SEPT2 loss might just allow for increased viral replication, and hence burdens (in vitro and in vivo). This is important as it might offer an alternative explanation to the model put forward by the authors.
- All In vivo transfection efficiency data needs to be shown.

Detailed comments

Introduction

• Macrophage biology is more complex than the simple M1/M2 paradigm, particularly in vivo (see doi: 10.12703/P6-13, doi: 10.1161/CIRCRESAHA.116.309194, doi: 10.3389/fimmu.2015.00370.). To keep relevant and up to date with the field, the authors must frame their work with the current view of macrophage biology, either or both in the introduction and discussion.

- Lines 61 – 63: During an infection, the TLR activator comes first, followed by IFN γ . By discussing M1 polarisation as firstly being primed with IFN γ , then being exposed to a TLR activator, the authors present an oxymoron that is inconsistent with what happens in vivo. The authors should clarify this statement, or instead talk about more specifically about why IFN γ is important for macrophage biology (see doi: 10.1189/jlb.0603252).

- In the introduction, the authors set up a framework for M1 polarisation by stating that IFN γ + LPS can fully activate the M1 phenotype, as can IFN γ -independent activation with T-cell-derived GM-CSF and during macrophage activation syndrome in IFN γ knockout cells. However, the authors specify that LPS treatment alone does not fully activate the M1 phenotype. Throughout their manuscript the authors have not measured the response of their IFN γ -independent macrophage activation against complete M1 polarisation either in the presence of IFN γ or an 'incomplete'. It is unclear if this activation is complete, or a partial response as one might see with LPS stimulation alone. I recommend the authors re-phrase their introduction by removing the discussion on endotoxin tolerance effects in lines 65 – 66. This paragraph may benefit from additional detail on the IFN γ -independent activation of macrophages and highlighting what is known and what is not known.

- In line 111, please replace the word 'controlling' with 'promoting' or another appropriate term that indicates the direction of control.

Methods

- The methods section lacks detail so much so that this work would not be able to be accurately repeated by independent researchers. Please improve the level of detail in all current and future methods sections to sufficient detail that other researchers could repeat the experiments in this manuscript.

- Mice and IFN- γ signaling blockade

- o Line 651, replace 'hybridized' with 'inter-crossed'.

- o Please provide more detail regarding the animal experiments performed including but not limited to: sex of animals used for cohorts, number of cohorts, age of animals at beginning of experiment, genetic background of animals.

- Cell lines and viruses

- o This section lacks detail on the generation of the iBMDMs (referring to a previous publication is fine but please provide some detail).

- o Please provide detail on the culturing conditions of all cell lines used in the study

- o Please provide more detail regarding the propagation of virus.

- o Please provide more detail regarding the animal experiments performed including but not limited to: sex of animals used for cohorts, number of cohorts, age of animals at beginning of experiment, genetic background of animals, vehicle of virus, control injections performed (if any).

- o Please provide more detail on the density of cells for in vitro experiments.

- Blood samples

- o Note the experimental groups are partially biased toward male participants. The controls are more balanced. The authors may consider some form of analysis to test if sex contributes to their differences. Please provide a rationale why sex would not contribute if analysis is not performed.

- Statistical Analysis

- o Please provide more information on the specific statistical analysis applied to each type of data.

- Plasmid and siRNA transfection

- o Please specify specific detail on how the transfections were performed. How much DNA/siRNA, how long?

- o This section also only specifies in vivo transfection of nucleic acid constructs. Please also specify the method used in vitro.

- High content screening

- o Please provide information on the statistical methods applied to this analysis

- Histopathology analysis

- o Please outline the pathology scoring parameters, whether the scoring was performed blinded and how many measures per animal were taken to obtain a final value.

- qRT-PCR
 - o Please provide the specific PCR protocol used.
- Western blotting and coimmunoprecipitation
 - o Please provide detail on the lysis buffer used for standard cell lysis.
 - o Please provide dilution of antibodies used and into what solution (e.g. BSA, skim milk)
- Enzyme linked immunosorbent assay
 - o Please specify whether samples were diluted and/or whether all sample values fell within the standard curve of the assay.
- RNA-sequencing
 - o Please provide GEO ID.
 - o Please provide citations for the software used to analyse this data.
 - o To demonstrate quality of the RNA-sequencing data, please show a MDS plot to show clustering of the data generated in the RNA-sequencing analysis in a main or supplemental figure.
- Transmission electron microscopy
 - o Please explain 'standard procedures'
 - o Please provide detail on the magnification and scale bar length used for the data presented in figure S3.
- Detection of the accumulation of unfolded proteins
 - o Please provide detail on how these experiments were performed including concentrations of NTPAN-MI.
- Dual-luciferase reporter assay
 - o Please provide detail on the transfection conditions (e.g. confluency), amount of DNA used.
- Chromatin immunoprecipitation assay
 - o Please specify sonication settings and micrococcal nuclease digestion conditions..

Results:

- Please provide molecular weight markers on all probes in all western blots. Please also ensure a loading control is provided in all western blots and that in the immunoprecipitations, you 1) probe for the protein you are pulling down with in the IP samples to ensure the IP is equal among samples, and 2) in the lysate to ensure there was equal starting amounts of protein in the input. This probe will also help explain whether any differences in the IP probe are a consequence of altered expression.
- The authors conclude that SEPT2 loss does not alter oxidative stress, which might explain the increased unfolded proteins and inflammation. However, their own data show dramatically increased iNOS levels in VSV infected SEPT2 knockout cells, and consequent increased nitric oxide generation is a known cause of oxidative stress. The authors should address whether iNOS/NO blockade reverses the UPR and heightened inflammatory responses.
SEPT2 is involved in IFN- γ -independent macrophage proinflammatory activation
- Please provide some form of results from the HCS that show SEPT2 was a hit in this experiment.
- SEPT2 knockout mice do not appear to have been previously described and it would be useful to know if deletion (at least in the myeloid compartment) causes any defect in vivo. Please mention whether these animals are phenotypically normal.
- Sentence starting at line 139 does not make sense. Are the authors referring to previous studies that have used anti-IFN γ to block signalling? If so please provide citations.
- Ideally, the Sept2^{fl/fl}Lys2-Cre animals would be crossed onto an IFN-g knockout. This would definitively show that the response is interferon gamma independent. However, I understand this would take too long for a revision. At a minimum, the authors should perform this infection experiment in the presence and absence of IFN γ to show that the IFN γ inhibitory antibodies are working and to test whether the function of SEPT2 is unveiled only in the absence of IFN γ signaling. Once results from this experiment are obtained, the authors should adjust their conclusions accordingly.
- Please rephrase the sentence beginning in line 176. The first clause of this sentence makes it sound like the authors contest this well-known fact. I suggest removing from lines 176 up to the citation and beginning the sentence with 'viral nucleic acids...'
- Line 187: 'probably' is colloquial and should be replaced with 'likely'.
- The Sept2^{fl/fl} animals do not start to die until around day 6, but the analysis performed in the last four panels F – I are done in <24 hours. Are these changes informative to how the disease progresses more than 5 days after the ex-vivo analysis is performed?

- Figure 1
 - o The pathology scoring from the supplemental could be moved in the main figure for clarity.
 - o The pathology difference in the Sept2 knockout animals does look mildly more severe, as reflected in the pathology score. Some clarification on how endpoint was determined for the survival curve would help assess how much the pathology informs on the survival of the animals.
 - o Please detail the scale bar lengths in the figure legend
 - o Please provide a gating strategy in the supplemental figures for the cell subsets presented in panel D.
 - o Related to panel C, Is there a change in basal cytokine production in the knockouts?
 - o Panels F and H: Please provide some rationale as to why these timepoints were selected when the model proceeds over 15 days.

- Figure S1

- o Figure legend title could be qualified with "nucleic acid sensing" before PRRs.
- o Please specify what the qPCR data is normalised to.
- o Please perform a statistical analysis should be performed on the data generated in S1E - O
- o The authors make a big conclusion that the effect of SEPT2 is independent of nucleic acid sensing PRRs, however the data presented is only representative of one independent repeat. The authors may keep the flow cytometry plots if they wish, but this data would be better presented in the form of a bar graph to compare the infection with no siRNA vs with the various siRNAs and to show all three independent replicates in one graph. The effect of the siRNAs is also hard to gauge, because the control (no siRNA) is in the main figure. The authors should show no-siRNA vs +siRNA upon infection in the one figure, preferably in a bar graph but also in a flow cytometry plot. Lastly, authors should show the efficiency of their siRNAs using western blot, especially if they claim that the siRNA has no effect on the phenotype.

SEPT2 deficiency regulates M1-like hyperpolarization through ER stress

- IL1b secretion is generally accompanied with pyroptotic cell death. Does infection cause cell death? This is an important point as cell death, or deficiency in cell death, has a big impact on viral infections, yet the authors do not determine if SEPT2 loss impacts viral-induced killing of the host cells.
- Line 234 - 236: Please specify the pathways you are looking at.
- In lines 233 - 237, the authors assume that the increased activation observed in figure S3D leads to increased ER stress; however, the data in figure S3D creates a chicken and egg scenario. These pro-inflammatory pathways can be activated by TNF. So, does the activation of these pathways lead to ER stress (as hypothesised by the authors), or does the heightened TNF in the Sept2 knockouts (Figure 1H/I, 2F/H) cause more activation of this pathway? Conclusions drawn in the main text should be adjusted to take this possibility into account.

- Figure 2

- o 2B - The loading control in lanes 5 and 6 do appear mildly increased and may explain the differences seen in ATF4, sXBP1, cleaved/full length ATF6a and pPERK. pIRE1a and peIF2a appear the most convincing. I would suggest repeating this WB to obtain a more consistent loading, or adjust the conclusions based on the clearest data. The blot for full length ATF6a is also over-exposed. Please provide an image with a lower exposure. Please also provide the MOI for this experiment in the figure legend.
- o 2E - the bar (second from left) does not sit at the mean for these samples. Are there additional dots sitting between 30 and 120 that are not shown? please amend graph to rectify this issue. Please also indicate the duration of 4PBA treatment and MOI in the figure legend.
- o Please show efficiency of the siRNAs using WB. As a major conclusion of the manuscript, this is an important control to include.
- o The authors data in figure 2G is surprising. One would presume that if all three are activated, then knocking down one pathway would have little effect. Instead, the authors see that knocking down any pathway has a large effect (on iNOS activity) due to redundancy between the pathways. The authors should employ an additional measure of using WB to examine pEIF2a to assess this. Notably, it is most surprising that siRNAs against PERK and ATF6 had an effect given the mild difference seen by the WB in figure 2B.
- o Please show that the 4PBA reduces ER stress and is working as expected.

- Figure S2

- o The authors should generate an iBMDM TET-ON line with a scramble shRNA to show that the effect seen are more specifically a consequence of SEPT2 knock-down, rather than a consequence

of shRNA expression.

o Do the animals that get corn oil receive anti-IFN-gamma?

- Figure S3

o In S3A, is the p-value the same for each pathway? Please clarify.

o As mentioned in the methods, please provide some data to show the consistency of each sample submitted for RNA-sequencing.

SEPT2 deficiency promotes the proteasomal degradation of HSPA5

- The rationale for looking at HSPA5 presented in the manuscript is not clear. It makes sense that SEPT2 might regulate the activation of PERK/IRE1/ATF6, but why specifically HSPA5? If the authors can convincingly show PERK, IRE1 and ATF6 are all regulated by SEPT2, then you may make this jump as HSPA5 is reported to regulate them all based on doi:

10.1016/j.gene.2017.03.005. However, upon ER stress, HSPA5 activates the UPR (doi:

10.1146/annurev-cellbio-101011-155826) Since the authors contend that Sept2 knockout cells are effectively deficient in HSPA5 due to increased turnover, how do the authors explain the increased activation of IRE1? The authors are proposing HSPA5 has the opposite function to its known and published role. How do the authors explain this?

- The title of this section is worded in the negative and would be clearer if worded as "SEPT2 limits the proteasomal degradation of HSPA5"

- Please provide some data from the HCS of the E3 ubiquitin ligases for HSPA5.

- Is the K327A mutant more stable than the WT? The authors should perform a CHX (cycloheximide)-chase assay in cells to answer this question.

- No data is presented to support the conclusion that "Abnormal degradation of HSPA5 leads to an exacerbation of unfolded proteins, resulting in M1-like hyperpolarization and excessive inflammation". This sentence should be removed.

- Figure 3

o Panel C is poorly controlled as addition of CHX will stop expression of HSPA5 irrespective of whether DOX was added or not. Please provide additional controls for this experiment.

o Panel D does not show ubiquitinated HSPA5. By performing a HSPA5 IP and then probing for ubiquitin, the authors are observing ubiquitinated species that could be on HSPA5 OR on proteins interacting with HSPA5. A better experiment would be to IP HSPA5 and perform USP21 cleavage on the IP eluate. Then, run these samples on a WB and probe for HSPA5. The authors should see laddering on their IP sample and if this is ubiquitin, the laddering would disappear in the DUB-cleavage samples. This would prove that HSPA5 is ubiquitinated. Another but less rigorous way to examine ubiquitinated HSPA5 would be to perform a ubiquitin IP (TUBE), then probe for HSPA5. The authors should see laddering or a smear with this probe.

o Panel D also lacks an important control. In all IPs, the authors must probe for the protein they are pulling down to show equal IP between samples. Otherwise, the increased smearing we are observing in lane 4 could simply be a consequence of the researcher unintentionally aliquoting more beads into that IP.

o Panel E lacks Dox-only and SCNN1B siRNA transfection only controls. Please amend and I suggest presenting the data in such a way that permits a statistical analysis to be performed. The authors should provide western blot verification of this data, including relevant controls (e.g. SCNN1B knockdown efficiency, SEPT2 levels etc).

o Panel F/G/I – please probe for HIS or HSPA5 to show equal IP.

o Panel H could be better labelled to highlight that you are looking at HSPA5.

o I could not see a methods section for the mass spectrometry data. Please provide this and detail the conditions used to generate the sample for Mass spec.

- Figure S4

o Panel A is an important result and I believe should be in the main figure.

o Panel I – This blot is empty? The authors should run an input sample alongside their IP samples to show that the antibodies incubations and detection has been performed correctly. This observation could equally be explained if the incorrect secondary antibody was added.

o Panel H/I/L – please probe for HSPA5 to show that the IP has worked equally across all samples.

SEPT2 promotes the acetylation of HSPA5 by recruiting ATAT1

- Examining the acetylation of HSPA5 followed a non-conventional logic, but the data appears sound.

- Does the acetylation defective mutant HSPA5 have a reduced interaction with SCNN1B?

- Do Sept2 and ATAT1 phenocopy one another if you compare them side-by-side?
- Figure S5A
 - o All IPs should clearly indicate what probes are from input/whole cell lysate samples and which are from the IP. Furthermore, all IPs should be probed with the protein you are pulling down to demonstrate equal IP per sample. All IPs should have an input probe of the protein they are pulling down and a loading control.
 - o Panels E/F are only explored at baseline and not during VSV infection. It would be interesting to show that this is not the case under settings where SEPT2 is known to regulate HSPA5.
 - o Please provide a lower exposure of the ubiquitin probe in panel G. There might be subtle difference not captured in the overexposed image
 - o Panel J – please see my comments for figure S4 panel I (above),
 - o Panel K – IMPORTANT – There are more GAPDH bands than HSPA5, Ac-Lysine bands. Conclusions can't be made from this data.
 - o Panel P figure legend lacks detail. Furthermore, the legends states "...p65 regulated by ATAT1 in iBMDMs..." however, ATAT1 is clearly not regulating p65 acetylation. Do the authors mean VSV infection?
 - o Line 366: I think the authors mean "native-PAGE", instead of "naïve-PAGE"

ATAT1 has a stronger affinity for HSPA5 than SCNN1B

- The interaction between SCNN1b and HSPA5 appears weak. Is their interaction direct or do you think there is more going on here such as another interacting protein or PTM? Perhaps a point of discussion?
- Figure S6
 - o Are these AlphaFold renders of the interaction or are they crystallised constructs? I am struggling to believe SCNN1B directly interacts with HSPA5 due to the high Kd. It is possible there is another unknown member of the complex that co-localises SCNN1B and HSPA5. This possibility has not been excluded.

K327-acetylated HSPA5 prevents M1-like activation and excessive inflammation

- Please show that the K327Q mutation actually stabilises HSPA5 and provide citations to papers that demonstrate K>Q mutations mimic acetylation.
- PLA does not specifically look at ubiquitination of HSPA5, as other proteins in proximity of HSPA5 may also be ubiquitinated and cross-react in this assay. Please show that the K327Q reduces ubiquitination of HSPA5 by IP and western blot or more ideally quantitative mass spec.
- Figure 6
 - o Panel B – What samples are being examined here?. Please provide detail in the figure legend. An additional control showing the levels of HSPA5 in animals that do not have the plasmid transfected are required to show that we aren't just looking at endogenous HSPA5.
 - o Stability and ubiquitination of HSPA5 327Q needs to be demonstrated by IP and western blotting (6G/S7G).
 - Figure S7
 - o There appears to be a slight increase in the acetylation of WT-HSPA5 in Sept2^{-/-} cells upon infection and with SCNN1b knockdown. Do the authors think this is an artifact? This would imply there might be other activators of ATAT1.

Discussion

- Some of this discussion is copied almost word for word from the introduction.
- SEPT2 is not well characterised, although there is some literature around its roles in the cytoskeleton (doi: 10.1101/gad.11.12.1535) and mitochondrial fission (doi: <https://doi.org/10.15252/embr.201541612>). The authors should consider discussing these other facets of SEPT2 biology and the possibility of these biological processes influencing the regulation of HSPA5.

Reviewer #2 (Remarks to the Author):

This is a comprehensive manuscript that investigates the role of the SEPT2 in antiviral activity by controlling IFN- γ -independent M1 polarization. The authors showed that SEPT2, which induces by ER stress, balances the competition between acetylation and ubiquitination of HSPA5 at position Lysine 327, thereby alleviating ER stress and constraining M1-like polarization and proinflammatory cytokine release. The study is interesting and my concerns are listed below.

1. Almost all infections can cause inflammatory response, such as virus and bacteria. The authors should confirm whether SEPT2 could play a similar role in the inflammatory response caused by bacteria.

2. In line 111, the authors said that SEPT2 participates in antiviral activity by controlling IFN- γ -independent M1 polarization. Please investigate whether SEPT2 can affect the production of interferon after RNA or DNA virus infection.

3. In Fig. 1D, the authors demonstrated that macrophages were increased in Sept2fl/fl Lyz2-Cre mice, and the result also showed that M1-like macrophage cluster were increased, but M2-like macrophage cluster were no difference. These results seemed to indicate that the number of macrophage had increased in Sept2fl/fl Lyz2-Cre mice. Please confirm that whether SEPT2 can affect the development of macrophage.

4. ER stress can induce apoptosis. The author had better confirm whether SEPT2 can affect the apoptosis of macrophages.

5. In Fig. 2B and 3D and so on, add the protein band of SEPT2 please, which used to confirm KO efficiency. The similar concern is involved in Fig. 4F and 7F.

6. In Fig. 2G and H, lack of control group, the author should add extra Sept2fl/fl group with knock-down related genes or not. The similar concern is involved in Fig. 6C-E.

7. In Fig. S4A and S4G, VSV infection could reduce HSPA5 protein in Sept2fl/fl Lyz2-Cre macrophages, why is there no difference in Fig. S4E after VSV infection? Similar confusion in Fig. 6B, why is there no difference of HSPA5 protein level between Sept2fl/fl and Sept2fl/fl Lyz2-Cre group after transfected with wt-HSPA5?

8. In Fig. 4D, the result showed that ATAT1 could bind to HSPA5 only in the presence of SEPT2. However, the Fig. 4E and Fig.5E, F showed that ATAT1 acetylated and interacted HSPA5 directly. These results are contradictory.

9. SEPT2 balances the competition between acetylation (by ATAT1) and ubiquitination (by SCNN1B) of HSPA5 at the same position of Lysine 327. It seems like a dynamic and complex regulatory process. The authors had better explore whether these proteins (SEPT2, ATAT1, SCNN1B and HSPA5) could form a large protein complex by immunoprecipitation assay.

10. The authors demonstrated that SEPT2 played an important role in antiviral activity by controlling IFN- γ -independent M1 polarization. Does SEPT2 have a similar effect if the IFN- γ is presented? Please validate or discuss this.

Reviewer #3 (Remarks to the Author):

There have been few studies on IFN γ independent macrophage activation. The role of septins in inflammation or viral infection is mostly unknown. Here the authors perform a tour de force, highlighting a new role for SEPT2 in antiviral activity (by controlling IFN γ -independent M1 polarization). The authors describe SEPT2 regulation of both VSV and HSV1; very little is known about septin control of viral infections. They show that ER stress induces SEPT2 expression,

balancing acetylation vs ubiquitination of HSPA5 (at position lysine 327). If disrupted, unfolded proteins accumulate and accelerate M1-like polarization, leading to excessive inflammation and tissue damage. Together, authors conclude that SEPT2 can negatively regulate IFN γ independent macrophage proinflammatory activation, providing clinical insights into control of excessive inflammation.

Despite the huge amount of work, I have some comments to help strengthen the manuscript message:

1. The septin cytoskeleton is widely viewed to act as hetero-oligomeric complexes with other septin family members. Have the authors other looked at other septins? Please highlight if / when other septins are involved. Or do the authors think SEPT2 is performing this role independently from other septins.

2. What is the precise role of SEPT2 / the septin cytoskeleton? Septins have long been considered to act as a scaffolding protein. The authors suggest that SEPT2 expression balances acetylation vs ubiquitination of HSPA5, but no microscopy / cell biology studies are performed to visualise this. Is SEPT2 interacting with the ER? Is this role specific to macrophages? Does SEPT2 perform this function as a filament or higher order structure?

3. Considering their crucial role in cellular physiology, it seems unlikely that septin expression would change +/- infection. As far as I know, significant changes in septin expression have not been described in the literature (except in the case of cancer studies)? Are these changes in septin expression described here at the transcriptional or protein level? What triggers the change in septin expression may be outside the scope of this manuscript. Quantify changes in expression as measured by immunoblot. Also confirm this change in SEPT2 expression by looking at other septins. If septin expression is significantly changing, how does this impact the other roles of septins (which are ubiquitously expressed) such as cell division, mitochondrial fission, etc.

4. For all figures, representative images are often shown. Where relevant please highlight the number of times an experiment was performed, quantify data across replicates and perform statistics. If not possible to quantify data across replicates, please provide replicate images as supplemental files so that readers can see all the data.

Responses to comments from Reviewer #1:

Content summary with my opinion of the major findings:

The manuscript presented by Beibei Fu and colleagues explores the consequences of Sept2 gene deletion during VSV and HSV infection in animals and in cells. Fu and colleagues reveal Sept2 unexpectedly limits inflammation caused by these infections. Using an array of biochemical and genetic tools, Fu and colleagues propose that Sept2 limits the accumulation of unfolded proteins and ER stress during infection by promoting the acetylation of HSPA5 by ATAT1, which can promote macrophage polarization in the absence of IFN gamma. The authors further suggest that SEPTIN-2 facilitates the acetylation of HSPA5 at lysine K327, which competitively limits K48-mediated ubiquitination at this same site by the E3 ubiquitin ligase SCNN1B. Acetylation of HSPA5 in turn is hypothesized to relieve the ER stress and limit disease-causing inflammation. In myeloid-deleted Sept2 knockout animals, acetylation of HSPA5 at K327 is reduced and is instead ubiquitinated at this site, causing its proteasomal degradation. The accumulation of unfolded proteins triggers the UPR and drives disease-causing inflammation.

RESPONSE: Thank you very much for sparing the time and energy to provide us with thoughtful suggestions on our paper. We feel that the reviewer's pertinent comments have helped us to greatly improve our manuscript. In our revised manuscript, we have added the missing details in Figure Legends and Methods. Additionally, we have supplemented the controls of *in vivo* data and immunoprecipitations. Below, we provide detailed responses to each of the reviewer's comments. We hope that we have addressed each point to the reviewer's satisfaction.

Comments: While the findings are of potential interest, the (sheer) number of experiments presented often lack important controls and there are a number of issues that require attention:

1. Lack of detail. In many places the figure legend or associated methods section lacks sufficient detail to know how an experiment has been performed or how many times each experiment was repeated, and results verified (which should be clearly stipulated in the figure legends).

RESPONSE: We thank the reviewer for pointing this out and agree that it is very important to include controls and details. We have now included details describing the experimental conditions, number of replicates, and verified results in the Figure Legends. In addition, based on your comments listed below, we have supplemented substantial content in the Methods section, such as cell culture conditions, mouse background, detailed experimental procedures and reagent dosage. We believe that this revision will improve the reproducibility of the study.

2. Lack of controls. Some *in vivo* data and most immunoprecipitations lack suitable controls that prevent conclusions that can be drawn from these experiments. Furthermore, molecular weight markers are missing from almost all western blots,

which makes it problematic to assess the bands shown.

RESPONSE: We thank the reviewer for this comment. We have included appropriate controls for the *in vivo* data and immunoprecipitations in the revised manuscript. All western blots have been labeled with molecular weight markers to identify the bands.

3. Confusing logic. In some places the logic is difficult to follow. These are noted below and require some clarification/rephrasing to improve the flow of the manuscript.

RESPONSE: Based on the reviewer's comments, we have improved some key experiments and rephrased some of the descriptions to enhance the logic. We believe that this modification will increase the comprehensibility of our manuscript.

4. Poor analysis of ubiquitination. As noted particularly in my comments for Figure 3, much of the analysis performed examining ubiquitination is not sufficient. For example, performing a co-IP and probing for ubiquitin does not mean that the protein you pulled down is ubiquitinated. The ubiquitin could be on other interacting proteins. Alternative methods (some I have noted below) should be employed to properly examine the ubiquitination of HSPA5.

RESPONSE: We agree with the reviewer that results obtained using TUBE analysis in this study should be more reliable. Therefore, we have repeated ubiquitination analyses using the TUBE method as suggested by the reviewer. The details of TUBE analysis have been included in the Supplementary Information on Page 7, Line 147-154.

5. I did not see any measure of viral titres throughout the manuscript. The authors need to determine if SEPT2 loss might just allow for increased viral replication, and hence burdens (in vitro and in vivo). This is important as it might offer an alternative explanation to the model put forward by the authors.

RESPONSE: We thank the reviewer for this comment and therefore we have included experiments to test whether SEPT2 loss affects viral replication. Peritoneal macrophages (PMs) derived from *Sept2^{fl/fl} Lyz2-Cre* and *Sept2^{fl/fl}* mice were infected with VSV at an MOI of 1 and viral titres were measured by plaque assay at 6 and 12 hours post infection. We also determined the viral burden in lung tissues obtained from mice infected with 1×10^7 PFU VSV. The results showed that SEPT2 deficiency did not affect viral replication *in vitro* and *in vivo*. The corresponding revisions are in Supplementary Fig. 1f, g and on Page 8, Line 170-173.

6. All *In vivo* transfection efficiency data needs to be shown.

RESPONSE: We have provided immunoblots to show the efficiency of all *in vivo* transfection models. The corresponding revisions are in Fig. 6b and Supplementary

Fig. 11b.

Detailed comments:

Introduction

7. Macrophage biology is more complex than the simple M1/M2 paradigm, particularly in vivo (see doi: 10.12703/P6-13, doi: 10.1161/CIRCRESAHA.116.309194, doi: 10.3389/fimmu.2015.00370.). To keep relevant and up to date with the field, the authors must frame their work with the current view of macrophage biology, either or both in the introduction and discussion.

RESPONSE: We appreciate the reviewer for this suggestion. We have read the above references carefully and introduced them in the revised manuscript. The corresponding revision is on Page 2-3, Line 42-56. Both the introduction and discussion have been framed with the latest macrophage biology perspectives in the revised manuscript. We hope that these improvements will increase the readability of the paper.

8. Lines 61 – 63: During an infection, the TLR activator comes first, followed by IFN γ . By discussing M1 polarisation as firstly being primed with IFN γ , then being exposed to a TLR activator, the authors present an oxymoron that is inconsistent with what happens in vivo. The authors should clarify this statement, or instead talk about more specifically about why IFN γ is important for macrophage biology (see doi: 10.1189/jlb.0603252).

RESPONSE: The reviewer is correct here; we have therefore omitted this sentence in the revised manuscript. Instead, we have included the functional role of IFN- γ and strengthened the importance of IFN- γ in macrophage biology as suggested by the reviewer. The corresponding revision is on Page 4, Line 72-79.

9. In the introduction, the authors set up a framework for M1 polarisation by stating that IFN γ + LPS can fully activate the M1 phenotype, as can IFN γ -independent activation with T-cell-derived GM-CSF and during macrophage activation syndrome in IFN γ knockout cells. However, the authors specify that LPS treatment alone does not fully activate the M1 phenotype. Throughout their manuscript the authors have not measured the response of their IFN γ -independent macrophage activation against complete M1 polarisation either in the presence of IFN γ or an 'incomplete'. It is unclear if this activation is complete, or a partial response as one might see with LPS stimulation alone. I recommend the authors re-phrase their introduction by removing the discussion on endotoxin tolerance effects in lines 65-66. This paragraph may benefit from additional detail on the IFN γ -independent activation of macrophages and highlighting what is known and what is not known.

RESPONSE: We thank the reviewer for this suggestion and have therefore removed the discussion on endotoxin tolerance effects in the revised manuscript. Also, we have

added additional details and background on IFN- γ -independent macrophage activation in accordance with the reviewer's suggestion. The corresponding revision is on Page 4-5, Line 87-103.

10. In line 111, please replace the word 'controlling' with 'promoting' or another appropriate term that indicates the direction of control.

RESPONSE: The word has been replaced as suggested.

Methods

11. The methods section lacks detail so much so that this work would not be able to be accurately repeated by independent researchers. Please improve the level of detail in all current and future methods sections to sufficient detail that other researchers could repeat the experiments in this manuscript.

RESPONSE: We thank the reviewer for this comment, and have supplemented substantial details to the Methods section based on the reviewer's suggestions. We hope that these revisions will improve the reproducibility of our experiments.

Mice and IFN- γ signaling blockade

12. Line 651, replace 'hybridized' with 'inter-crossed'.

RESPONSE: The word has been replaced as suggested.

13. Please provide more detail regarding the animal experiments performed including but not limited to: sex of animals used for cohorts, number of cohorts, age of animals at beginning of experiment, genetic background of animals.

RESPONSE: In this study, we generated three transgenic mice, *Sept2^{fl/fl} Lyz2-Cre*, *Sept2^{fl/fl} Lyz2-Cre-ERT2* and HSPA5^{K327Q} site-directed mutated mice. Their genetic background and construction methods have been provided in the Methods section. For animal experiments, six- to eight-week-old mice were randomly assigned to each group. Mice of both sexes (equal distribution) were used and all mice in different groups were age- and sex-matched. The number of cohorts in each experiment has been indicated in the Figure Legends. The above details have been added to the Methods section (Page 35-36, Line 760-786) and the Reporting summary.

Cell lines and viruses

14. This section lacks detail on the generation of the iBMDMs (referring to a previous publication is fine but please provide some detail).

RESPONSE: We thank the reviewer for this suggestion. The iBMDM cell line used in this study was a gift from Dr. Feng Shao (National Institute of Biological Sciences, Beijing, China). We have also included details about the generation of iBMDMs in the

revised manuscript. The corresponding revision is on Page 36-37, Line 790-795.

15. Please provide detail on the culturing conditions of all cell lines used in the study.

RESPONSE: As suggested by the reviewer, we have supplemented the generation methods of all cell lines (iBMDM^{Tet-on SEPT2 shRNA}, *Atat1*^{-/-}iBMDM, *Atat1*^{-/-}*Scnn1b*^{-/-}iBMDM and *Xbp1*^{-/-}iBMDM) in the revised manuscript (Page 37, Line 795-813). For the culturing conditions, iBMDMs, PMs isolated from mice, PBMCs isolated from human blood samples, and TC-1 cells were cultured in RPMI 1640 medium with 10% FBS (Gibco) at 37°C, 5% CO₂. NIH-3T3, L929 and HEK-293FT cells were cultured in DMEM medium with 10% FBS (Gibco) at 37°C, 5% CO₂. The above information has been included in the Methods section of the revised manuscript on Page 38, Line 818-821.

16. Please provide more detail regarding the propagation of virus.

RESPONSE: We thank the reviewer for this suggestion and have provided the source and propagation of all viruses used in this study to the Methods section. The corresponding revision is on Page 38, Line 825-836.

17. Please provide more detail regarding the animal experiments performed including but not limited to: sex of animals used for cohorts, number of cohorts, age of animals at beginning of experiment, genetic background of animals, vehicle of virus, control injections performed (if any).

RESPONSE: We have included detailed information regarding the sex and age of mice, the vehicle of virus, the setting of control group, the dose and duration of IFN- γ blockade and the principle of the endpoint determination in animal experiments on Page 39-40, Line 854-864 of the revised manuscript. The number of cohorts and the genetic background of mice have been provided in the Figure Legends, as well as the Methods section on Page 35-36, Line 760-786.

18. Please provide more detail on the density of cells for in vitro experiments.

RESPONSE: Cells were plated at a density of 1×10^6 cells/60 mm plastic dish for viral infection. The details of *in vitro* infection experiments have been supplemented to the Methods section (Page 40, Line 864-867). The procedures of other *in vitro* experiments have been added to the Supplementary Information.

Blood samples

19. Note the experimental groups are partially biased toward male participants. The controls are more balanced. The authors may consider some form of analysis to test if sex contributes to their differences. Please provide a rationale why sex would not

contribute if analysis is not performed.

RESPONSE: We thank the reviewer for this suggestion. Due to the limitations of the clinical sample source, a larger proportion of male participants were included in the experimental group. Following the reviewer's suggestion and the journal guidance, we have included additional analysis to test whether sex contributes to differences in SEPT2 expression, and the results showed no significant difference in SEPT2 expression levels between males and females. Therefore, we hypothesized that sex may not affect the SEPT2-regulated inflammatory responses. The corresponding revision is in Supplementary Fig. 12m and Page 29, Line 632-635.

Statistical Analysis

20. Please provide more information on the specific statistical analysis applied to each type of data.

RESPONSE: In accordance with the reviewer's suggestion, detailed statistical analysis of each figure has been provided in the Figure Legends. In addition, we have included some information to the "Statistical Analysis" section. The corresponding revision is on Page 40, Line 869-877.

Plasmid and siRNA transfection

21. Please specify specific detail on how the transfections were performed. How much DNA/siRNA, how long?

RESPONSE: Cell transfection was performed using a LONZA 4D-Nucleofector system. Plasmids (1 μg) or siRNAs (40 pmol) were transfected into 1×10^6 cells by electroporation. The time constant value was 5 ms. The knockdown efficiency was determined at least 24 hours after transfection. The *in vivo* transfection was performed by intravenous injection using an *in vivo*-jetPEI delivery reagent. Nucleic acids (40 μg) were transfected into mice every 3 days during the experiments. The corresponding revision is in the Supplementary Information on Page 2, Line 28-33.

22. This section also only specifies *in vivo* transfection of nucleic acid constructs. Please also specify the method used *in vitro*.

RESPONSE: The details of *in vitro* transfection have been provided in the Supplementary Information on Page 2, Line 26-33: Cell transfection was performed using a LONZA 4D-Nucleofector system according to the manufacturer's instruction. When cell confluency reached 75%, 1×10^6 cells were collected and mixed with 1 μg plasmids or 40 pmol siRNA. The cell-nucleic acid mixture was added into Nucleofector Solution with Supplement and transferred into Nucleocuvette Vessel. The time constant was set to 5 ms. After electroporation, cells were transferred to the plastic dish for further culture. The transfection efficiency was detected after 12 hours (overexpressing plasmid) or 24 hours (siRNA).

High content screening

23. Please provide information on the statistical methods applied to this analysis.

RESPONSE: Statistical analysis was performed by CellReporterXpress automated imaging analysis software (Molecular Devices). The mean GFP fluorescence intensity was calculated based on cell area and GFP fluorescence intensity. To obtain reliable statistical results, one image of the cells in each well was taken under 20× objective lens and three images were taken under 40× objective lens for quantitative analysis. This information has been provided in the Supplementary Information on Page 3, Line 49-55.

Histopathology analysis

24. Please outline the pathology scoring parameters, whether the scoring was performed blinded and how many measures per animal were taken to obtain a final value.

RESPONSE: We have included the scoring system of pathology in the Supplementary Information (Page 3-4, Line 63-67). The scoring was performed blinded. Six mice from each group were randomly selected for histopathology analysis. One representative slice obtained from each mouse was selected for scoring. The final score for each group depended on the average score of six mice.

qRT-PCR

25. Please provide the specific PCR protocol used.

RESPONSE: The detailed PCR protocol has been provided in the Supplementary Information on Page 5, Line 98-106: Purified RNA (1 µg) was reverse-transcribed to cDNA in 20 µL reaction using a SYBR PrimeScript RT-PCR Kit (Takara, Otsu, Shiga, Japan). qRT-PCR was performed using a TB Green Premix ExTaq II Kit (Takara) on Bio-Rad CFX-96 system (Bio-Rad, Hercules, CA, USA). cDNA (2 µL) was amplified in 25 µL reaction containing 12.5 µL TB Green Premix ExTaq II and 0.4 µM forward/reverse primers. The cycling conditions were as follows: predenaturation (95°C, 30 s), denaturation (95°C, 5 s, 39 cycles) and extension (60°C, 30 s). Results were normalized to GAPDH mRNA levels according to the $2^{-\Delta\Delta C_t}$ method.

Western blotting and coimmunoprecipitation

26. Please provide detail on the lysis buffer used for standard cell lysis.

RESPONSE: Cell lysis buffer [20 mM Tris (pH 7.5), 150 mM NaCl, 1% Triton X-100, sodium pyrophosphate, β-glycerophosphate, EDTA, Na₃VO₄ and leupeptin] (P0013, Beyotime) containing protease inhibitor cocktail was used for standard cell lysis. We have added this information to the Supplementary Information on Page 6, Line 113-115.

27. Please provide dilution of antibodies used and into what solution (e.g. BSA, skim milk)

RESPONSE: Phosphorylation-specific antibodies were diluted in 1% BSA. Antibodies other than phosphorylation-specific antibodies used for western blotting and immunoprecipitation were diluted in 5% skim milk. The detailed dilution of each antibody has been provided in the Supplementary Information on Page 6-7, Line 118-140.

Enzyme linked immunosorbent assay

28. Please specify whether samples were diluted and/or whether all sample values fell within the standard curve of the assay.

RESPONSE: BALF samples obtained from virus-infected mice were diluted 10-fold with Assay Diluent A (contains 0.09% sodium azide, provided by the ELISA Kit). The dilution ratio of cell culture supernatant was determined according to the empirical data from pilot experiments. All diluted samples fell within the standard curve. We have provided the above information in the Supplementary Information on Page 8, Line 156-164.

RNA-sequencing

29. Please provide GEO ID.

RESPONSE: The GEO ID has been provided in the “Data Availability” (Page 40-41, Line 880-881) and the Supplementary Information on Page 9, Line 185-186.

30. Please provide citations for the software used to analyse this data.

RESPONSE: Differential expression analysis and functional enrichment were performed using DESeq2 software on the Majorbio Cloud Platform. The filtering threshold is $|\log_2FC| \geq 1$ & $p_{adj} < 0.01$. We have included the reference (Love et al., 2014) for DESeq2 software in the Supplementary Information on Page 9, Line 183-184.

Reference:

Love MI, Huber W, Anders S. Moderated estimation of fold change and dispersion for RNA-seq data with DESeq2. *Genome Biol.* 2014;15(12):550.

31. To demonstrate quality of the RNA-sequencing data, please show a MDS plot to show clustering of the data generated in the RNA-sequencing analysis in a main or supplemental figure.

RESPONSE: We thank the reviewer for this suggestion. We have included a principal component analysis (PCA) to show the clustering of the RNA-seq data, and the

corresponding revision is in Supplementary Fig. 5a.

Transmission electron microscopy

32. Please explain 'standard procedures'

RESPONSE: The standard procedures for preparation of the TEM samples are as follows: Cells were fixed with 2.5% glutaraldehyde and 1% OsO₄, followed by gradient dehydration with ethanol at 4°C. After embedding and solidification, the samples were cut into 70 nm slices by an ultra-thin slicer and dyed with 3% uranium acetate-lead citrate. Finally, imaging was performed with a JEOL JEM-1400 plus transmission electron microscope. This information has been provided in the Supplementary Information on Page 9, Line 190-196.

33. Please provide detail on the magnification and scale bar length used for the data presented in figure S3.

RESPONSE: Data presented in Supplementary Fig. 5c (Fig. S3B of the original manuscript) were obtained by JEOL JEM-1400 plus transmission electron microscope (magnification: ×12.0k, voltage: 80.0 kV). Scale bar = 5 μm. These details have been added to the Supplementary Information (Page 9, Line 194-196) and the legend of Supplementary Fig. 5c.

Detection of the accumulation of unfolded proteins

34. Please provide detail on how these experiments were performed including concentrations of NTPAN-MI.

RESPONSE: The NTPAN-MI staining was performed as previously described (Owyong et al., 2020). NTPAN-MI was dissolved in DMSO as 2 mM stocks. Cells were treated with 50 μM NTPAN-MI for 30 min at room temperature. Prior to imaging, cells were stained with DAPI and fixed with 4% PFA for 20 min. Images were obtained on a Leica DMi8 inverted fluorescence microscope. The excitation wavelength and emission wavelength were as follows: NTPAN-MI (excitation: 405 nm; emission: 527 nm), DAPI (excitation: 350 nm; emission: 460 nm). We have supplemented these contents to the Supplementary Information on Page 10, Line 204-211.

Reference:

Owyong TC, Subedi P, Deng J, Hinde E, Paxman JJ, White JM, et al. A Molecular Chameleon for Mapping Subcellular Polarity in an Unfolded Proteome Environment. *Angewandte Chemie*. 2020;59(25):10129-35.

Dual-luciferase reporter assay

35. Please provide detail on the transfection conditions (e.g. confluency), amount of

DNA used.

RESPONSE: HEK-293FT cells were seeded on 24-well plates and cultured overnight to reach 75% confluency. The luciferase reporter plasmids (300 ng) and internal control plasmid pRL-SV40 (10 ng) were cotransfected into cells for 24 hours. The corresponding revision is in the Supplementary Information on Page 14, Line 292-294.

Chromatin immunoprecipitation assay

36. Please specify sonication settings and micrococcal nuclease digestion conditions.

RESPONSE: ChIP assay was performed using a SimpleChIP Plus Enzymatic Chromatin IP Kit (9004S) according to the manufacturer's instruction. Pellet nuclei was collected and mixed with micrococcal nuclease (0.5 μ L per 4×10^6 cells). The lysate was incubated at 37°C for 20 min with frequent mixing to digest chromatin to a length of approximately 150-900 bp. Then lysate was subjected to sonication to break nuclear membrane. After digestion and sonication, lysate was used for immunoprecipitation with anti-sXBP1 antibody. We have included the detailed procedures in the Supplementary Information on Page 14, Line 303-308.

Results

37. Please provide molecular weight markers on all probes in all western blots. Please also ensure a loading control is provided in all western blots and that in the immunoprecipitations, you 1) probe for the protein you are pulling down with in the IP samples to ensure the IP is equal among samples, and 2) in the lysate to ensure there was equal starting amounts of protein in the input. This probe will also help explain whether any differences in the IP probe are a consequence of altered expression.

RESPONSE: We thank the reviewer for this suggestion. We have included the molecular weight markers and controls in the revised manuscript according to the reviewer's requirements. We believe that these revisions will improve the readability of the manuscript.

38. The authors conclude that SEPT2 loss does not alter oxidative stress, which might explain the increased unfolded proteins and inflammation. However, their own data show dramatically increased iNOS levels in VSV infected SEPT2 knockout cells, and consequent increased nitric oxide generation is a known cause of oxidative stress. The authors should address whether iNOS/NO blockade reverses the UPR and heightened inflammatory responses.

RESPONSE: We appreciate this insightful comment and have taken it into consideration in our revised manuscript. We believe that this confusion may have been caused by the detecting probes. In the original manuscript, Dihydroethidium and Hydrogen Peroxide Assay Kit were used for the measurement of $\cdot\text{O}_2^-$ and H_2O_2 ,

respectively. However, neither of them is suitable for the detection of oxidative stress caused by NO and ONOO⁻. In the revised manuscript, we used the H₂DCFDA probe (D399, Invitrogen, an indicator of H₂O₂, ONOO⁻ and ROO[•]) to detect oxidative stress in PMs following VSV infection. The results showed that SEPT2-deficient cells indeed produced higher level of reactive oxygen species (Supplementary Fig. 6b). We have modified the related content on Page 14, Line 304-306, and added experimental details to the Supplementary Information on Page 10, Line 213-220.

In addition, we are interested in the question raised by the reviewer regarding the potential role of NO-induced oxidative stress on UPR and inflammation. In this context, an iNOS specific inhibitor, 1400W dihydrochloride (100 μM), was used to treat *Sept2^{fl/fl}* and *Sept2^{fl/fl} Lyz2-Cre* PMs during VSV infection. The results showed no significant difference in oxidative stress between WT and SEPT2-deficient PMs under iNOS-blocking condition (Supplementary Fig. 6c, d), and iNOS blockade failed to attenuate the excessive UPR and proinflammatory factor secretion in SEPT2-deficient cells (Supplementary Fig. 6e, f). Therefore, we hypothesize that oxidative stress is not the primary cause of SEPT2-regulated high level of ER stress. We have included this result in the revised manuscript on Page 14-15, Line 306-313.

SEPT2 is involved in IFN-γ-independent macrophage proinflammatory activation 39. Please provide some form of results from the HCS that show SEPT2 was a hit in this experiment.

RESPONSE: In accordance with the reviewer's suggestion, we have included the results showing that SEPT2 knockdown significantly increased GFP intensity in both VSV and HSV-1 infected groups. The corresponding revision is in Supplementary Fig. 1a.

40. SEPT2 knockout mice do not appear to have been previously described and it would be useful to know if deletion (at least in the myeloid compartment) causes any defect in vivo. Please mention whether these animals are phenotypically normal.

RESPONSE: We thank the reviewer for this suggestion, which was also raised by Reviewer #3. We have measured the body weights of six- to eight-week-old *Sept2^{fl/fl} Lyz2-Cre* mice (without infection) and no significant difference was observed when compared to their littermates (Supplementary Fig. 1b), and there was no significant difference in the numbers of macrophages isolated from spleen and lung (Supplementary Fig. 1c, d). These results indicated that *Sept2^{fl/fl} Lyz2-Cre* mice were phenotypically normal. The corresponding revision is on Page 8, Line 156-160.

41. Sentence starting at line 139 does not make sense. Are the authors referring to previous studies that have used anti-IFNγ to block signalling? If so please provide citations.

RESPONSE: Yes, IFN-γ blockade in mouse models was performed as previously

described (Bailey et al., 2022). We have modified this sentence and added the citation on Page 8, Line 160-162.

Reference:

Bailey SR, Vatsa S, Larson RC, Bouffard AA, Scarfò I, Kann MC, Berger TR, Leick MB, Wehrli M, Schmidts A, Silva H, Lindell KA, Demato A, Gallagher KME, Frigault MJ, Maus MV. Blockade or Deletion of IFN γ Reduces Macrophage Activation without Compromising CAR T-cell Function in Hematologic Malignancies. *Blood Cancer Discov.* 2022 Mar 1;3(2):136-153.

42. Ideally, the Sept2^{fl/fl}/Lys2-Cre animals would be crossed onto an IFN-g knockout. This would definitively show that the response is interferon gamma independent. However, I understand this would take too long for a revision. At a minimum, the authors should perform this infection experiment in the presence and absence of IFN γ to show that the IFN γ inhibitory antibodies are working and to test whether the function of SEPT2 is unveiled only in the absence of IFN γ signaling. Once results from this experiment are obtained, the authors should adjust their conclusions accordingly.

RESPONSE: We thank the reviewer for this valuable and considerate suggestion. We have supplemented the infection experiments in the presence or absence of IFN- γ (with or without injection of IFN- γ antibody, Supplementary Fig. 4b-d). The validation of IFN- γ blockade was verified by detecting the release of IFN- γ in BALF (Supplementary Fig. 1e). The results indicated that SEPT2 could modulate M1-like polarization and inflammation in the absence of IFN- γ . In this context, we showed that SEPT2-mediated UPR was independent of the IFN- γ pathway. The corresponding revisions are on Page 8, Line 162-164, and Page 11, Line 234-241.

43. Please rephrase the sentence beginning in line 176. The first clause of this sentence makes it sound like the authors contest this well-known fact. I suggest removing from lines 176 up to the citation and beginning the sentence with ‘viral nucleic acids...’.

RESPONSE: This sentence has been modified as suggested.

44. Line 187: ‘probably’ is colloquial and should be replaced with ‘likely’.

RESPONSE: This sentence has been modified as suggested.

45. The Sept2^{fl/fl} animals do not start to die until around day 6, but the analysis performed in the last four panels F – I are done in <24 hours. Are these changes informative to how the disease progresses more than 5 days after the ex-vivo analysis is performed?

RESPONSE: We thank the reviewer for this question on timing selection. The regulation of innate immunity is mainly reflected in the early stage of infection, while adaptive immunity will play its main function in later stages. Therefore, the timing of 5-6 days post-infection may be a bit late to evaluate the role of SEPT2 in this study. According to a previous study, samples obtained from early infection (when mice have not started to die) were used to assess the innate immune responses (Lian et al., 2018). Therefore, we infected mouse PMs with viruses *in vitro* for 0-12 hours in this study to evaluate the immunomodulatory function of SEPT2 in the early innate immunity.

Reference:

Lian H, Zang R, Wei J, Ye W, Hu MM, Chen YD, Zhang XN, Guo Y, Lei CQ, Yang Q, Luo WW, Li S, Shu HB. The Zinc-Finger Protein ZCCHC3 Binds RNA and Facilitates Viral RNA Sensing and Activation of the RIG-I-like Receptors. *Immunity*. 2018 Sep 18;49(3):438-448.e5.

Figure 1

46. The pathology scoring from the supplemental could be moved in the main figure for clarity.

RESPONSE: We thank the reviewer for this suggestion and the pathology scoring has been moved to the main figure (Fig. 1c, 7c).

47. The pathology difference in the Sept2 knockout animals does look mildly more severe, as reflected in the pathology score. Some clarification on how endpoint was determined for the survival curve would help assess how much the pathology informs on the survival of the animals.

RESPONSE: We appreciate this suggestion. Mice that reached the humanitarian endpoint (>15% weight loss) or completed the observation period were euthanized by cervical dislocation. We have provided this information in the Methods section on Page 40, Line 862-864.

48. Please detail the scale bar lengths in the figure legend.

RESPONSE: In accordance with the reviewer's suggestion, we have included the scale bar lengths in the Figure Legends.

49. Please provide a gating strategy in the supplemental figures for the cell subsets presented in panel D.

RESPONSE: The gating strategy has been provided in Supplementary Fig. 1h.

50. Related to panel C, Is there a change in basal cytokine production in the knockouts?

RESPONSE: Through statistical analysis, we found no significant difference in basal cytokine production in the knockout mice. We have added “n.s.” marker in Fig. 1d of the revised manuscript.

51. Panels F and H: Please provide some rationale as to why these timepoints were selected when the model proceeds over 15 days.

RESPONSE: We thank the reviewer for this suggestion, which was also raised in Comment #45. The regulation of innate immunity is mainly reflected in the early stage of infection, while the adaptive immunity will play its main function in later stages. Therefore, the timing of 5-6 days post-infection may be a bit late to evaluate the role of SEPT2 in this study. In this context, we infected mouse PMs with viruses *in vitro* for 0-12 hours in this study to evaluate the immunomodulatory function of SEPT2 in the early innate immunity. Please see also response to Comment #45.

Figure S1

52. Figure legend title could be qualified with “nucleic acid sensing” before PRRs.

RESPONSE: The figure legend title has been modified as suggested.

53. Please specify what the qPCR data is normalised to.

RESPONSE: The qRT-PCR data was normalized to GAPDH expression. We have included this information in the Figure Legends.

54. Please perform a statistical analysis should be performed on the data generated in S1E – O.

RESPONSE: We thank the reviewer for this comment. Considering it with Comment #55, we have included the results of three biological replicates in the form of histogram and performed statistical analysis in Supplementary Fig. 2n-s, v, w of the revised manuscript.

55. The authors make a big conclusion that the effect of SEPT2 is independent of nucleic acid sensing PRRs, however the data presented is only representative of one independent repeat. The authors may keep the flow cytometry plots if they wish, but this data would be better presented in the form of a bar graph to compare the infection with no siRNA vs with the various siRNAs and to show all three independent replicates in one graph. The effect of the siRNAs is also hard to gauge, because the control (no siRNA) is in the main figure. The authors should show no-siRNA vs +siRNA upon infection in the one figure, preferably in a bar graph but also in a flow cytometry plot. Lastly, authors should show the efficiency of their siRNAs using western blot, especially if they claim that the siRNA has no effect on the phenotype.

RESPONSE: We thank the reviewer for these suggestions and agree that it is important to include three replicates in one graph. We now include the results of three biological replicates in the form of histogram and perform statistical analysis in the revised manuscript. The no-siRNA (NC siRNA) controls have also been supplemented. The efficiency of RNA interference has been assessed by western blotting. The corresponding revisions are in Supplementary Fig. 2h-w of the revised manuscript.

SEPT2 deficiency regulates M1-like hyperpolarization through ER stress

56. IL1b secretion is generally accompanied with pyroptotic cell death. Does infection cause cell death? This is an important point as cell death, or deficiency in cell death, has a big impact on viral infections, yet the authors do not determine if SEPT2 loss impacts viral-induced killing of the host cells.

RESPONSE: We acknowledge the reviewer for this valuable comment. Considering it together with Reviewer #2's Comment #4, we have added experiments to detect pyroptosis and apoptosis in VSV-infected *Sept2*^{fl/fl} *Lyz2*-Cre and *Sept2*^{fl/fl} PMs. The results showed that there was no significant difference in cell viability between WT

and SEPT2-deficient cells at 0-24 hours post VSV infection (Supplementary Fig. 3c). Further, we found that there was no significant difference in pyroptosis and apoptosis at 12 hours post-infection (Supplementary Fig. 3d, e). Since the duration of *in vitro* infection in this study did not exceed 12 hours, these data suggested that the excessive secretion of IL-1 β may not be caused by pyroptosis. According to previous reports, IL-1 β can be released through secretory lysosome, secretory autophagy, exosomes and microvesicle shedding (Piccioli and Rubartelli, 2013). We hypothesize that the secretion of IL-1 β in the early stage of infection may be associated with these pathways. We have supplemented the above data to the revised manuscript on Page 10-11, Line 220-227.

Reference:

Piccioli P, Rubartelli A. The secretion of IL-1 β and options for release. *Semin Immunol.* 2013 Dec 15;25(6):425-9.

57. Line 234 - 236: Please specify the pathways you are looking at.

RESPONSE: The pathway names (IRE1-TRAF6-IKK, JNK-AP1 and NF- κ B) have been specified on Page 13, Line 276-278.

58. In lines 233 – 237, the authors assume that the increased activation observed in figure S3D leads to increased ER stress; however, the data in figure S3D creates a chicken and egg scenario. These pro-inflammatory pathways can be activated by TNF. So, does the activation of these pathways lead to ER stress (as hypothesised by the authors), or does the heightened TNF in the Sept2 knockouts (Figure 1H/I, 2F/H) cause more activation of this pathway? Conclusions drawn in the main text should be adjusted to take this possibility into account.

RESPONSE: We thank the reviewer for this interesting question. Indeed, currently we cannot rule out the possibility that elevated TNF- α in SEPT2 knockout cells leads to more activation of these pathways. Therefore, and in consideration of the reviewer's suggestion, we have adjusted the conclusion to "the overactivation of these pathways may arise from the excessive ER stress caused by SEPT2 deletion, and the downstream product TNF- α can also trigger these proinflammatory pathways, further accelerating their activation". The corresponding revision is on Page 13, Line 280-283.

Figure 2

59. 2B – The loading control in lanes 5 and 6 do appear mildly increased and may explain the differences seen in ATF4, sXBP1, cleaved/full length ATF6a and pPERK. pIRE1a and pelf2a appear the most convincing. I would suggest repeating this WB to obtain a more consistent loading, or adjust the conclusions based on the clearest data. The blot for full length ATF6a is also over-exposed. Please provide an image with a

lower exposure. Please also provide the MOI for this experiment in the figure legend.

RESPONSE: We thank the reviewer for this suggestion. We have repeated the western blotting for ATF4, sXBP1, full length ATF6 α , p-PERK and GAPDH in Fig. 2b. We believe this revision will provide for a more convincing argument. The MOI of 1 has also been provided in the figure legend as suggested.

60. 2E – the bar (second from left) does not sit at the mean for these samples. Are there additional dots sitting between 30 and 120 that are not shown? please amend graph to rectify this issue. Please also indicate the duration of 4PBA treatment and MOI in the figure legend.

RESPONSE: We thank the reviewer for this suggestion. We have adjusted the Y-axis range to show all the data points (Fig. 2f of the revised manuscript). The duration of 4-PBA (12 hours) and MOI of 1 have been indicated in the figure legend.

61. Please show efficiency of the siRNAs using WB. As a major conclusion of the manuscript, this is an important control to include.

RESPONSE: We thank the reviewer for this important comment. The knockdown efficiency of the siRNAs has been verified using western blotting in the Supplementary Fig. 5h of the revised manuscript.

62. The authors data in figure 2G is surprising. One would presume that if all three are activated, then knocking down one pathway would have little effect. Instead, the authors see that knocking down any pathway has a large effect (on iNOS activity) due to redundancy between the pathways. The authors should employ an additional measure of using WB to examine pEIF2 α to assess this. Notably, it is most surprising that siRNAs against PERK and ATF6 had an effect given the mild difference seen by the WB in figure 2B.

RESPONSE: We thank the reviewer for this insightful comment. We agree with the reviewer that if all three pathways are activated, knocking down one pathway would have little effect. We speculate that the result shown in Fig. 2h (Fig. 2G of the original manuscript) may be specific to the condition of SEPT2 deficiency. To confirm this hypothesis, we included knockdown experiments in wild-type cells. The results showed that suppression of one pathway by siRNA barely affected the total iNOS activity in wild-type cells (Fig. 2h). To further address this question, we performed western blotting to examine the p-eIF2 α level when PERK, IRE1 or ATF6 was knockdown in both *Sept2*^{fl/fl} *Lyz2*-Cre and *Sept2*^{fl/fl} PMs, as suggested by the reviewer. The results showed that suppression of any pathway could inhibit p-eIF2 α level in SEPT2 deletion cells rather than in wild-type cells (Supplementary Fig. 5i). In this context, SEPT2 may exert a potential influence on the redundancy among the pathways; however, this needs to be verified further experimentally.

As for the quality of western blotting in Fig. 2b, we have repeated part of the

experiments as suggested by the reviewer in Comment #59. We apologize for the confusion caused by the quality of western blotting in the original manuscript. We have modified and supplemented the related information to Fig. 2b, h, Supplementary Fig. 5i and on Page 14, Line 292-299.

63. Please show that the 4PBA reduces ER stress and is working as expected.

RESPONSE: The specificity of 4-PBA has been verified in Supplementary Fig. 5g.

Figure S2

64. The authors should generate an iBMDM TET-ON line with a scramble shRNA to show that the effect seen are more specifically a consequence of SEPT2 knock-down, rather than a consequence of shRNA expression.

RESPONSE: We thank the reviewer for this suggestion. An iBMDM Tet-on cell line with a scramble shRNA has been generated accordingly. This control experiment proved that the changes in iNOS activity and proinflammatory cytokines were specifically a consequence of SEPT2 knockdown. The corresponding revision is in Supplementary Fig. 4f-h.

65. Do the animals that get corn oil receive anti-IFN-gamma?

RESPONSE: Yes, mice in the control group also received anti-IFN- γ antibody. We have included this information in the legend of Supplementary Fig. 4a.

Figure S3

66. In S3A, is the p-value the same for each pathway? Please clarify.

RESPONSE: The red bars shown in Fig. S3A of the original manuscript represented the Padjust (p-value corrected by Benjamini-Hochberg method). We have changed Padjust to the original p-values in the revised manuscript (Supplementary Fig. 5b).

67. As mentioned in the methods, please provide some data to show the consistency of each sample submitted for RNA-sequencing.

RESPONSE: According to the Comment #31, we have supplemented PCA to show the consistency of samples. The corresponding revision is in Supplementary Fig. 5a and on Page 12, Line 249-250.

SEPT2 deficiency promotes the proteasomal degradation of HSPA5

68. The rationale for looking at HSPA5 presented in the manuscript is not clear. It makes sense that SEPT2 might regulate the activation of PERK/IRE1/ATF6, but why specifically HSPA5? If the authors can convincingly show PERK, IRE1 and ATF6

are all regulated by SEPT2, then you may make this jump as HSPA5 is reported to regulate them all based on doi: 10.1016/j.gene.2017.03.005. However, upon ER stress, HSPA5 activates the UPR (doi: 10.1146/annurev-cellbio-101011-155826). Since the authors contend that Sept2 knockout cells are effectively deficient in HSPA5 due to increased turnover, how do the authors explain the increased activation of IRE1? The authors are proposing HSPA5 has the opposite function to its known and published role. How do the authors explain this?

RESPONSE: We thank the reviewer for this comment. In the revised manuscript, Fig. 2b, c showed that PERK, IRE1 and ATF6 were all regulated by SEPT2. In Fig. 3a, b, we observed the abnormal degradation of HSPA5 in SEPT2-deficient cells, which was clearly distinct from the expression of other ER stress markers (CHOP and ATF4). Based on this, we believe it is reasonable to select HSPA5 for further study. In addition, we carefully read the references recommended by the reviewer (Wang et al., 2017, Korennykh and Walter, 2012). The authors summarized four inputs which effect the high-order oligomerization of IRE1 (the activated form of IRE1), (a) binding of HSPA5 to the luminal domain, (b) docking of unfolded proteins to the luminal domain, (c) binding of cofactors to the ATP pocket of the kinase domain, and (d) *trans*-autophosphorylation of the kinase domain. In model (a), the unfolded proteins activate IRE1 by binding to HSPA5, and dissociating HSPA5 from the sensor domain of IRE1. In the absence of unfolded proteins, HSPA5 interacts with IRE1 and inhibits its high-order oligomerization. In this case, HSPA5 has an inhibitory rather than an activation effect on IRE1. In our study, the degradation of HSPA5 in SEPT2-deficient cells results in the excessive UPR, which does not conflict with the model proposed by Korennykh and Walter. We hope that the above evidences and explanations meet your requirements.

Reference:

Wang J, Lee J, Liem D, Ping P. HSPA5 Gene encoding Hsp70 chaperone BiP in the endoplasmic reticulum. *Gene*. 2017 Jun 30;618:14-23.

Korennykh A, Walter P. Structural basis of the unfolded protein response. *Annu Rev Cell Dev Biol*. 2012;28:251-77.

69. The title of this section is worded in the negative and would be clearer if worded as “SEPT2 limits the proteasomal degradation of HSPA5”

RESPONSE: The title has been modified as suggested.

70. Please provide some data from the HCS of the E3 ubiquitin ligases for HSPA5.

RESPONSE: A catalog of siRNA library for HCS has been provided in Supplementary Data 2. Seven E3 ubiquitin ligases [HSPA5::GFP intensity (fold change) > 2] were selected for further validation (data shown in Supplementary Fig. 7j).

71. Is the K327A mutant more stable than the WT? The authors should perform a CHX (cycloheximide)-chase assay in cells to answer this question.

RESPONSE: We thank the reviewer for this suggestion. We have performed a CHX-chase assay to evaluate the protein stability of the WT HSPA5 and K327A mutant. The results showed that the K327A mutant was more stable than the WT HSPA5. The corresponding revision is in Supplementary Fig. 7f, n and on Page 16, Line 349-351, Page 18, Line 382-384.

72. No data is presented to support the conclusion that “Abnormal degradation of HSPA5 leads to an exacerbation of unfolded proteins, resulting in M1-like hyperpolarization and excessive inflammation”. This sentence should be removed.

RESPONSE: This sentence has been removed as suggested.

Figure 3

73. Panel C is poorly controlled as addition of CHX will stop expression of HSPA5 irrespective of whether DOX was added or not. Please provide additional controls for this experiment.

RESPONSE: We thank the reviewer for this suggestion. We have added a control group without DOX treatment. The results showed that without using DOX to reduce SEPT2 expression, the degradation of HSPA5 was suppressed. In addition, we have included the data of three independent biological replicates in the form of line graph in the revised manuscript, and performed statistical analysis instead of showing a representative image. The corresponding revision is in Fig. 3e.

74. Panel D does not show ubiquitinated HSPA5. By performing a HSPA5 IP and then probing for ubiquitin, the authors are observing ubiquitinated species that could be on HSPA5 OR on proteins interacting with HSPA5. A better experiment would be to IP HSPA5 and perform USP21 cleavage on the IP eluate. Then, run these samples on a WB and probe for HSPA5. The authors should see laddering on their IP sample and if this is ubiquitin, the laddering would disappear in the DUB-cleavage samples. This would prove that HSPA5 is ubiquitinated. Another but less rigorous way to examine ubiquitinated HSPA5 would be to perform a ubiquitin IP (TUBE), then probe for HSPA5. The authors should see laddering or a smear with this probe.

RESPONSE: We thank the reviewer for pointing this out and agree with this assessment. We have repeated HSPA5 ubiquitination blots using the ubiquitin IP (TUBE) method in response to the reviewer's comments. We believe this revision will provide a more reliable argument.

75. Panel D also lacks an important control. In all IPs, the authors must probe for the

protein they are pulling down to show equal IP between samples. Otherwise, the increased smearing we are observing in lane 4 could simply be a consequence of the researcher unintentionally aliquoting more beads into that IP.

RESPONSE: We thank the reviewer for this suggestion. We have now included controls to demonstrate equivalence of IPs between samples (Fig. 3f of the revised manuscript).

76. Panel E lacks Dox-only and SCNN1B siRNA transfection only controls. Please amend and I suggest presenting the data in such a way that permits a statistical analysis to be performed. The authors should provide western blot verification of this data, including relevant controls (e.g. SCNN1B knockdown efficiency, SEPT2 levels etc).

RESPONSE: In accordance with the reviewer's suggestion, we have supplemented DOX-only and SCNN1B siRNA-only controls in the experiments. In order to show statistical analyses, data collected from three biological replicates were presented in the form of line graph. SCNN1B knockdown efficiency and SEPT2 expression levels have been verified by western blotting. The corresponding revision is in Fig. 3h, Supplementary Fig. 7k.

77. Panel F/G/I – please probe for HIS or HSPA5 to show equal IP.

RESPONSE: We thank the reviewer for this suggestion. We have included controls to show the equivalence of IPs between samples (Fig. 3i, j and m of the revised manuscript).

78. Panel H could be better labelled to highlight that you are looking at HSPA5.

RESPONSE: We have modified this figure in accordance as suggested by the reviewer (Fig. 3l of the revised manuscript).

79. I could not see a methods section for the mass spectrometry data. Please provide this and detail the conditions used to generate the sample for Mass spec.

RESPONSE: Detailed methods and sample preparation procedures for mass spectrum have been provided in the Supplementary Information on Page 11, Line 222-233: PMs obtained from *Sept2^{fl/fl}* and *Sept2^{fl/fl} Lyz2-Cre* mice were infected with VSV (MOI = 1) for 12 hours. To obtain the protein samples, cells were lysed with lysis buffer (8 M urea containing protease inhibitor cocktail) and centrifuged at 12,000×g at 4°C for 10 min. The supernatant was treated with trypsin at a ratio of 1:50 overnight for digestion. Anti-lysine acetylation antibody beads and anti-lysine ubiquitin antibody beads were used to enrich acetylated and ubiquitinated peptides, respectively. Next, enriched peptides were separated by the EASY-nLC HPLC system and detected by tandem mass spectrometry in the Q-Exactive mass spectrometer

(Thermo Fisher Scientific). Data were analyzed using Mascot software. The increase of 42.01 Da and 114.1 Da of lysine residue was calculated to determine acetylation and ubiquitination, respectively.

Figure S4

80. Panel A is an important result and I believe should be in the main figure.

RESPONSE: We thank the reviewer for this suggestion and this panel has been moved to Fig. 3a of the revised manuscript.

81. Panel I – This blot is empty? The authors should run an input sample alongside their IP samples to show that the antibodies incubations and detection has been performed correctly. This observation could equally be explained if the incorrect secondary antibody was added.

RESPONSE: We thank the reviewer for this suggestion. We have provided controls to confirm that the experiment was performed correctly (Supplementary Fig. 7i of the revised manuscript).

82. Panel H/I/L – please probe for HSPA5 to show that the IP has worked equally across all samples.

RESPONSE: We thank the reviewer for this suggestion. We have included controls to show the equivalence of IPs between samples (Supplementary Fig. 7h, i and m of the revised manuscript).

SEPT2 promotes the acetylation of HSPA5 by recruiting ATAT1

83. Examining the acetylation of HSPA5 followed a non-conventional logic, but the data appears sound.

RESPONSE: We thank the reviewer for the kind words pertaining to this part of the work.

84. Does the acetylation defective mutant HSPA5 have a reduced interaction with SCNN1B?

RESPONSE: To address this question, we have detected the interaction between K327A mutant of HSPA5 and SCNN1B (Fig. 3m of the revised manuscript). The results showed that K327A mutation did not affect the interaction of HSPA5-SCNN1B complex.

85. Do Sept2 and ATAT1 phenocopy one another if you compare them side-by-side?

RESPONSE: We thank the reviewer for this comment. SEPT2 and ATAT1 are two

very different proteins with distinct functions. SEPT2 belongs to the SEPTIN family that is considered a novel component of the cytoskeleton, while ATAT1 is an acetyltransferase. In this study, both SEPT2 and ATAT1 are indispensable for the successful acetylation of HSPA5. To be specific, ATAT1 is an acetyltransferase that directly acetylates HSPA5, while SEPT2 helps to recruit ATAT1 and form the ATAT1-HSPA5 complex. In this context, although the phenotypes (acetylation levels of HSPA5) in SEPT2-deficient cells and ATAT1-deficient cells are similar, based on the current experimental evidences, we cannot draw a conclusion that SEPT2 and ATAT1 phenocopy one another. This is an interesting question that needs further verification through experiments.

Figure S5A

86. All IPs should clearly indicate what probes are from input/whole cell lysate samples and which are from the IP. Furthermore, all IPs should be probed with the protein you are pulling down to demonstrate equal IP per sample. All IPs should have an input probe of the protein they are pulling down and a loading control.

RESPONSE: We thank the reviewer for this suggestion. We have repeated the ubiquitination blots using the TUBE method and have included the IP probe, the input probe, as well as the loading control to show the equivalence of IPs between samples.

87. Panels E/F are only explored at baseline and not during VSV infection. It would be interesting to show that this is not the case under settings where SEPT2 is known to regulate HSPA5.

RESPONSE: Indeed, we agree with the reviewer that it would be interesting to detect the ubiquitination of both substrates under VSV infection. However, WDTC1 and GRK2 were only used as random substrates to verify that the E3 ubiquitin ligase function of SCNN1B was not impaired by SEPT2 knockout, and further exploration of their relationship with SCNN1B and SEPT2 is beyond the scope of this study. Therefore, we decided not to include this part in the revised manuscript and hope our justification is thus suitable.

88. Please provide a lower exposure of the ubiquitin probe in panel G. There might be subtle difference not captured in the overexposed image.

RESPONSE: We thank the reviewer for this suggestion and have replaced panel G with a new one (Supplementary Fig. 8g of the revised manuscript). The result confirmed that there were almost no differences among the lanes.

89. Panel J – please see my comments for figure S4 panel I (above).

RESPONSE: In accordance with the reviewer's suggestion, we have provided controls to confirm that the experiment was performed correctly (Supplementary Fig.

8k of the revised manuscript).

90. Panel K – IMPORTANT – There are more GAPDH bands than HSPA5, Ac-Lysine bands. Conclusions can't be made from this data.

RESPONSE: We thank the reviewer for identifying this unfortunate error. In order to avoid edge effects of SDS-PAGE, we usually load a random sample to the edge lane (if there is an extra empty lane); however, we forgot to remove the random sample when plotting this figure. We have modified this figure in the revised manuscript (Supplementary Fig. 8I). The uncropped image has been provided in the “Source Data” file.

91. Panel P figure legend lacks detail. Furthermore, the legends states “...p65 regulated by ATAT1 in iBMDMs...” however, ATAT1 is clearly not regulating p65 acetylation. Do the authors mean VSV infection?

RESPONSE: The legend of this figure was not correctly described in the original manuscript and we apologize for the confusion. ATAT1 did not regulate the acetylation of p65 and we have modified this issue in the revised manuscript. In addition, we have included more details in the legend. The corresponding revision is on Page 67, Line 1553-1556: ATAT1 cannot regulate the acetylation of NF- κ B p65. WT iBMDMs were transfected with V5-tagged ATAT1 for 24 hours, followed by VSV infection (MOI = 1) for 12 hours. The acetylation of p65 was detected by western blotting. n = 3 in each group.

92. Line 366: I think the authors mean “native-PAGE”, instead of “naïve-PAGE”

RESPONSE: This word has been corrected as suggested.

ATAT1 has a stronger affinity for HSPA5 than SCNN1B

93. The interaction between SCNN1b and HSPA5 appears weak. Is their interaction direct or do you think there is more going on here such as another interacting protein or PTM? Perhaps a point of discussion?

RESPONSE: We thank the reviewer for this insightful comment and agree that although we have showed that SCNN1B interacts with HSPA5 directly (Fig. 3i, 5f), this direct interaction between SCNN1B and HSPA5 appear to be weak (Fig. 5h). We hypothesize that other interacting proteins or posttranslational modifications may be involved in the formation of this complex. We have also included this discussion in the revised manuscript on Page 33, Line 721-724.

Figure S6

94. Are these AlphaFold renders of the interaction or are they crystallised constructs?

I am struggling to believe SCNN1B directly interacts with HSPA5 due to the high Kd. It is possible there is another unknown member of the complex that co-localises SCNN1B and HSPA5. This possibility has not been excluded.

RESPONSE: The molecular structure docking was rendered by AlphaFold. As we have discussed in Comment #93, we cannot rule out the possibility that other interacting proteins or posttranslational modifications are involved in the formation of this complex in the *in vivo* system. We thank the reviewer for raising this question and have included this possibility in the Discussion section of the revised manuscript.

K327-acetylated HSPA5 prevents M1-like activation and excessive inflammation
95. Please show that the K327Q mutation actually stabilises HSPA5 and provide citations to papers that demonstrate K>Q mutations mimic acetylation.

RESPONSE: We thank the reviewer for this comment. In accordance with your suggestion from Comment #71, we have supplemented CHX-chase assay to prove that K327Q mutation actually stabilized HSPA5 (Supplementary Fig. 7f, Fig. 6c). The papers that demonstrate K>Q mutation mimics acetylation (listed below) have been cited in the revised manuscript on Page 35, Line 767-768.

Reference:

Chi Z, Chen S, Xu T, Zhen W, Yu W, Jiang D, Guo X, Wang Z, Zhang K, Li M, Zhang J, Fang H, Yang D, Ye Q, Yang X, Lin H, Yang F, Zhang X, Wang D. Histone Deacetylase 3 Couples Mitochondria to Drive IL-1 β -Dependent Inflammation by Configuring Fatty Acid Oxidation. *Mol Cell*. 2020 Oct 1;80(1):43-58.e7.

You Z, Jiang WX, Qin LY, Gong Z, Wan W, Li J, Wang Y, Zhang H, Peng C, Zhou T, Tang C, Liu W. Requirement for p62 acetylation in the aggregation of ubiquitylated proteins under nutrient stress. *Nat Commun*. 2019 Dec 19;10(1):5792.

96. PLA does not specifically look at ubiquitination of HSPA5, as other proteins in proximity of HSPA5 may also be ubiquitinated and cross-react in this assay. Please show that the K327Q reduces ubiquitination of HSPA5 by IP and western blot or more ideally quantitative mass spec.

RESPONSE: We thank the reviewer for this suggestion. We have included the results showing that the K327Q reduces HSPA5 ubiquitination by ubiquitin IP method in the revised manuscript. The corresponding revisions are in Fig. 6i, j, Supplementary Fig. 11h, i.

Figure 6

97. Panel B – What samples are being examined here? Please provide detail in the figure legend. An additional control showing the levels of HSPA5 in animals that do

not have the plasmid transfected are required to show that we aren't just looking at endogenous HSPA5.

RESPONSE: We thank the reviewer for this comment. The samples were lungs collected from *in vivo*-transfected mice without VSV infection at day 7. The sample information has been provided in the figure legend. Also, we have included additional controls to show the expression of exogenous plasmids in accordance with your suggestion. The corresponding revision is in Fig. 6b.

98. Stability and ubiquitination of HSPA5 K327Q needs to be demonstrated by IP and western blotting (6G/S7G).

RESPONSE: In accordance with your previous suggestions, we have included CHX-chase assay and ubiquitin IP to show the stability and ubiquitination of HSPA5 K327Q. The corresponding revision is in Fig. 6c, i, j, Supplementary Fig. 11h, i.

Figure S6

99. There appears to be a slight increase in the acetylation of WT-HSPA5 in *Sept2*^{-/-} cells upon infection and with *SCNN1b* knockdown. Do the authors think this is an artifact? This would imply there might be other activators of ATAT1.

RESPONSE: We thank the reviewer for pointing this out. By examining IP controls, we found that the slight increase in the acetylation of HSPA5 was indeed an artifact. We have repeated this experiment by using ubiquitin IP method, and the results showed that the acetylation of WT HSPA5 was not increased in *Sept2*^{fl/fl} *Lyz2*-Cre cells with *SCNN1B* knockdown (Supplementary Fig. 10f, g).

Discussion

100. Some of this discussion is copied almost word for word from the introduction.

RESPONSE: We thank the reviewer for this comment. We have carefully gone through and revised the Discussion section. We hope that the revised version will meet your requirements.

101. SEPT2 is not well characterised, although there is some literature around its roles in the cytoskeleton (doi: 10.1101/gad.11.12.1535) and mitochondrial fission (doi: <https://doi.org/10.15252/embr.201541612>). The authors should consider discussing these other facets of SEPT2 biology and the possibility of these biological processes influencing the regulation of HSPA5.

RESPONSE: We thank the reviewer for this valuable suggestion. We have included content regarding the other facets of SEPT2 biology and discussed the possibility of their interaction with HSPA5 in the Discussion section of revised manuscript. The corresponding revision is on Page 31, Line 662-668, Page 34, Line 736-745.

Responses to comments from Reviewer #2:

Comments: This is a comprehensive manuscript that investigates the role of the SEPT2 in antiviral activity by controlling IFN- γ -independent M1 polarization. The authors showed that SEPT2, which induces by ER stress, balances the competition between acetylation and ubiquitination of HSPA5 at position Lysine 327, thereby alleviating ER stress and constraining M1-like polarization and proinflammatory cytokine release. The study is interesting and my concerns are listed below.

RESPONSE: We thank Reviewer #2 for the positive evaluation of our paper. We appreciate the time and effort invested by the reviewer to provide insightful comments and suggestions regarding our study. To address your concerns, we have supplemented experiments on the regulatory role of SEPT2 in bacterial infection in the revised manuscript. We have also examined the effects of SEPT2 deletion on IFN- γ secretion, apoptosis and macrophage development. In addition, a number of control experiments have been added as suggested by the reviewer. In our point-by-point responses below, we hope to provide satisfactory responses to all concerns and comments raised by the reviewer.

1. Almost all infections can cause inflammatory response, such as virus and bacteria. The authors should confirm whether SEPT2 could play a similar role in the inflammatory response caused by bacteria.

RESPONSE: We thank the reviewer for this valuable comment. The role of SEPT2 in inflammatory response varies depending on different bacterial infections. For example, we used *Escherichia coli* (MOI = 20), *Listeria monocytogenes* (MOI = 20), *Mycobacterium tuberculosis* (MOI = 10), and the TLR4 receptor agonist LPS (100 ng/mL) to infect/stimulate *Sept2^{fl/fl} Lyz2-Cre* PMs for 12 hours. The expression levels of iNOS, CD80 and CD86 were detected by flow cytometry. The results indicated that there was no significant difference in M1-like polarization between SEPT2-deficient and wild-type macrophages upon *E.coli* and *Listeria* infections or LPS stimulation (Supplementary Fig. 3a, b). However, SEPT2 deletion in *M.tuberculosis* infection led to excessive macrophage activation (Supplementary Fig. 3a, b). It would be interesting to further investigate the underlying mechanisms. We have supplemented these results in the revised manuscript. The corresponding revision is on Page 10, Line 213-219.

2. In line 111, the authors said that SEPT2 participates in antiviral activity by controlling IFN- γ -independent M1 polarization. Please investigate whether SEPT2 can affect the production of interferon after RNA or DNA virus infection.

RESPONSE: We appreciate this insightful comment. We have included experiments to detect the secretion of IFN- γ in the BALF obtained from *Sept2^{fl/fl} Lyz2-Cre* and *Sept2^{fl/fl}* mice infected with VSV or HSV-1. The results showed that SEPT2 did not affect the production of IFN- γ after RNA or DNA virus infection. This data has been included in the Results section. The corresponding revision is in Supplementary Fig.

le and on Page 8, Line 162-164.

3. In Fig. 1D, the authors demonstrated that macrophages were increased in *Sept2*^{fl/fl} *Lyz2*-Cre mice, and the result also showed that M1-like macrophage cluster were increased, but M2-like macrophage cluster were no difference. These results seemed to indicate that the number of macrophage had increased in *Sept2*^{fl/fl} *Lyz2*-Cre mice. Please confirm that whether SEPT2 can affect the development of macrophage.

RESPONSE: We thank the reviewer for this comment. In order to address this concern, we have included experiments to measure the numbers of macrophages in spleen and lung derived from *Sept2*^{fl/fl} *Lyz2*-Cre and *Sept2*^{fl/fl} mice. The results showed that SEPT2 did not affect the development of macrophage in mice without viral infection. These data have been included in the revised manuscript. The corresponding revision is in Supplementary Fig. 1c, d and on Page 8, Line 156-160.

4. ER stress can induce apoptosis. The author had better confirm whether SEPT2 can affect the apoptosis of macrophages.

RESPONSE: We thank the reviewer for this suggestion. Considering it together with Reviewer #1's Comment #56, we have included experiments to detect apoptosis and pyroptosis in VSV-infected *Sept2*^{fl/fl} *Lyz2*-Cre and *Sept2*^{fl/fl} PMs. The results indicated that there was no significant difference in cell viability between WT and SEPT2-deficient cells at 0-24 hours post VSV infection (Supplementary Fig. 3c). Further, we found that there was no significant difference in pyroptosis and apoptosis at 12 hours post infection (Supplementary Fig. 3d, e). Since the duration of *in vitro* infection in this study did not exceed 12 hours, these data suggested that SEPT2 may not affect the apoptosis of macrophages at the early stage of viral infection. We have provided these data on Page 10-11, Line 220-227.

5. In Fig. 2B and 3D and so on, add the protein band of SEPT2 please, which used to confirm KO efficiency. The similar concern is involved in Fig. 4F and 7F.

RESPONSE: We thank the reviewer for this suggestion. We have included the protein band of SEPT2 to confirm the KO efficiency in Fig. 2b, 3a, 3f, 4a, 5c, 6b, 6i, Supplementary Fig. 7h and Supplementary Fig. 8i of the revised manuscript. The protein bands of ATAT1 (Fig. 4g, Supplementary Fig. 8n) and sXBP1 (Fig. 7h) have also been added in the paper.

6. In Fig. 2G and H, lack of control group, the author should add extra *Sept2*^{fl/fl} group with knock-down related genes or not. The similar concern is involved in Fig. 6C-E.

RESPONSE: We appreciate this constructive comment. We have added control groups in Fig. 2h, i, as suggested by the reviewer. The results showed that suppression

of one pathway by siRNA barely affected the total iNOS activity (Fig. 2h) and proinflammatory cytokine release (Fig. 2i) in wild-type cells. We speculate that this phenomenon may be specific to the condition of SEPT2 deficiency. To confirm this hypothesis, we performed western blotting to examine the p-eIF2 α level when PERK, IRE1 or ATF6 was knockdown in both *Sept2*^{fl/fl} *Lyz2*-Cre and *Sept2*^{fl/fl} PMs, The results showed that suppression of any pathway could inhibit p-eIF2 α level in SEPT2 deletion cells rather than in wild-type cells (Supplementary Fig. 5i). In this context, SEPT2 may exert a potential influence on the redundancy among the pathways; however, this needs to be verified further experimentally.

As for Fig. 6d (Fig. 6C of the original manuscript), we have included *Sept2*^{fl/fl} groups in this experiment. However no mice died in the *Sept2*^{fl/fl} + HSPA5 WT and *Sept2*^{fl/fl} + HSPA5 K327Q groups (probably due to the presence of SEPT2, endogenous HSPA5 could control the excessive polarization). Therefore, we decided not to include control groups in the subsequent experiments (Fig. 6e-g). We hope that this modification will meet your requirements.

7. In Fig. S4A and S4G, VSV infection could reduce HSPA5 protein in *Sept2*^{fl/fl} *Lyz2*-Cre macrophages, why is there no difference in Fig. S4E after VSV infection? Similar confusion in Fig. 6B, why is there no difference of HSPA5 protein level between *Sept2*^{fl/fl} and *Sept2*^{fl/fl} *Lyz2*-Cre group after transfected with wt-HSPA5?

RESPONSE: We thank the reviewer for drawing our attention to this point. We have therefore repeated this experiment in Fig. S4E. The latest result showed that the expression of HSPA5 was decreased after VSV infection in SEPT2 knockout cells. We speculate that the abnormality in the original manuscript might be due to the uneven exposure of western blotting. We have renewed this figure in Supplementary Fig. 7d of the revised manuscript.

As for Fig. 6b, the knockdown efficiency validation was performed without viral infection. Therefore, there was no significant difference in the expression of HSPA5 between the *Sept2*^{fl/fl} *Lyz2*-Cre and *Sept2*^{fl/fl} groups. In the revised manuscript, we have included more details about the experiment procedures in the legend and Methods section to avoid potential confusion.

8. In Fig. 4D, the result showed that ATAT1 could bind to HSPA5 only in the presence of SEPT2. However, the Fig. 4E and Fig. 5E, F showed that ATAT1 acetylated and interacted HSPA5 directly. These results are contradictory.

RESPONSE: We thank the reviewer for this insightful comment. This is indeed an interesting question, of which we were already aware. In the *in vitro* system, we found that ATAT1 did interact with HSPA5 directly (Fig. 4f, 5f, g). On the contrary, in the *in vivo* system, the priority of ATAT1 binding to HSPA5 had to be displayed in the presence of SEPT2 (Fig. 4e), suggesting a potential role of SEPT2 in ATAT1 recruitment. We hypothesize that under physiological conditions there may be a negative regulator that hinders the direct binding of ATAT1 to HSPA5, and that the

role of SEPT2 is to remove this negative regulator, allowing ATAT1 to bind preferentially to HSPA5. We believe that further IP-MS experiments may help to identify potential regulators; however, testing this hypothesis is time-consuming and exceeds the scope of our current study. Accordingly, we have included a discussion on this topic in the Discussion section (Page 33-34, Line 724-731).

9. SEPT2 balances the competition between acetylation (by ATAT1) and ubiquitination (by SCNN1B) of HSPA5 at the same position of Lysine 327. It seems like a dynamic and complex regulatory process. The authors had better explore whether these proteins (SEPT2, ATAT1, SCNN1B and HSPA5) could form a large protein complex by immunoprecipitation assay.

RESPONSE: We thank the reviewer for this suggestion. We have included experiments to test whether SEPT2, ATAT1, SCNN1B and HSPA5 could form a large protein complex. The results showed that SCNN1B was indeed exist in the ATAT1-HSPA5 complex in the presence of SEPT2 (Fig. 4e), suggesting that these proteins may form a large complex. However, this hypothesis needs to be verified by further experiments. We have included a discussion on this topic in the Discussion section on Page 34, Line 731-734.

10. The authors demonstrated that SEPT2 played an important role in antiviral activity by controlling IFN- γ -independent M1 polarization. Does SEPT2 have a similar effect if the IFN- γ is presented? Please validate or discuss this.

RESPONSE: We appreciate this constructive suggestion, which was also raised by Reviewer #1. As suggested by the reviewers, we have supplemented the infection experiments in the presence or absence of IFN- γ (with or without injection of IFN- γ antibody, Supplementary Fig. 4b-d). The results showed that SEPT2 modulated M1-like polarization and inflammation in the presence of IFN- γ . In this context, we showed that SEPT2-mediated UPR was independent of the IFN- γ pathway. The corresponding revision is on Page 11, Line 234-241.

Responses to comments from Reviewer #3:

Comments: There have been few studies on IFN γ independent macrophage activation. The role of septins in inflammation or viral infection is mostly unknown. Here the authors perform a tour de force, highlighting a new role for SEPT2 in antiviral activity (by controlling IFN γ -independent M1 polarization). The authors describe SEPT2 regulation of both VSV and HSV1; very little is known about septin control of viral infections. They show that ER stress induces SEPT2 expression, balancing acetylation vs ubiquitination of HSPA5 (at position lysine 327). If disrupted, unfolded proteins accumulate and accelerate M1-like polarization, leading to excessive inflammation and tissue damage. Together, authors conclude that SEPT2 can negatively regulate IFN γ independent macrophage proinflammatory activation, providing clinical insights into control of excessive inflammation.

RESPONSE: We thank the reviewer for the encouraging words and thoughtful suggestions on our paper. In accordance with your suggestions, we have supplemented experiments, discussions and references regarding SEPTIN family members in the revised manuscript. These contents would definitely help to deepen the understanding the role of SEPTIN family in inflammation and viral infections. In the following lines, we provide detailed responses to your concerns and suggestions. We thank the reviewer for taking time and effort to help us improve the paper, and we hope that our revisions will meet your requirements.

1. The septin cytoskeleton is widely viewed to act as hetero-oligomeric complexes with other septin family members. Have the authors other looked at other septins? Please highlight if / when other septins are involved. Or do the authors think SEPT2 is performing this role independently from other septins.

RESPONSE: We thank the reviewer for this interesting and insightful suggestion. Indeed, several studies have reported that SEPT2 can form hetero-oligomeric complexes with SEPT6, SEPT7 and SEPT9 (Low and Macara, 2006, Kim et al., 2011). Therefore, we have performed additional experiments to test whether these SEPTINs participated in the assembly of the SEPT2-HSPA5 complex. The results showed that SEPT6, SEPT7 and SEPT9 did not interact with HSPA5. These data indicated that SEPT2 performs the role in balancing HSPA5 acetylation and ubiquitination independently of other SEPTINs. We have supplemented these data in Supplementary Fig. 9d and on Page 21, Line 446-450.

Reference:

- Low C, Macara IG. Structural analysis of septin 2, 6, and 7 complexes. *J Biol Chem.* 2006 Oct 13;281(41):30697-706.
- Kim MS, Froese CD, Estey MP, Trimble WS. SEPT9 occupies the terminal positions in septin octamers and mediates polymerization-dependent functions in abscission. *J Cell Biol.* 2011 Nov 28;195(5):815-26.

2. What is the precise role of SEPT2 / the septin cytoskeleton? Septins have long been considered to act as a scaffolding protein. The authors suggest that SEPT2 expression balances acetylation vs ubiquitination of HSPA5, but no microscopy / cell biology studies are performed to visualise this. Is SEPT2 interacting with the ER? Is this role specific to macrophages? Does SEPT2 perform this function as a filament or higher order structure?

RESPONSE: We thank the reviewer for this comment. The *in situ* Duolink PLA assay (Fig. 5e, 6h, Supplementary Fig. 10h and Supplementary Fig. 11g) had been used to show that the acetylation and ubiquitination of intracellular HSPA5 were regulated by SEPT2. In order to further visualize the precise role of SEPT2, we examined the interaction between SEPT2 and HSPA5, as well as the colocalization of SEPT2 and ER (using SEC61B as an ER marker) by immunofluorescence. The results showed that VSV infection could enhance the colocalization of SEPT2 and HSPA5, as expected; however, the colocalization of ER and SEPT2 was not significantly changed upon infection (Supplementary Fig. 9h).

Next, in order to investigate whether the described role of SEPT2 is specific to macrophages, we performed SEPT2 knockdown in NIH-3T3, L929 and TC-1 cell lines. We found that SEPT2 knockdown resulted in an increase in HSPA5 ubiquitination in all cell types, but not necessarily accompanied with a decrease in acetylation levels (Supplementary Fig. 10c). Therefore, we speculate that the role of SEPT2 in balancing HSPA5 acetylation and ubiquitination might be specific in macrophages.

Furthermore, we have confirmed that SEPT2 could form homo-oligomeric complex when performing its function (Supplementary Fig. 9a-c). However, according to the results shown in Supplementary Fig. 9h, no apparent filamentation was observed in the interaction between SEPT2 and HSPA5. We have included the above data in the revised manuscript. The corresponding revisions are on Page 21-22, Line 441-446, 459-465, Page 23, Line 487-491.

3. Considering their crucial role in cellular physiology, it seems unlikely that septin expression would change +/- infection. As far as I know, significant changes in septin expression have not been described in the literature (except in the case of cancer studies)? Are these changes in septin expression described here at the transcriptional or protein level? What triggers the change in septin expression may be outside the scope of this manuscript. Quantify changes in expression as measured by immunoblot. Also confirm this change in SEPT2 expression by looking at other septins. If septin expression is significantly changing, how does this impact the other roles of septins (which are ubiquitously expressed) such as cell division, mitochondrial fission, etc.

RESPONSE: We appreciate this valuable comment and agree with the reviewer that the expression of SEPTINs is relatively stable in most cases. In this study, we showed that SEPT2 expression was increased in viral-infected macrophages at both transcriptional and protein levels (Supplementary Fig. 12e showed transcriptional

level and Fig. 7f showed protein level). We have also included quantitative analysis affiliated with immunoblots to confirm these changes (Fig. 7g). The expression of SEPTINs is relatively stable in most cell lines, as in these cases the role of SEPTINs is closely associated with cytoskeleton (similar to Tubulin), and constant expression levels of SEPTINs contribute to cell proliferation and division. In this study, we found that SEPT2 plays an important role in controlling inflammatory response and avoiding tissue damage caused by excessive inflammation in macrophages. The specific function of SEPT2 is quite different from the cytoskeleton, and we speculate that under different conditions, the expression level of SEPT2 will change to maintain homeostasis. In addition, in order to examine whether changes in SEPT2 expression would affect the normal functions of other SEPTINs, we have included experiments to test the expression levels of *Septin 5, 6, 7, 9, 10, 11* (which are ubiquitously expressed and functionally associated with cell division and mitochondrial fission). The results showed that VSV-induced SEPT2 upregulation barely affected the expression of other SEPTINs in macrophages except for SEPT6 and SEPT9 (Supplementary Fig. 9e). Since SEPT6 and SEPT9 were not involved in the assembly of the SEPT2-HSPA5 complex (Supplementary Fig. 9d), we speculate that they may not participate in the SEPT2 deficiency-induced M1-like hyperpolarization directly. In accordance with the reviewer's suggestion, we have included experiments to examine whether upregulation of SEPT6 and SEPT9 impacts other roles of SEPTINs. The results showed that knockdown of SEPT6 or SEPT9 had no significant effect on the mitochondrial fission (we did not test the effect on cell division as macrophages are terminally differentiated cells) in macrophages (Supplementary Fig. 9g). These data confirmed our hypothesis that SEPT2 plays a role in balancing HSPA5 acetylation and ubiquitination independently of other SEPTINs. We have supplemented the above results on Page 21, Line 450-459.

4. For all figures, representative images are often shown. Where relevant please highlight the number of times an experiment was performed, quantify data across replicates and perform statistics. If not possible to quantify data across replicates, please provide replicate images as supplemental files so that readers can see all the data.

RESPONSE: We thank the reviewer for this suggestion. We have included the statistical analysis of western blotting and flow cytometry data in the revised manuscript. In the histogram of statistical data, the individual points represent data obtained from biological replicates.

REVIEWER COMMENTS

Reviewer #1 (Remarks to the Author):

I commend the authors for their efforts. Other than the remaining comments below, my concerns have been addressed.

Results:

- Comment 45: I thank the authors for their explanation.
- Comment 55: The authors have fleshed out this data in great detail. I think this data is much improved from the initial submission, however the interpretation is complicated by the fact that siRNA knockdown of any PRR seems to affect the variables measured (in Supplemental Figure 2 N-S, V,W). I would only say that the incorrect comparisons are being made. To determine that the SEPT2 effect is independent of the PRRs, the magnitude of change between NC-siRNA and X-siRNA groups should be the same between the Sept2^{fl/fl} and Sept2^{Lyz2^{cre}} groups. For ease of interpretation/reading, I suggest labelling the datasets with the infection present in each dataset.
- Comment 59: Please confirm whether the repeated probes are from the same lysates as the original submission? Only some of the probes have been repeated and I find it unlikely that the same lysates have given such starkly different results. Note: PERK, eIF2 α , p-eIF2 α , IRE1 α and cleaved ATF6 α are the same in both figures. In good scientific practice, all probes in a Western blot panel should reflect different probes from the exact same sample. Otherwise, showing the GAPDH as a loading control is meaningless as it only applies to the repeated probes – presuming that the GAPDH is the same lysate as the other repeated probes. If they are the exact same lysates, please explain how the GAPDH and p-PERK probings look starkly different to the original submission (this is a source of concern).
- Comment 66: The authors should use the most appropriate p-value. For this kind of data, I believe the adjusted p-value is a more appropriate measure to plot here. I should clarify that my initial concern was simply that it is rare to see so many pathways with almost identically significant Adj. p-values and was querying it with the authors in case there was an error.
- Comment 96: Regarding Supplemental Figure 11h and Figure 6i – Why is there less HSPA5/FLAG in the input sample and a lot in the IP? The predominant band in the IP is the unmodified HSPA5/FLAG, so one would expect to see a similar band in the input sample. The only explanation I can think of is that the researcher took an 'input' sample from the lysates after the IP had been performed.

Reviewer #2 (Remarks to the Author):

The authors have answered almost all my concerns. However, there are two questions still should be elucidate.

1. In the revised manuscript, the authors showed that SEPT2 did not affect the production of IFN- γ after RNA or DNA virus infection. As type I interferon (such as IFN- α and IFN- β) and its downstream genes (such as ISGs and MX1 and so on) are important for antiviral response. So, the type I interferon and its downstream genes should be confirmed.
2. As LPS + IFN- γ was a type of classical manner to induce M1 macrophage polarization, and it might be different with the IFN- γ -independent M1 polarization. In this revised manuscript, the authors showed that SEPT2 both modulated M1-like polarization and inflammation with or without IFN- γ presence. The authors should demonstrated whether SEPT2 performed the same function in this two process (with or without IFN- γ presence).

Reviewer #3 (Remarks to the Author):

I appreciate the authors have advanced an enormous amount of work, but in my case the concerns have not been appropriately addressed.

1. The authors have not shown whether SEPT2 is working alone or in a complex with other septins. Perform IP against SEPT2 and test if pulls down with (i) other septins (as septins are known to interact with each other in macrophages) and (ii) HSPA5. Also perform immunofluorescence experiments to test if SEPT7 (core septin complex member) can localise with SEPT2 and repeat Figure S9h. Ideally at a better resolution. Please adjust septin complex conclusions accordingly. In the absence of compelling evidence I strongly recommend to leave the door open for either scenario.

The authors suggest that septin filamentation is not observed. This cannot be concluded from immunofluorescence studies, especially at this resolution Figure S9h. Please remove any associated text from the manuscript.

2. SEPT2, SEPT6, SEPT9 changes in expression upon VSV infection (FigS9e), suggesting they work together in a complex. Has this been tested by WB? Does the upregulation of SEPT2 have any impact on macrophage biology other than HSPA5? (only SEPT6, SEPT9 was tested; both are septins viewed as not essential for all septin function). SEPT2 is well known to induce mitochondrial fission (Pagliuso et al, EMBO Reports 2016). Please perform the appropriate experiments and/or adjust conclusions accordingly.

Responses to comments from Reviewer #1:

Comments: I commend the authors for their efforts. Other than the remaining comments below, my concerns have been addressed.

RESPONSE: We appreciate the kind words and valuable comments from the reviewer, and this manuscript has been greatly improved as a result of your valuable feedback. In the following lines, we detail our responses to each of your comments. We hope that the revised paper will meet your requirements.

1. Comment 45: I thank the authors for their explanation.

RESPONSE: We thank the reviewer for this approval.

2. Comment 55: The authors have fleshed out this data in great detail. I think this data is much improved from the initial submission, however the interpretation is complicated by the fact that siRNA knockdown of any PRR seems to affect the variables measured (in Supplemental Figure 2 N-S, V,W). I would only say that the incorrect comparisons are being made. To determine that the SEPT2 effect is independent of the PRRs, the magnitude of change between NC-siRNA and X-siRNA groups should be the same between the Sept2^{fl/fl} and Sept2^{Lyz2cre} groups. For ease of interpretation/reading, I suggest labelling the datasets with the infection present in each dataset.

RESPONSE: We thank the reviewer for this suggestion. Accordingly, we have modified the comparisons in the Supplemental Fig. 2n-s, Supplemental Fig. 3e, f and we have also labelled the panels with the infection of virus present in each dataset. We hope that these improvements will increase the readability of the paper.

3. Comment 59: Please confirm whether the repeated probes are from the same lysates as the original submission? Only some of the probes have been repeated and I find it unlikely that the same lysates have given such starkly different results. Note: PERK, eIF2a, p-eIF2a, IRE1a and cleaved ATF6a are the same in both figures. In good scientific practice, all probes in a Western blot panel should reflect different probes from the exact same sample. Otherwise, showing the GAPDH as a loading control is meaningless as it only applies to the repeated probes – presuming that the GAPDH is the same lysate as the other repeated probes. If they are the exact same lysates, please explain how the GAPDH and p-PERK probings look starkly different to the original submission (this is a source of concern).

RESPONSE: We thank the reviewer for pointing this out. The repeated probes were not from the same lysates as the original submission. We agree with the reviewer that all probes in a Western blot panel should be from the same sample, and therefore we have replaced all the probes in the original submission. In the revised paper, all the probes were from the same lysates obtained during the last revision, and the results showed that ER stress pathways are more highly activated in SEPT2-deficient PMs

than in control PMs. The corresponding revisions are in Fig. 2b, c.

4. Comment 66: The authors should use the most appropriate p-value. For this kind of data, I believe the adjusted p-value is a more appropriate measure to plot here. I should clarify that my initial concern was simply that it is rare to see so many pathways with almost identically significant Adj. p-values and was querying it with the authors in case there was an error.

RESPONSE: We thank the reviewer for this comment. Indeed, the adjusted p-value should be a more appropriate measure to plot here, we have therefore replaced the data with the adjusted p-value (Supplemental Fig. 5b).

5. Comment 96: Regarding Supplemental Figure 11h and Figure 6i – Why is there less HSPA5/FLAG in the input sample and a lot in the IP? The predominant band in the IP is the unmodified HSPA5/FLAG, so one would expect to see a similar band in the input sample. The only explanation I can think of is that the researcher took an ‘input’ sample from the lysates after the IP had been performed.

RESPONSE: We thank the reviewer for this comment and apologize for our carelessness as we did take the input sample from the lysates after the IP was performed. We have repeated the input probes with the correct samples and have replaced Supplemental Fig. 11h and Fig. 6i with new ones.

Responses to comments from Reviewer #2:

Comments: The authors have answered almost all my concerns. However, there are two questions still should be elucidate.

RESPONSE: We thank the reviewer for the nice words and kind suggestions on the revised manuscript. In response to your comments, we have investigated the type I interferon response and the SEPT2 function in the presence of IFN- γ . Below, we provide detailed responses to each of your comments. We hope that our revisions will meet your requirements.

1. In the revised manuscript, the authors showed that SEPT2 did not affect the production of IFN- γ after RNA or DNA virus infection. As type I interferon (such as IFN- α and IFN- β) and its downstream genes (such as ISGs and MX1 and so on) are important for antiviral response. So, the type I interferon and its downstream genes should be confirmed.

RESPONSE: We thank the reviewer for this comment. We agree with the reviewer that type I interferon and its downstream genes are important for antiviral response. In fact, our manuscript has showed that the function of SEPT2 is independent of the type I interferon pathway by knockdown of type I IFN receptor subunits IFNAR1 or IFNAR2 (Supplemental Fig. 3c-f). According to the suggestion from the reviewer, we have supplemented experiments to further confirm the type I interferon (IFN- α , IFN- β) and its downstream genes. The results showed that the production of IFN- α , IFN- β , MX1 and ISG15 were not affected by SEPT2 deletion (Supplemental Fig. 3a, b). We have included this data in the revised manuscript. The corresponding revision is on Page 10, Line 210-218.

2. As LPS + IFN- γ was a type of classical manner to induce M1 macrophage polarization, and it might be different with the IFN- γ -independent M1 polarization. In this revised manuscript, the authors showed that SEPT2 both modulated M1-like polarization and inflammation with or without IFN- γ presence. The authors should demonstrated whether SEPT2 performed the same function in this two process (with or without IFN- γ presence).

RESPONSE: We thank the reviewer for this valuable suggestion, accordingly we have performed SEPT2 knockdown in iBMDMs with or without IFN- γ presence and detected the acetylation and ubiquitination of HSPA5 in the revised manuscript. The results showed that SEPT2 could also balance the HSPA5 modification upon IFN- γ stimulation, indicating the same function of SEPT2 in the presence or absence of IFN- γ (Supplemental Fig. 10c). The corresponding revision is on Page 24, Line 512-513.

Responses to comments from Reviewer #3:

Comments: I appreciate the authors have advanced an enormous amount of work, but in my case the concerns have not been appropriately addressed.

RESPONSE: We regret that the last revision did not meet your satisfaction and thank you very much for offering us further comments. We have carefully considered your suggestions and supplemented relevant experiments to reveal the relationship between the role of SEPT2 and other SEPTINs, as well as the impact of SEPT2 upregulation on the macrophage biology. Some of the conclusions have been modified based on the latest results. In the following lines, we provide detailed responses to your comments and requests. We hope that these revisions meet your requirements.

1. The authors have not shown whether SEPT2 is working alone or in a complex with other septins. Perform IP against SEPT2 and test if pulls down with (i) other septins (as septins are known to interact with each other in macrophages) and (ii) HSPA5. Also perform immunofluorescence experiments to test if SEPT7 (core septin complex member) can localise with SEPT2 and repeat Figure S9h. Ideally at a better resolution. Please adjust septin complex conclusions accordingly. In the absence of compelling evidence I strongly recommend to leave the door open for either scenario. The authors suggest that septin filamentation is not observed. This cannot be concluded from immunofluorescence studies, especially at this resolution Figure S9h. Please remove any associated text from the manuscript.

RESPONSE: We thank the reviewer for this insightful and thought-provoking comment. We agree with the reviewer that IP against SEPT2 is a direct way to test whether SEPT2 works alone. In fact, we did perform IP against SEPT2 during the first round of revision. Since SEPT2 has been reported to form hetero-oligomeric complexes generally with SEPT6, SEPT7 and SEPT9 (Low and Macara, 2006, Kim et al., 2011), it is not surprising that the results showed that SEPT2 interacts with SEPT6, SEPT7, SEPT9 and HSPA5 when we performed IP against SEPT2. However, multiprotein complexes do not allow definitive conclusions to be drawn from coimmunoprecipitation, in which case we could neither confirm whether these SEPTINs (SEPT6, SEPT7 and SEPT9) are actually present in the SEPT2-HSPA5 complex, nor whether these hetero-oligomeric complexes are involved in SEPT2-regulated macrophage activation from the positive results of IP against SEPT2. This was the reason that we performed IP against HSPA5 to indirectly to determine whether SEPT2 works alone or in a complex with other SEPTINs.

In this context, we have included the results of IP against both SEPT2 and HSPA5 in the revised manuscript (Supplemental Fig. 9d, e). Considering that we could not yet conclude whether SEPT6, SEPT7 and SEPT9 are involved in SEPT2-regulated macrophage proinflammatory activation, we designed specific siRNAs to knock down these SEPTINs and examined their effects on M1-like polarization (Supplemental Fig. 9f). Interestingly, we found that knockdown of SEPT7, instead of SEPT6 and SEPT9, promoted M1-like hyperpolarization after viral infection (Supplemental Fig. 9g). Further, we performed additional immunofluorescence experiments to test the

localization of SEPT2 and SEPT7 in accordance with your suggestion. The results showed that SEPT2 and SEPT7 did co-localise, however the co-localisation did not change upon VSV infection (Supplemental Fig. 9h). These data suggest that SEPT7 may not directly participate in the SEPT2-HSPA5 complex. According to a previous report, depletion of SEPT7 induces partial codepletion of SEPT2 (Pagliuso et al., 2016). We then examined whether knockdown of SEPT6, SEPT7 and SEPT9 would affect the expression of SEPT2. The results showed that the expression of SEPT2 was downregulated with SEPT7 knockdown (Supplemental Fig. 9i). In this case, we speculate that knockdown of SEPT7 may promote M1-like hyperpolarization by indirectly reducing SEPT2. However, this hypothesis needs to be verified by further experiments. We have adjusted the conclusion on SEPTIN complex accordingly in the revised manuscript. The corresponding revision is on Page 21-22, Line 453-473. Moreover, we repeated the immunofluorescence experiments in Supplemental Fig. 9m (Supplemental Fig. 9h in the first round of revision) to obtain a better resolution in accordance with the reviewer's suggestion. The results showed that VSV infection could enhance the co-localization of SEPT2 and HSPA5 as expected; however, the co-localization of ER and SEPT2 was not significantly changed upon infection (Supplemental Fig. 9m). In addition, we decided to remove the SEPTIN filamentation section from the revised manuscript as suggested by the reviewer. We hope these modifications meet your requirements.

References:

- Low C, Macara IG. Structural analysis of septin 2, 6, and 7 complexes. *J Biol Chem.* 2006 Oct 13;281(41):30697-706.
- Kim MS, Froese CD, Estey MP, Trimble WS. SEPT9 occupies the terminal positions in septin octamers and mediates polymerization-dependent functions in abscission. *J Cell Biol.* 2011 Nov 28;195(5):815-26.
- Pagliuso A, Tham TN, Stevens JK, Lagache T, Persson R, Salles A, Olivo-Marin JC, Oddos S, Spang A, Cossart P, Stavru F. A role for septin 2 in Drp1-mediated mitochondrial fission. *EMBO Rep.* 2016 Jun;17(6):858-73.

2. SEPT2, SEPT6, SEPT9 changes in expression upon VSV infection (FigS9e), suggesting they work together in a complex. Has this been tested by WB? Does the upregulation of SEPT2 have any impact on macrophage biology other than HSPA5? (only SEPT6, SEPT9 was tested; both are septins viewed as not essential for all septin function). SEPT2 is well known to induce mitochondrial fission (Pagliuso et al, EMBO Reports 2016). Please perform the appropriate experiments and/or adjust conclusions accordingly.

RESPONSE: We thank the reviewer for this comment. In accordance with your suggestion, we have validated this data by western blotting (Supplemental Fig. 9k). Indeed, SEPT2, SEPT6, SEPT9 changes in expression upon VSV infection; however, according to the result shown in Supplemental Fig. 9g, knockdown of SEPT6 and SEPT9 did not affect M1-like hyperpolarization after viral infection, suggesting that

SEPT6 and SEPT9 may not work together in a complex with SEPT2.

In addition, we have supplemented SEPT2 knockdown assay and investigated the effect of changes in SEPT2 expression on mitochondria fission. The results showed that knockdown of SEPT2 significantly decreased the mitochondrial fission rate (Supplemental Fig. 9I), which is consistent with previous data that SEPT2 depletion induces mitochondrial elongation (Pagliuso et al., 2016). Further, we found that the magnitudes of change in mitochondrial fission rate by SEPT2 siRNA treatment were the same between control and VSV infection groups (Supplemental Fig. 9I). This suggests that the impact on mitochondrial fission by SEPT2 knockdown may not be closely linked to the M1-like hyperpolarization process.

We have modified the related conclusion accordingly in the revised manuscript. The corresponding revision is on Page 22-23, Line 474-485.

Reference:

Pagliuso A, Tham TN, Stevens JK, Lagache T, Persson R, Salles A, Olivo-Marin JC, Oddos S, Spang A, Cossart P, Stavru F. A role for septin 2 in Drp1-mediated mitochondrial fission. *EMBO Rep.* 2016 Jun;17(6):858-73.

REVIEWERS' COMMENTS

Reviewer #2 (Remarks to the Author):

The concerns have been appropriately addressed.

Reviewer #3 (Remarks to the Author):

Thank you for addressing all comments and for your interesting contribution to this exciting field.

Responses to comments from Reviewer #2:

Comments: The concerns have been appropriately addressed.

RESPONSE: Thank you for your approval of our revisions.

Responses to comments from Reviewer #3:

Comments: Thank you for addressing all comments and for your interesting contribution to this exciting field.

RESPONSE: Thank you for your kind words and the recognition of our study.